# Signal Strength Estimation in Logistic Regression Using Data Splitting

**Weihao Li** [1]  **Jun S. Liu** [1]

## Abstract

Logistic regression is widely used in applications; however, when the dimension scales with the sample size, theory reveals that the asymptotic behavior of common M-estimators depends on nonzero bias and variance factors, which are functions of the signal strength. To leverage the theory to design valid statistical methodologies, it is essential to obtain accurate estimates of the signal strength. In this work, we utilize a data-splitting strategy to efficiently estimate the signal strength. To alleviate issues caused by separable data, we analyze the exact asymptotics of an M-estimator with a data-driven, non-decomposable regularizer that adapts to the true covariance structure. We justify the validity of our method through both theoretical analysis and numerical experiments.

## 1. Introduction

Logistic regression is widely used in data science and many applied areas. In practice, the dimension of a data set ($p$) may be large and may scale with the sample size ($n$). Under such a moderately high-dimensional regime, the classical maximum likelihood estimation (MLE) can be biased or may not even exist (Sur & Candès, 2019). Consider i.i.d. observations $\{(\boldsymbol{X}_i, Y_i)\}_{i=1}^n$ with Gaussian feature vectors $\boldsymbol{X}_i \sim N(\boldsymbol{0}, \boldsymbol{\Sigma})$, where the responses $Y_i \in \{0, 1\}$. Let $\boldsymbol{\beta}_*$ denote the true regression coefficient and define the signal strength as $\kappa^2 := \mathrm{Var}(\boldsymbol{X}^\top \boldsymbol{\beta}_*)$. The conditional distribution of $Y_i \mid \boldsymbol{X}_i$ is given by

$$\mathbb{P}(Y_i = 1) = \rho'(\boldsymbol{X}_i^\top \boldsymbol{\beta}_*),$$

where $\rho(t) = \log(1 + \exp(t))$.

A widely used estimator for $\boldsymbol{\beta}_*$ is the MLE $\widehat{\boldsymbol{\beta}}$. In the moderate-dimensional regime where $n/p = \delta \in (2, \infty)$,

---

[1]Department of Statistics and Data Science, Tsinghua University, Beijing, China. Correspondence to: Jun S. Liu <junsliu@tsinghua.edu.cn>.

*Proceedings of the 43rd International Conference on Machine Learning*, Seoul, South Korea. PMLR 306, 2026. Copyright 2026 by the author(s).

Sur & Candès (2019) showed that when the design is isotropic with $\boldsymbol{\Sigma} = \mathbf{I}_p$ and the MLE exists, $\widehat{\boldsymbol{\beta}}$ is approximately distributed as

$$\widehat{\boldsymbol{\beta}} \approx \alpha_* \boldsymbol{\beta}_* + \sigma_* N\left(0, \frac{1}{p}\mathbf{I}_p\right), \tag{1}$$

where $\alpha_*$ and $\sigma_*$ (together with $\tilde{r}_*$) solve the following system of equations for given $\kappa$ and $\delta$:

$$\begin{cases} \dfrac{\sigma^2}{2\delta} = \mathbb{E}\left[\rho'\left(-\kappa Z_1\right)\left(Q(\alpha, \sigma) - \mathrm{Prox}_{\tilde{r}\rho}\left(Q(\alpha, \sigma)\right)\right)^2\right] \\[2mm] -\dfrac{\alpha}{2\delta} = \mathbb{E}\left[\rho''\left(-\kappa Z_1\right)\mathrm{Prox}_{\tilde{r}\rho}\left(Q(\alpha, \sigma)\right)\right] \\[2mm] 1 - \dfrac{1}{\delta} = 2\mathbb{E}\left[\dfrac{\rho'\left(-\kappa Z_1\right)}{1 + \tilde{r}\rho''\left(\mathrm{Prox}_{\tilde{r}\rho}\left(Q(\alpha, \sigma)\right)\right)}\right], \end{cases} \tag{2}$$

with $Q(\alpha, \sigma) := \kappa \alpha Z_1 + \sigma Z_2$ and $\mathrm{Prox}_{\tilde{r}\rho}(x) = \arg\min_t [\rho(t) + \frac{1}{2\tilde{r}}(x - t)^2]$ and $Z_1, Z_2 \sim N(0, 1)$. Equivalently, $\alpha_*, \sigma_*$ can be viewed as functions of both the sampling ratio $\delta$ and the signal strength $\kappa$. When the context is clear, we suppress the dependence on $\delta$ and $\kappa$ in the notation. We refer to $\alpha_*$ and $\sigma_*$ as the bias factor and variance factor, respectively. Extensions to general covariance matrices $\boldsymbol{\Sigma}$ have been studied in Zhao et al. (2022). Similar results have also been derived for regularized logistic regression with decomposable regularizers of the form $\lambda \sum_{j=1}^p g_j(\beta_j)$ when $\boldsymbol{\Sigma} = \mathbf{I}_p$ (Salehi et al., 2019), where the bias and variance factors depend additionally on the regularization parameter $\lambda$.

Building on Sur & Candès (2019), the asymptotic representation (1) enables the construction of confidence intervals for $\boldsymbol{\beta}_*$ even when $\widehat{\boldsymbol{\beta}}$ is biased. This representation is also useful for transfer learning tasks, such as computing cosine similarity between coefficient vectors across different data sources. To apply (1), one must estimate the constants $\alpha_*$ and $\sigma_*$. Since these constants solve a system of equations depending on the signal strength $\kappa$ and other known parameters (e.g., $\delta$), a natural approach is to first estimate $\kappa$ and then substitute it into the system. However, this is challenging because $\boldsymbol{\beta}_*$ is unknown, and consistent estimation of $\boldsymbol{\beta}_*$ is generally impossible in the moderate-dimensional regime without additional structural assumptions.

To estimate the signal strength, Sur & Candès (2019) propose a heuristic method called ProbeFrontier, building

on the MLE existence theory of Candès & Sur (2020). Yadlowsky et al. (2021) use a leave-one-out technique to estimate the limiting variance $\lim_{n\to\infty} \mathrm{Var}(\boldsymbol{X}^\top \widehat{\boldsymbol{\beta}}) = \kappa^2 \alpha_*^2 + \sigma_*^2$, and then reparameterize (2) to eliminate the dependence on the signal strength. Both approaches lack rigorous theoretical justification and fail when the MLE does not exist, a frequent occurrence when $\kappa$ is large or the sample size is small. Bellec (2025) propose a more involved construction to estimate $\alpha_*$ for $\kappa = 1$, but their method requires knowledge of the true covariance matrix $\boldsymbol{\Sigma}$ when the data are separable[1].

At a high level, the signal strength estimation strategies of Sur & Candès (2019) and Yadlowsky et al. (2021) follow a common principle: identify an empirically tractable proxy $\eta$ with a one-to-one theoretical relationship to the signal strength (i.e., $\eta = f(\kappa)$), estimate $\eta$ to obtain $\hat{\eta}$, and recover $\kappa$ by inverting the map, $\hat{\kappa} = f^{-1}(\hat{\eta})$. Specifically, Yadlowsky et al. (2021) use the corrupted signal strength $\eta = \sqrt{\kappa^2 \alpha_*^2(\kappa) + \sigma_*^2(\kappa)}$, while Sur & Candès (2019) adopt a subsampling approach to identify the critical boundary of $n/p$ where the MLE exists with probability $1/2$, and then invoke the phase transition theory of Candès & Sur (2020) to recover the signal strength.

Consistent estimation of $\boldsymbol{\beta}_*$ is generally impossible when the dimension scales with the sample size without additional structural assumptions. Beyond this fundamental limitation, the unknown covariance structure poses further challenges. When the observed data are separable and the MLE does not exist, regularization is typically introduced to stabilize estimation. However, commonly studied regularizers in the moderate-dimensional regime do not preserve the covariance structure of $\boldsymbol{X}$. For example, existing theory for the decomposable regularizer $\lambda \|\boldsymbol{\beta}\|_2^2$ only consider $\boldsymbol{\Sigma} = \mathbf{I}_p$. Furthermore, we cannot consistently estimate $\boldsymbol{\Sigma}$ when dimension scales with sample size (Johnstone & Lu, 2009).

In summary, existing approaches to signal strength estimation when the dimension scales with the sample size in logistic regression suffer from at least one of three limitations: (1) they rely on heuristic principles without rigorous justification, (2) they fail when the data are separable and the MLE does not exist, or (3) they require knowledge of the true covariance matrix $\boldsymbol{\Sigma}$, which is unavailable in practice. Missing from the literature is a theoretically principled and covariance-adaptive procedure for estimating $\kappa$ that remains valid across a wide range of signal strengths and sampling ratios. The present work addresses this gap.

---

[1] Data are said to be separable if there exists a coefficient vector $\boldsymbol{\beta}$ such that $(2Y_i - 1)\boldsymbol{X}_i^\top \boldsymbol{\beta} > 0$ for all $i$, in which case the logistic log-likelihood is unbounded and the MLE does not exist.

## 1.1. Contributions

The contributions of our work are threefold.

**First**, we rigorously analyze a generalized ridge estimator $\widehat{\boldsymbol{\beta}}_\lambda$ that uses the sample covariance matrix as the weighting matrix when the dimension scales with the sample size. The analysis of this generalized ridge falls outside the scope of existing analyses, such as those in Salehi et al. (2019), because the regularization cannot be written as $\sum_{j=1}^p g_j(\beta_j)$. We show that (i) the proposed regularization guarantees the existence of the estimator when $\delta > 1$, and (ii) the asymptotic normality of the generalized ridge estimator. Specifically, defining the projection matrix $P = \boldsymbol{\beta}_* \boldsymbol{\beta}_*^\top / \|\boldsymbol{\beta}_*\|^2$ we have $\mathbf{P}\widehat{\boldsymbol{\beta}}_\lambda \approx \alpha_* \boldsymbol{\beta}_*$ and $(\mathbf{I} - \mathbf{P})\widehat{\boldsymbol{\beta}}_\lambda \approx \sigma_* N(0, \mathbf{I}/p)$, where $(\alpha_*, \sigma_*)$ are function of $\kappa$ and regularization level $\lambda$.

**Second**, we investigate the relationship between $\kappa^2 \alpha_*^2 + \sigma_*^2$ and $\kappa$. Our results reveal that this mapping is not one-to-one: distinct values of $\kappa$ can produce identical values of $\kappa^2 \alpha_*^2 + \sigma_*^2$. Consequently, this quantity alone does not uniquely determine $\kappa$, highlights a limitation of previously proposed strategies, such as those in Yadlowsky et al. (2021). Moreover, we show that (i) as the signal strength $\kappa$ increases, the $\sigma_*$ monotonically decreases, and the decreasing speed becomes larger when $n/p$ is smaller; and (ii) as the signal strength $\kappa$ increases, $\kappa^2 \alpha_*^2$ is monotonically increasing. These observations suggest that (i) there is a one-to-one relationship between $\sigma_*^2$ and $\kappa$, and (ii) there is a one-to-one relationship between $\kappa^2 \alpha_*^2$ and $\kappa$. Therefore, if either $\kappa^2 \alpha_*^2$ or $\sigma_*^2$ can be consistently estimated, then $\kappa$ can be consistently estimated.

**Third**, building on these theoretical insights, we propose a simple data-splitting strategy to estimate the variance factor $\sigma_*^2$ and $\kappa^2 \alpha_*^2$. We show a one-to-one relationship between these quantities and the signal strength, which enables estimation of $\kappa$. We further show that the proposed signal strength estimators are consistent.

## 1.2. Related work

Our work is related to three main areas: asymptotic theory for M-estimation in moderate dimensions, estimation of bias and variance factors, and data-splitting methods.

**Asymptotics in moderate dimensions.** Over the past two decades, substantial progress has been made in characterizing the asymptotic behavior of MLEs and regularized estimators under regimes where the ratio $n/p$ converges to a constant. There is a rich literature on M-estimators in linear models (Bayati & Montanari, 2011b; El Karoui et al., 2013; El Karoui, 2018; Thrampoulidis et al., 2018; Celentano et al., 2023), binary regression models (Sur & Candès, 2019; Salehi et al., 2019; Taheri et al., 2020; Deng et al., 2022; Zhao et al., 2022; Sterzinger & Kosmidis,

2026), generalized linear models (Dai et al., 2023b), and more general teacher–student models (Loureiro et al., 2021). The main technical tools underlying these developments include approximate message passing (AMP) (Donoho et al., 2009; Bayati & Montanari, 2011a), the Convex Gaussian Min–Max Theorem (CGMT) (Thrampoulidis et al., 2015; 2018), and leave-one-out analysis (El Karoui et al., 2013; El Karoui, 2018). Our asymptotic characterization builds on the CGMT framework and extends it to estimators with non-decomposable, data-driven regularization.

**Estimation of bias and variance factors.** Accurate estimation of bias and variance factors in moderate dimensions is essential for inference tasks such as confidence interval construction and squared error prediction. In linear models, Bayati et al. (2013); Miolane & Montanari (2021) use Stein's unbiased risk estimator to estimate noise levels in Lasso regression, while El Karoui (2018) study variance in ridge-regularized generalized robust regression. Bellec (2023) derive generic estimators for squared error of M-estimators with convex penalties, and Celentano et al. (2023) use leave-one-out techniques to estimate noise levels in Lasso. For logistic regression, bias and variance factors are typically obtained by first estimating the signal strength and then plugging it into the corresponding system of equations (Sur & Candès, 2019; Yadlowsky et al., 2021). Bellec (2025) provide estimators of bias and variance factors in single-index models when the signal strength is fixed at one. Our work proposes an alternative strategy to estimate the signal strength even when the data are separable.

**Data splitting.** Data splitting has a long history in statistical methodology, beginning with its use in assessing predictive performance (Moran, 1973; Stone, 1974; Cox, 1975). It has since been widely adopted to address complex dependence structures in statistical inference (Wasserman & Roeder, 2009). Data splitting has also been used extensively in variable selection procedures (Rubin et al., 2006; Ignatiadis et al., 2016; Dai et al., 2023a). More recently, Dai et al. (2023b) study variable selection in moderate dimensions and use data splitting to bypass direct estimation of bias and variance factors. In contrast, our work leverages data splitting to consistently estimate bias and variance factors themselves, which in turn enables consistent inference of the signal strength.

## 2. Overview of theory and estimation method

We begin by analyzing an estimator based on a generalized ridge regularizer with regularization level $\lambda$. Under an isotropic design, this estimator admits an asymptotic representation analogous to (1), with bias and variance factors characterized by the solution to a different system of equations that depends on $(\delta, \kappa, \lambda)$. Building on our investigation of the theoretical relationships among these quantities, we

select $\eta = \sigma_*$ and $\eta = \kappa^2 \alpha_*^2$ as proxy quantities of interest and use them to indirectly infer the signal strength $\kappa$.

To estimate $\kappa$, we exploit the fact that when the ratio $\delta = n/p$ is reduced to $\delta/2$, the estimator becomes more variance-inflated, which in turn induces a clearer relationship between $\eta$ and $\kappa$. Our estimation of $\sigma_*(\delta/2)$ is based on a data-splitting strategy. Specifically, when the original data are split into two equal-sized halves, the generalized ridge estimator satisfies the following asymptotic relations:

$$\widehat{\boldsymbol{\beta}}_\lambda^{(1)} \approx \alpha_*(\delta/2)\boldsymbol{\beta}_* + \sigma_*(\delta/2)\boldsymbol{Z}^{(1)}$$
$$\widehat{\boldsymbol{\beta}}_\lambda^{(2)} \approx \alpha_*(\delta/2)\boldsymbol{\beta}_* + \sigma_*(\delta/2)\boldsymbol{Z}^{(2)}$$

where $(\alpha_*(\delta/2), \sigma_*(\delta/2))$ is the solution to the corresponding system of equations with parameters $(\delta/2, \kappa, \lambda)$, and $\boldsymbol{Z}^{(1)}$ and $\boldsymbol{Z}^{(2)}$ are independent noise vectors distributed as $N(0, \frac{1}{p}\mathbf{I}_p)$. This representation motivates the estimator

$$\widehat{\sigma_*}(\delta/2) := \frac{1}{\sqrt{2}} \left\| \widehat{\boldsymbol{\beta}}_\lambda^{(1)} - \widehat{\boldsymbol{\beta}}_\lambda^{(2)} \right\|_2,$$

which yields a consistent estimator of $\sigma_*(\delta/2)$. Estimation of another proxy quantity will be introduced in Section 5.

The remainder of the paper is organized as follows. Section 3 analyzes the asymptotic behavior of the generalized ridge estimator in moderate dimensions and characterizes its bias and variance factors through a new system of equations. Section 4 examines two candidate proxies for the signal strength $\kappa$ and explains why the proxy used in Yadlowsky et al. (2021) fails when the data are separable. Section 5 presents the data-splitting procedures for estimating the proxy quantities and inferring the signal strength. Numerical studies are reported in Section 6.

## 3. A generalized ridge estimator for separable data

According to Candès & Sur (2020), data become separable when $\kappa$ is large. For example, when $\kappa = 3$ and $\delta = 3$, all 100 independently generated data sets are separable (see Appendix A.1 for additional experiments).

To address the data separability issue, we introduce regularization. To account for the covariance structure of $\boldsymbol{X}$ in the regularization, if $\boldsymbol{\Sigma}$ is known, we could use $\lambda\boldsymbol{\beta}^\top\boldsymbol{\Sigma}\boldsymbol{\beta}$. Then, following the results in Salehi et al. (2019), we would be able to understand the asymptotic behavior of $\boldsymbol{\Sigma}^{1/2}\widehat{\boldsymbol{\beta}}$ as a function of the signal strength $\|\boldsymbol{\Sigma}^{1/2}\boldsymbol{\beta}_*\|^2$. However, $\boldsymbol{\Sigma}$ is unknown in practice. We therefore replace $\boldsymbol{\Sigma}$ with $\widehat{\boldsymbol{\Sigma}} = \frac{1}{n}\sum_{i=1}^n \boldsymbol{X}_i\boldsymbol{X}_i^\top$, which leads to the following estimator:

$$\widehat{\boldsymbol{\beta}}_\lambda = \arg\min_{\boldsymbol{\beta}} \frac{1}{n}\sum_{i=1}^n [\rho(\boldsymbol{X}_i^\top\boldsymbol{\beta}) - Y_i\boldsymbol{X}_i^\top\boldsymbol{\beta}] + \frac{\lambda}{2n}\sum_{i=1}^n (\boldsymbol{X}_i^\top\boldsymbol{\beta})^2.$$
$$(3)$$

The special formulation in (3) brings both challenges and benefits. The regularization term $\sum_{i=1}^{n}(\boldsymbol{X}_i^\top \boldsymbol{\beta})^2$ cannot be written as a sum of decomposable functions of $\beta_j$; therefore, we cannot apply the results in Salehi et al. (2019) directly. Beyond this theoretical challenge, there are two advantages in using such a regularization:

1. Under the condition $n/p > 1$, the minimum eigenvalue of $\boldsymbol{\Sigma}$ is bounded away from zero with high probability, which further guarantees that $\widehat{\boldsymbol{\beta}}_\lambda$ lies in a compact set. This provides a sufficient condition for using the CGMT to analyze the asymptotic behavior of $\widehat{\boldsymbol{\beta}}_\lambda$.

2. It suffices to analyze the case $\boldsymbol{\Sigma} = \boldsymbol{I}_p$; the theory for a general $\boldsymbol{\Sigma}$ then follows from the techniques in Zhao et al. (2022).

To understand the relationship between $\widehat{\boldsymbol{\beta}}_\lambda$ and $\boldsymbol{\beta}_*$, we decompose $\widehat{\boldsymbol{\beta}}_\lambda$ as follows. Define $\mathbf{P} = \boldsymbol{\theta}_*\boldsymbol{\theta}_*^\top / \|\boldsymbol{\theta}_*\|^2$ with $\boldsymbol{\theta}_* := \boldsymbol{\Sigma}^{1/2}\boldsymbol{\beta}_*$ and $\mathbf{P}^\perp = \boldsymbol{I}_p - \mathbf{P}$.

$$
\begin{aligned}
\boldsymbol{\Sigma}^{1/2}\widehat{\boldsymbol{\beta}}_\lambda &= \mathbf{P}\boldsymbol{\Sigma}^{1/2}\widehat{\boldsymbol{\beta}}_\lambda + \mathbf{P}^\perp\boldsymbol{\Sigma}^{1/2}\widehat{\boldsymbol{\beta}}_\lambda \\
&= \frac{\boldsymbol{\beta}_*^\top\boldsymbol{\Sigma}\widehat{\boldsymbol{\beta}}_\lambda}{\|\boldsymbol{\Sigma}^{1/2}\boldsymbol{\beta}_*\|^2}\boldsymbol{\Sigma}^{1/2}\boldsymbol{\beta}_* + \|\mathbf{P}^\perp\boldsymbol{\Sigma}^{1/2}\widehat{\boldsymbol{\beta}}_\lambda\|_2\boldsymbol{h},
\end{aligned}
\tag{4}
$$

where $\boldsymbol{h}$ denotes the direction of $\mathbf{P}^\perp\boldsymbol{\Sigma}^{1/2}\widehat{\boldsymbol{\beta}}_\lambda$ and can be viewed as a noisy direction. We define $\alpha(p) := \frac{\boldsymbol{\beta}_*^\top\boldsymbol{\Sigma}\widehat{\boldsymbol{\beta}}_\lambda}{\|\boldsymbol{\Sigma}^{1/2}\boldsymbol{\beta}_*\|^2}$ and $\sigma(p) := \|\mathbf{P}^\perp\boldsymbol{\Sigma}^{1/2}\widehat{\boldsymbol{\beta}}_\lambda\|_2$ to be the bias and variance factors, respectively. We will show below that the limits of these two factors are characterized by a system of equations.

The following theorem summarizes the performance of the regularized estimator.

**Theorem 3.1.** *Let $\lambda > 0$ be a fixed constant and assume $n/p = \delta > 1$. Then we have:*

1. *Assume there exists a universal constant $c$ such that $0 < c < \lambda_{min}(\boldsymbol{\Sigma})$ as $p \to \infty$. Then, with probability at least $1 - 2\exp(-\frac{1}{8}(1 - \sqrt{1/\delta})^2 n)$,*

$$
\|\widehat{\boldsymbol{\beta}}_\lambda\|^2 \le \frac{8\log 2}{c\lambda(1 - \sqrt{1/\delta})^2}.
$$

2. *Let $\|\boldsymbol{\Sigma}^{1/2}\boldsymbol{\beta}_*\|_2 = \kappa$ and denote by $(\alpha_*, \sigma_*, \tilde{r}_*)$ the*

*solution of the following system of equations:*

$$
\begin{cases}
\begin{aligned}
\frac{\sigma^2}{2\delta} =\ & \frac{\tilde{r}^2\lambda^2(\kappa^2\alpha^2 + \sigma^2)}{2} + \\
& \mathbb{E}\left[\rho'(-\kappa Z_1)\left(\tilde{Q}(\alpha,\sigma,\tilde{r}) - \operatorname{Prox}_{\tilde{r}\rho}(\tilde{Q}(\alpha,\sigma,\tilde{r}))\right)^2\right] \\
& + 2\lambda\tilde{r}\mathbb{E}\left[\rho'(-\kappa Z_1)\tilde{Q}(\alpha,\sigma,\tilde{r})Q(\alpha,\sigma)\right] \\
& - 2\lambda\tilde{r}\mathbb{E}\left[\rho'(-\kappa Z_1)\operatorname{Prox}_{\tilde{r}\rho}(\tilde{Q}(\alpha,\sigma,\tilde{r}))Q(\alpha,\sigma)\right]
\end{aligned} \\
-\frac{\alpha}{2\delta} = \mathbb{E}[\rho''(-\kappa Z_1)\operatorname{Prox}_{\tilde{r}\rho}(\tilde{Q}(\alpha,\sigma,\tilde{r})))] \\
1 - \frac{1}{\delta} = 2(1 - \lambda\tilde{r})E\left(\frac{\rho'(-\kappa Z_1)}{1 + \tilde{r}\rho''(\operatorname{Prox}_{\tilde{r}\rho}(\tilde{Q}(\alpha,\sigma,\tilde{r})))}\right)
\end{cases}
\tag{5}
$$

*Here $\tilde{Q}(\alpha,\sigma,\tilde{r}) := (1 - \lambda\tilde{r})[Q(\alpha,\sigma)]$ and $Q(\alpha,\sigma)$ and $\operatorname{Prox}$ are defined after (2). Then $(\alpha_*, \sigma_*, \tilde{r}_*)$ is unique and we have*

$$
\alpha(p) \xrightarrow{\mathbb{P}} \alpha_* \tag{6}
$$

$$
\sigma(p) \xrightarrow{\mathbb{P}} \sigma_* \tag{7}
$$

*Remark 3.2.* The first result indicates that when $\lambda$ is bounded away from zero, the resulting estimator $\widehat{\boldsymbol{\beta}}_\lambda$ lies in a compact set, and thus the tuning parameter needs to scale as a constant, i.e., $\lambda \propto 1$. This agrees with our intuition, as sufficient regularization is required to stabilize the estimation. By the Borel–Cantelli lemma, with $\tilde{C} = \frac{8\log 2}{c\lambda(1-\sqrt{1/\delta})^2}$, we have

$$
\mathbb{P}\left(\{\|\widehat{\boldsymbol{\beta}}_\lambda\|^2 > \tilde{C}\} \text{ happens infinitely many times}\right) = 0.
$$

This allows us to safely constrain the feasible set $\mathcal{S}_\beta$ to be a ball centered at the origin with a constant radius in $\mathbb{R}^p$ for all $p$. This result provides a necessary condition for applying the CGMT.

Result 2 in Theorem 3.1 characterizes the key relationship between $\widehat{\boldsymbol{\beta}}_\lambda$ and $\boldsymbol{\beta}_*$. Together with (4), we obtain

$$
\|\boldsymbol{\Sigma}^{1/2}\widehat{\boldsymbol{\beta}}_\lambda\|^2 \xrightarrow{\mathbb{P}} \alpha_*^2\kappa^2 + \sigma_*^2 \tag{8}
$$

$$
\|\boldsymbol{\Sigma}^{1/2}\widehat{\boldsymbol{\beta}}_\lambda - \boldsymbol{\Sigma}^{1/2}\boldsymbol{\beta}_*\|^2 \xrightarrow{\mathbb{P}} (\alpha_* - 1)^2\kappa^2 + \sigma_*^2 \tag{9}
$$

When $\lambda = 0$, the system of equations (5) reduces to (2), as expected. In Appendix A.2, we plot both the theoretical values and their empirical counterparts to verify the derived theory. Equation (9) implies that $\|\widehat{\boldsymbol{\beta}}_\lambda - \boldsymbol{\beta}_*\|_2$ is of constant order, which matches the minimax lower bound of order $O(p/n)$ for the estimation error in GLMs (Chen et al., 2016). This indicates that $\widehat{\boldsymbol{\beta}}_\lambda$ is inconsistent. However, by leveraging a data-splitting technique and this inconsistent estimator, we can consistently estimate the signal strength $\|\boldsymbol{\Sigma}^{1/2}\boldsymbol{\beta}_*\|_2$ (Section 5).

Equations (6) and (7) serve as an important bridge to further applications of the asymptotic theory. Following Theorems 3.1–3.3 in Zhao et al. (2022), we obtain the following asymptotic result:

**Proposition 3.3.** *Let $\lambda > 0$ be a fixed constant and assume $n/p = \delta > 1$. Let $(\alpha_*, \sigma_*, \tilde{r}_*)$ be the solution to the system of equations (5),*

*(1) For any fixed index set $\mathcal{S} \subset \{1, \ldots, p\}$, let $\boldsymbol{\Theta} = \boldsymbol{\Sigma}^{-1}$. Suppose that $\sqrt{p}\left(\boldsymbol{\beta}_{*,\mathcal{S}}^{\top}\boldsymbol{\Theta}_{\mathcal{S}}^{-1}\boldsymbol{\beta}_{*,\mathcal{S}}\right)^{\frac{1}{2}} = O(1)$, then*

$$\frac{\sqrt{p}\boldsymbol{\Theta}_{\mathcal{S}}^{-1/2}(\widehat{\boldsymbol{\beta}}_{\lambda,\mathcal{S}} - \alpha_*\boldsymbol{\beta}_{*,\mathcal{S}})}{\sigma_*} \xrightarrow{\mathrm{d}} N\left(\boldsymbol{0}, \boldsymbol{I}_{|\mathcal{S}|}\right).$$

*(2) Let $v_j^2 = \mathrm{Var}\left(X_{i,j} \mid \boldsymbol{X}_{i,-j}\right)$ denote the conditional variance of $X_{i,j}$ given all other covariates. Furthermore, assume that the empirical distribution $\frac{1}{p}\sum_{j=1}^{p}\chi_{\sqrt{p}v_j\beta_{*,j}}$ converges weakly to a distribution $\Pi$ with a finite second moment, and $\sum_{j=1}^{p} v_j^2 \beta_{*,j}^2 \xrightarrow{\mathbb{P}} \mathbb{E}\left[\beta^2\right]$ for $\beta \sim \Pi$. Then, for any locally Lipschitz function[2] $\Psi : \mathbb{R} \times \mathbb{R} \to \mathbb{R}$, or for the indicator function $\Psi(a, t) = \mathbf{1}\{|a/\sigma_*| \leq t\}$ with any fixed $t > 0$, we have*

$$\frac{1}{p}\sum_{j=1}^{p}\Psi\left(\sqrt{p}v_j(\widehat{\boldsymbol{\beta}}_{\lambda,j} - \alpha_*\boldsymbol{\beta}_{*,j}), \sqrt{p}v_j\boldsymbol{\beta}_{*,j}\right) \xrightarrow{\mathbb{P}} \mathbb{E}[\Psi(\sigma_*Z, \beta)].$$

*where $Z \sim N(0, 1)$ independent of $\beta \sim \Pi$.*

Part (1) establishes asymptotic normality for any fixed (finite-dimensional) subset of coordinates, enabling inference on small groups of coefficients. Part (2) provides a rigorous theorem characterizing the limiting behavior of the estimator $\widehat{\boldsymbol{\beta}}_\lambda$ in terms of bias and variance factors. The moment convergence assumptions in Part (2) are standard in the literature for settings where the dimension scales with the sample size (Bayati & Montanari, 2011b; Donoho et al., 2009; Javanmard & Montanari, 2013; Zhao et al., 2022).

These results enable two key applications. First, using Part (2) with $\Psi(a, t) = \mathbf{1}\{|a/\sigma_*| \leq t\}$, we construct adjusted confidence intervals for individual coefficients, as demonstrated in our numerical experiments (Section 6). Second, one can compute the limiting classification error of the plug-in rule $\widehat{Y} = \mathbf{1}\{\boldsymbol{X}_{\mathrm{new}}^{\top}\widehat{\boldsymbol{\beta}}_\lambda \geq 0\}$; details are deferred to Appendix F.

## 4. On the theoretical one-to-one relationship

Since the estimation of the signal strength $\kappa$ relies on finding a reliable proxy that has a monotone relationship with $\kappa$, this section begins by investigating the relationship between the proxy $\eta^2 = \kappa^2\alpha_*^2 + \sigma_*^2$ used in Yadlowsky et al. (2021). We show that the relationship between $\eta^2$ and $\kappa^2$ may not be monotone. We then investigate the relationship between $\sigma_*$ and $\kappa$ under different ratios $n/p$, and show that a reduced ratio $n/p$ leads to a sharper contrast between $\sigma_*$ and $\kappa$.

To solve the system of equations (5), we use a fixed-point

---

[2]A function $\Psi : \mathbb{R}^m \to \mathbb{R}$ is said to be locally-Lipschitz if there exists a constant $L > 0$ such that for all $\boldsymbol{t}_0, \boldsymbol{t}_1 \in \mathbb{R}^m$, $\|\Psi\left(\boldsymbol{t}_0\right) - \Psi\left(\boldsymbol{t}_1\right)\| \leq L\left(1 + \|\boldsymbol{t}_0\| + \|\boldsymbol{t}_1\|\right)\|\boldsymbol{t}_0 - \boldsymbol{t}_1\|.$

iterative method (Berinde, 2007, Ch. 1.2). The relationship between the proxy $\eta^2(\kappa) = \kappa^2\alpha_*^2(\kappa) + \sigma_*^2(\kappa)$ and signal strength $\kappa$ is shown in Figure 1, showing that the proxy $\eta^2(\kappa) = \kappa^2\alpha_*^2(\kappa) + \sigma_*^2(\kappa)$ does not exhibit a consistent one-to-one relationship with $\kappa$.

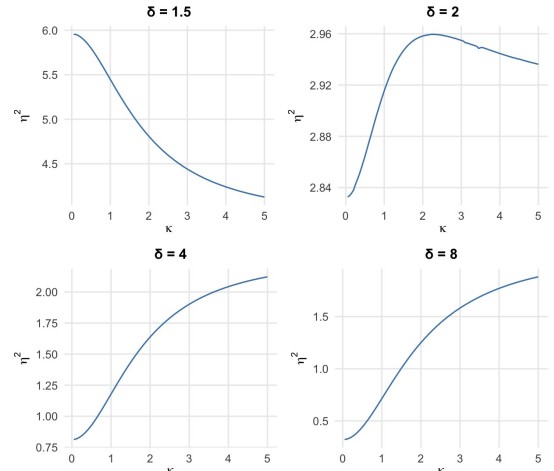

*Figure 1.* Relationship of $\kappa^2\alpha_*^2(\kappa) + \sigma_*^2(\kappa)$ against the value of $\kappa$. We fix $\lambda = 0.1$.

In particular, when $\delta = 2$, a single value of $\eta^2$ can correspond to two different values of $\kappa$. Equivalently, different values of $\kappa$ can lead to the same limiting value of $\|\boldsymbol{\Sigma}^{1/2}\widehat{\boldsymbol{\beta}}_\lambda\|^2$. This phenomenon suggests that the estimation strategy adopted in Yadlowsky et al. (2021) may not be valid in the small-$\delta$ regime for our estimator (3). Small $\delta$ is typical in high-dimensional settings and corresponds to scenarios where the MLE does not exist (Candès & Sur, 2020). Therefore, we seek another proxy for $\kappa$ that maintains a monotone relationship with $\kappa$.

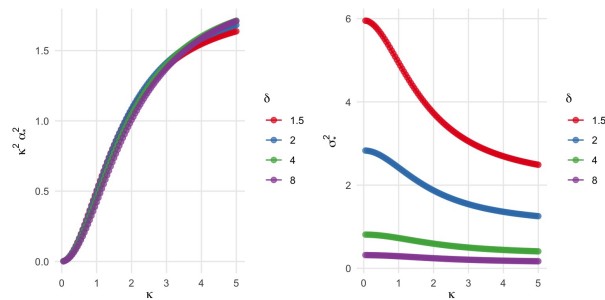

*Figure 2.* Left panel: $\kappa^2\alpha_*^2(\kappa)$ as a function of $\kappa$; right panel: $\sigma_*^2(\kappa)$ as a function of $\kappa$. We fix $\lambda = 0.1$.

We also plot the relationships between $\kappa^2\alpha_*^2(\kappa)$ and $\sigma_*^2(\kappa)$ as functions of $\kappa$ in Figure 2. This partially explains why the mapping between $\kappa^2\alpha_*^2(\kappa) + \sigma_*^2(\kappa)$ and $\kappa$ is not monotone. From the left panel, we observe that $\kappa^2\alpha_*^2(\kappa)$ does not vary substantially across different values of $\delta$. In contrast, the right panel shows that $\sigma_*(\kappa)$ varies significantly as $\delta$ changes. In particular, when $\delta$ is small, a given change in $\kappa$ leads to a much larger change in $\sigma_*(\kappa)$. The rapid decrease of $\sigma_*(\kappa)$ for small $\delta$ accounts for the non-monotone relationship between $\kappa^2\alpha_*^2(\kappa) + \sigma_*^2(\kappa)$ and $\kappa$.

Up to this point, we have made the following observations regarding different proxies for $\kappa$. (i) The quantities $\sigma_*^2(\delta, \kappa) := F_1(\delta, \kappa, \lambda)$ and $\kappa^2\alpha_*^2(\delta, \kappa) := F_2(\delta, \kappa, \lambda)$ are strictly monotone with respect to $\kappa$. (ii) The clearer separation observed for smaller values of $\delta = n/p$ suggests that estimating $F_1$ is preferable when the sample size is small. (iii) The similar patterns of $\kappa^2\alpha_*^2(\delta/2, \kappa)$ and $\kappa^2\alpha_*^2(\delta, \kappa)$ as functions of $\kappa$ suggest that either quantity may be used as a proxy.

These observations motivate us to estimate either $\sigma_*^2(\delta/2, \kappa)$ or $\kappa^2\alpha_*^2(\delta/2, \kappa)$. Once such an estimator is obtained, we can recover $\kappa$ via either $F_1^{-1}(\delta/2, \cdot, \lambda)$ or $F_2^{-1}(\delta/2, \cdot, \lambda)$. In the next section, we adopt a data-splitting approach to construct efficient and consistent estimators for both $\kappa^2\alpha_*^2(\delta/2, \kappa)$ and $\sigma_*^2(\delta/2, \kappa)$. For notational simplicity, we suppress the dependence on $\lambda$ in the proxy quantities in the subsequent section.

## 5. Data splitting to estimate proxy quantities

In this section, we adopt the data-splitting technique of Dai et al. (2023b) to provide simple estimators for both $\kappa^2\alpha_*^2(\delta/2, \kappa)$ and $\sigma_*^2(\delta/2, \kappa)$. To the best of our knowledge, no existing work provides consistent estimators of these quantities for logistic regression in the moderate-dimensional regime when covariance is unknown. For clarity of presentation, we first consider the isotropic design and then discuss the general covariance case at the end of this section.

### 5.1. Isotropic covariance $\mathbf{\Sigma} = \mathbf{I}_p$

We begin by splitting the data into two equal halves. Let the original data be $D = \{(\mathbf{X}_1, Y_1), (\mathbf{X}_2, Y_2), \ldots, (\mathbf{X}_n, Y_n)\}$. Without loss of generality, we define $D_1 = \{(\mathbf{X}_i, Y_i)\}_{i=1}^{n/2}$, and $D_2 = \{(\mathbf{X}_i, Y_i)\}_{i=n/2+1}^{n}$. Then, based on the decomposition in (4), we obtain the following results:

$$\widehat{\boldsymbol{\beta}}_\lambda^{(1)} = \frac{\boldsymbol{\beta}_*^\top\widehat{\boldsymbol{\beta}}_\lambda^{(1)}}{\|\boldsymbol{\beta}_*\|^2}\boldsymbol{\beta}_* + \|\mathbf{P}^\perp\widehat{\boldsymbol{\beta}}_\lambda^{(1)}\|_2\frac{\mathbf{P}^\perp\widehat{\boldsymbol{\beta}}_\lambda^{(1)}}{\|\mathbf{P}^\perp\widehat{\boldsymbol{\beta}}_\lambda^{(1)}\|_2}$$

$$\widehat{\boldsymbol{\beta}}_\lambda^{(2)} = \frac{\boldsymbol{\beta}_*^\top\widehat{\boldsymbol{\beta}}_\lambda^{(2)}}{\|\boldsymbol{\beta}_*\|^2}\boldsymbol{\beta}_* + \|\mathbf{P}^\perp\widehat{\boldsymbol{\beta}}_\lambda^{(2)}\|_2\frac{\mathbf{P}^\perp\widehat{\boldsymbol{\beta}}_\lambda^{(2)}}{\|\mathbf{P}^\perp\widehat{\boldsymbol{\beta}}_\lambda^{(2)}\|_2}$$

where $\widehat{\boldsymbol{\beta}}_\lambda^{(1)}$ denotes the estimator in (3) based on the data $D_1$. We define $\alpha^{(1)}(p) := \frac{\boldsymbol{\beta}_*^\top\widehat{\boldsymbol{\beta}}_\lambda^{(1)}}{\|\boldsymbol{\beta}_*\|^2}$ and $\sigma^{(1)}(p) := \|\mathbf{P}^\perp\widehat{\boldsymbol{\beta}}_\lambda^{(1)}\|_2$ as the bias and variance factors, respectively. Similarly, we define $\widehat{\boldsymbol{\beta}}_\lambda^{(2)}, \alpha^{(2)}(p), \sigma^{(2)}(p)$.

Since the projection matrix $\mathbf{P}^\perp = \mathbf{I}_p - \frac{\boldsymbol{\beta}_*\boldsymbol{\beta}_*^\top}{\|\boldsymbol{\beta}_*\|^2}$ projects a vector onto the subspace orthogonal to $\boldsymbol{\beta}_*$, we have

$$\widehat{\kappa^2\alpha_*^2}(\delta/2, \kappa) := \left\langle\widehat{\boldsymbol{\beta}}_\lambda^{(1)}, \widehat{\boldsymbol{\beta}}_\lambda^{(2)}\right\rangle = \alpha^{(1)}(p)\alpha^{(2)}(p)\|\boldsymbol{\beta}_*\|^2$$
$$+ \sigma^{(1)}(p)\sigma^{(2)}(p)\left\langle\frac{\mathbf{P}^\perp\widehat{\boldsymbol{\beta}}_\lambda^{(1)}}{\|\mathbf{P}^\perp\widehat{\boldsymbol{\beta}}_\lambda^{(1)}\|_2}, \frac{\mathbf{P}^\perp\widehat{\boldsymbol{\beta}}_\lambda^{(2)}}{\|\mathbf{P}^\perp\widehat{\boldsymbol{\beta}}_\lambda^{(2)}\|_2}\right\rangle.$$

Following Theorem 3.1, together with $\|\boldsymbol{\beta}_*\|_2 = \kappa$ and Slutsky's theorem, we obtain

$$\begin{aligned} \alpha^{(1)}(p) &\xrightarrow{\mathbb{P}} \alpha_*(\delta/2, \kappa) &, \quad \alpha^{(2)}(p) &\xrightarrow{\mathbb{P}} \alpha_*(\delta/2, \kappa) \\ \sigma^{(1)}(p) &\xrightarrow{\mathbb{P}} \sigma_*(\delta/2, \kappa) &, \quad \sigma^{(2)}(p) &\xrightarrow{\mathbb{P}} \sigma_*(\delta/2, \kappa) \end{aligned} \quad (10)$$

To establish that $\left\langle\widehat{\boldsymbol{\beta}}_\lambda^{(1)}, \widehat{\boldsymbol{\beta}}_\lambda^{(2)}\right\rangle \xrightarrow{\mathbb{P}} \kappa^2\alpha_*^2(\delta/2, \kappa)$, it suffices to show that the cross term vanishes

$$\left\langle\frac{\mathbf{P}^\perp\widehat{\boldsymbol{\beta}}_\lambda^{(1)}}{\|\mathbf{P}^\perp\widehat{\boldsymbol{\beta}}_\lambda^{(1)}\|_2}, \frac{\mathbf{P}^\perp\widehat{\boldsymbol{\beta}}_\lambda^{(2)}}{\|\mathbf{P}^\perp\widehat{\boldsymbol{\beta}}_\lambda^{(2)}\|_2}\right\rangle \xrightarrow{\mathbb{P}} 0. \quad (11)$$

Specifically, the following result holds.

**Theorem 5.1.** *Assume* $\boldsymbol{X}_i \sim N(0, \mathbf{I}_p)$*, then we have*

$$\frac{\mathbf{P}^\perp\widehat{\boldsymbol{\beta}}_\lambda^{(1)}}{\left\|\mathbf{P}^\perp\widehat{\boldsymbol{\beta}}_\lambda^{(1)}\right\|} \stackrel{d}{=} \frac{\mathbf{P}^\perp\boldsymbol{Z}^{(1)}}{\left\|\mathbf{P}^\perp\boldsymbol{Z}^{(1)}\right\|}$$

*where* $\boldsymbol{Z}^{(1)} \sim N(0, \mathbf{I_p})$.

The proof of Theorem 5.1 is provided in Appendix D. As a consequence, we obtain

$$\left\langle\frac{\mathbf{P}^\perp\widehat{\boldsymbol{\beta}}_\lambda^{(1)}}{\|\mathbf{P}^\perp\widehat{\boldsymbol{\beta}}_\lambda^{(1)}\|}, \frac{\mathbf{P}^\perp\widehat{\boldsymbol{\beta}}_\lambda^{(2)}}{\|\mathbf{P}^\perp\widehat{\boldsymbol{\beta}}_\lambda^{(2)}\|}\right\rangle \stackrel{d}{=} \left\langle\frac{\mathbf{P}^\perp\boldsymbol{Z}^{(1)}}{\|\mathbf{P}^\perp\boldsymbol{Z}^{(1)}\|}, \frac{\mathbf{P}^\perp\boldsymbol{Z}^{(2)}}{\|\mathbf{P}^\perp\boldsymbol{Z}^{(2)}\|}\right\rangle$$
$$= \frac{\sqrt{p}}{\|\mathbf{P}^\perp\boldsymbol{Z}^{(1)}\|}\frac{\sqrt{p}}{\|\mathbf{P}^\perp\boldsymbol{Z}^{(2)}\|}$$
$$\times \frac{1}{p}\left(\boldsymbol{Z}^{(1)\top}\boldsymbol{Z}^{(2)} - \frac{(\boldsymbol{\beta}_*^\top\boldsymbol{Z}^{(1)})(\boldsymbol{\beta}_*^\top\boldsymbol{Z}^{(2)})}{\|\boldsymbol{\beta}_*\|^2}\right) \xrightarrow{a.s.} 0$$

This proves (11) and hence $\left\langle\widehat{\boldsymbol{\beta}}_\lambda^{(1)}, \widehat{\boldsymbol{\beta}}_\lambda^{(2)}\right\rangle \xrightarrow{\mathbb{P}} \kappa^2\alpha_*^2(\delta/2, \kappa)$.

We now turn to the consistency of the variance estimator

introduced in Section 2.

$$\widehat{\sigma_*^2}(\delta/2,\kappa) := \frac{1}{2}\left\|\widehat{\boldsymbol{\beta}}_\lambda^{(1)} - \widehat{\boldsymbol{\beta}}_\lambda^{(2)}\right\|^2$$

$$= \frac{1}{2}[\alpha^{(1)}(p) - \alpha^{(2)}(p)]^2\|\boldsymbol{\beta}_*\|^2 + \frac{1}{2}\|\mathbf{P}^\perp\widehat{\boldsymbol{\beta}}_\lambda^{(1)} - \mathbf{P}^\perp\widehat{\boldsymbol{\beta}}_\lambda^{(2)}\|^2$$

$$= \frac{1}{2}[\alpha^{(1)}(p) - \alpha^{(2)}(p)]^2\|\boldsymbol{\beta}_*\|^2 + \frac{1}{2}[\sigma^{(1)}(p)]^2 + \frac{1}{2}[\sigma^{(2)}(p)]^2$$

$$\qquad + \sigma^{(1)}(p)\sigma^{(2)}(p)\left\langle \frac{\mathbf{P}^\perp\widehat{\boldsymbol{\beta}}_\lambda^{(1)}}{\|\mathbf{P}^\perp\widehat{\boldsymbol{\beta}}_\lambda^{(1)}\|_2}, \frac{\mathbf{P}^\perp\widehat{\boldsymbol{\beta}}_\lambda^{(2)}}{\|\mathbf{P}^\perp\widehat{\boldsymbol{\beta}}_\lambda^{(2)}\|_2}\right\rangle$$

$$\xrightarrow{\mathbb{P}} \sigma_*^2(\delta/2,\kappa),$$

where last convergence is a direct consequence of (10) and (11). The above derivation shows that our data-splitting–based estimator is consistent. To reduce variability, one may further aggregate estimates across multiple random splits. We recommend using 50 random splits in practice, the effect of the number of splits is investigated in Appendix A.6.

## 5.2. General covariance $\boldsymbol{\Sigma} \neq \mathbf{I}_p$

We now consider the setting with a general feature covariance matrix $\boldsymbol{\Sigma} \neq \mathbf{I}_p$. In this case, the signal strength is $\kappa^2 = \boldsymbol{\beta}_*^\top\boldsymbol{\Sigma}\boldsymbol{\beta}_*$. Accordingly, the definitions of the population-level quantities must be modified. The bias coefficient $\alpha^{(1)}(p)$ is redefined as $\frac{\boldsymbol{\beta}_*^\top\boldsymbol{\Sigma}\widehat{\boldsymbol{\beta}}_\lambda^{(1)}}{\|\boldsymbol{\Sigma}^{1/2}\boldsymbol{\beta}_*\|^2}$ and the noise level $\sigma^{(1)}(p)$ becomes $\|\left(\mathbf{I}_p - \frac{\boldsymbol{\Sigma}^{1/2}\boldsymbol{\beta}_*\boldsymbol{\beta}_*^\top\boldsymbol{\Sigma}^{1/2}}{\|\boldsymbol{\Sigma}^{1/2}\boldsymbol{\beta}_*\|^2}\right)\boldsymbol{\Sigma}^{1/2}\widehat{\boldsymbol{\beta}}_\lambda^{(1)}\|_2$. The key distinction between the general covariance case and the isotropic case lies in the construction of the estimators. In particular, the estimators $\widehat{\sigma_*^2}(\delta/2,\kappa)$ and $\widehat{\kappa^2\alpha_*^2}(\delta/2,\kappa)$ now explicitly depend on the unknown covariance matrix $\boldsymbol{\Sigma}$. They are given by $\widehat{\sigma_*^2}(\delta/2,\kappa) = \frac{1}{2}\left\|\boldsymbol{\Sigma}^{1/2}\widehat{\boldsymbol{\beta}}_\lambda^{(1)} - \boldsymbol{\Sigma}^{1/2}\widehat{\boldsymbol{\beta}}_\lambda^{(2)}\right\|^2$, $\widehat{\kappa^2\alpha_*^2}(\delta/2,\kappa) = \left\langle\boldsymbol{\Sigma}^{1/2}\widehat{\boldsymbol{\beta}}_\lambda^{(1)}, \boldsymbol{\Sigma}^{1/2}\widehat{\boldsymbol{\beta}}_\lambda^{(2)}\right\rangle$. Since estimating $\boldsymbol{\Sigma}$ is challenging in moderate dimensions, we instead construct a fully computable alternative estimator. We begin with the following observation. Let $\boldsymbol{X} \sim N(0,\boldsymbol{\Sigma})$.

$$\|\boldsymbol{\Sigma}^{1/2}\widehat{\boldsymbol{\beta}}_\lambda^{(1)}\|^2 = \text{Var}(\boldsymbol{X}^\top\widehat{\boldsymbol{\beta}}_\lambda^{(1)} \mid \widehat{\boldsymbol{\beta}}_\lambda^{(1)}),$$

$$\|\boldsymbol{\Sigma}^{1/2}\widehat{\boldsymbol{\beta}}_\lambda^{(2)}\|^2 = \text{Var}(\boldsymbol{X}^\top\widehat{\boldsymbol{\beta}}_\lambda^{(2)} \mid \widehat{\boldsymbol{\beta}}_\lambda^{(2)}),$$

$$\widehat{\boldsymbol{\beta}}_\lambda^{(1)\top}\boldsymbol{\Sigma}\widehat{\boldsymbol{\beta}}_\lambda^{(2)} = \text{Cov}(\boldsymbol{X}^\top\widehat{\boldsymbol{\beta}}_\lambda^{(1)}, \boldsymbol{X}^\top\widehat{\boldsymbol{\beta}}_\lambda^{(2)} \mid \widehat{\boldsymbol{\beta}}_\lambda^{(1)\top}, \widehat{\boldsymbol{\beta}}_\lambda^{(2)}). \quad (12)$$

Based on data splitting, we use the covariates in $D_2$ to form the independent sequence $\{\boldsymbol{X}_i^\top\widehat{\boldsymbol{\beta}}_\lambda^{(2)}\}_{i=1}^{n/2}$. Consequently, sample variances can be used to approximate $\|\boldsymbol{\Sigma}^{1/2}\widehat{\boldsymbol{\beta}}_\lambda^{(2)}\|^2$.

We therefore define

$$S_1(\widehat{\boldsymbol{\beta}}_\lambda^{(1)}) := \frac{1}{n/2}\sum_{i\in D_2}\left[\boldsymbol{X}_i^\top\widehat{\boldsymbol{\beta}}_\lambda^{(1)} - \frac{1}{n/2}\sum_{i\in D_2}\boldsymbol{X}_i^\top\widehat{\boldsymbol{\beta}}_\lambda^{(1)}\right]^2.$$

$$S_2(\widehat{\boldsymbol{\beta}}_\lambda^{(2)}) := \frac{1}{n/2}\sum_{i\in D_1}\left[\boldsymbol{X}_i^\top\widehat{\boldsymbol{\beta}}_\lambda^{(2)} - \frac{1}{n/2}\sum_{i\in D_1}\boldsymbol{X}_i^\top\widehat{\boldsymbol{\beta}}_\lambda^{(2)}\right]^2.$$

The covariance term in (12) is more challenging to estimate, since each $\boldsymbol{X}_i$ exhibits nontrivial dependence with either $\widehat{\boldsymbol{\beta}}_\lambda^{(1)}$ or $\widehat{\boldsymbol{\beta}}_\lambda^{(2)}$. To address this issue, we introduce a leave-one-out construction that removes this dependence. Let $\widehat{\boldsymbol{\beta}}_{-i,\lambda}^{(1)}$ denote the solution of (3) based on the dataset $D_1\backslash\{i\}$ and define $\widehat{\boldsymbol{\beta}}_{-i,\lambda}^{(2)}$ analogously using $D_2\backslash\{i\}$. Since removing a single observation has a negligible effect on the estimator, we define

$$S_3(\widehat{\boldsymbol{\beta}}_\lambda^{(1)}, \widehat{\boldsymbol{\beta}}_\lambda^{(2)}) = \frac{1}{2}\left[\frac{1}{n/2}\sum_{i\in D_1}\left(\boldsymbol{X}_i^\top\widehat{\boldsymbol{\beta}}_{-i,\lambda}^{(1)}\right)\left(\boldsymbol{X}_i^\top\widehat{\boldsymbol{\beta}}_\lambda^{(2)}\right)\right.$$

$$\left. + \frac{1}{n/2}\sum_{i\in D_2}\left(\boldsymbol{X}_i^\top\widehat{\boldsymbol{\beta}}_\lambda^{(1)}\right)\left(\boldsymbol{X}_i^\top\widehat{\boldsymbol{\beta}}_{-i,\lambda}^{(2)}\right)\right].$$

To establish the consistency of the quantities $S_1$, $S_2$, and $S_3$, the following result holds.

**Theorem 5.2.** *Let $\lambda > 0$ be a fixed constant and $\delta > 2$. Assume there exist positive constants $c$ and $C$ such that $c < \lambda_{min}(\boldsymbol{\Sigma}) \leq \lambda_{max}(\boldsymbol{\Sigma}) < C$, and assume $\boldsymbol{\beta}_*^\top\boldsymbol{\Sigma}\boldsymbol{\beta}_* \to \kappa^2$, then we have*

1. $\frac{1}{2}[S_1(\widehat{\boldsymbol{\beta}}_\lambda^{(1)}) + S_2(\widehat{\boldsymbol{\beta}}_\lambda^{(2)})] \xrightarrow{\mathbb{P}} \kappa^2\alpha_*^2(\delta/2,\kappa) + \sigma_*^2(\delta/2,\kappa)$

2. *Further assume that $\max_{i\in D_1}\|\widehat{\boldsymbol{\beta}}_{-i,\lambda}^{(1)} - \widehat{\boldsymbol{\beta}}_\lambda^{(1)}\|_2 \xrightarrow{\mathbb{P}} 0$ and $\max_{i\in D_2}\|\widehat{\boldsymbol{\beta}}_{-i,\lambda}^{(2)} - \widehat{\boldsymbol{\beta}}_\lambda^{(2)}\|_2 \xrightarrow{\mathbb{P}} 0$, then we have*

$$S_3(\widehat{\boldsymbol{\beta}}_\lambda^{(1)}, \widehat{\boldsymbol{\beta}}_\lambda^{(2)}) \xrightarrow{\mathbb{P}} \kappa^2\alpha_*^2(\delta/2,\kappa)$$

*Remark* 5.3. For the first statement in Theorem 5.2, symmetry implies that $S_1(\widehat{\boldsymbol{\beta}}_\lambda^{(1)})$ and $S_2(\widehat{\boldsymbol{\beta}}_\lambda^{(2)})$ converge to the same limit, averaging them reduces variance. For the second statement, we impose a leave-one-out stability condition, which is mild and closely related to conditions established in El Karoui (2018); Sur et al. (2019); Yadlowsky et al. (2021). A sufficient condition for this stability requirement is discussed in Appendix E.3.

Theorem 5.2 provides a practical route to estimating both the variance factor and $\kappa^2\alpha_*^2(\delta/2,\kappa)$ without knowledge of the true covariance matrix $\boldsymbol{\Sigma}$. Specifically, we estimate $\kappa^2\alpha_*^2(\delta/2,\kappa)$ by $\widehat{\kappa^2\alpha_*^2}(\delta/2,\kappa) := S_3(\widehat{\boldsymbol{\beta}}_\lambda^{(1)}, \widehat{\boldsymbol{\beta}}_\lambda^{(2)})$ and estimate $\sigma_*^2(\delta/2,\kappa)$ by $\widetilde{\sigma_*^2}(\delta/2,\kappa) := \frac{1}{2}[S_1(\widehat{\boldsymbol{\beta}}_\lambda^{(1)}) + S_2(\widehat{\boldsymbol{\beta}}_\lambda^{(2)})] - S_3(\widehat{\boldsymbol{\beta}}_\lambda^{(1)}, \widehat{\boldsymbol{\beta}}_\lambda^{(2)})$.

---
**Algorithm 1** Signal strength estimation via data splitting

---

1: **Input:** Dataset $D = \{(X_i, Y_i)\}_{i=1}^n$; ratio $\delta = n/p$; regularization level $\lambda$.
2: **Step 1 (Theoretical map):** Solve the system of equations with varying $\kappa$ and fixed parameters $(\delta/2, \lambda)$. Denote the resulting map by $F_2(\kappa) = \kappa^2 \alpha_*^2(\delta/2, \kappa)$.
3: **Step 2 (Data splitting):** Randomly split $D$ into two independent subsets $D_1$ and $D_2$ of equal size.
4: **Step 3 (Proxy estimation):** Compute the empirical estimate of $\kappa^2 \alpha_*^2(\delta/2, \kappa)$ based on $D_1$ and $D_2$:

$$\widetilde{\kappa^2 \alpha_*^2}(\delta/2, \kappa) := S_3(\widehat{\boldsymbol{\beta}}_\lambda^{(1)}, \widehat{\boldsymbol{\beta}}_\lambda^{(2)}).$$

5: **Output:** $\widehat{\kappa}$, the root of $F_2(\kappa) - \widetilde{\kappa^2 \alpha_*^2}(\delta/2, \kappa) = 0$ using bisection method.

---

### 5.3. Estimate signal strength

Motivated by the monotone relationships established in theory (see Figure 2), we estimate the signal strength $\kappa$ by matching empirical proxies with their theoretical counterparts. Specifically, we define the estimator $\widehat{\kappa}$ as the solution to one of the following equations:

$$\widehat{\kappa} = \{\kappa : \widetilde{\sigma_*^2}(\delta/2, \kappa) = \sigma_*^2(\delta/2, \kappa)\} \tag{13}$$

$$\text{or } \widehat{\kappa} = \{\kappa : \widetilde{\kappa^2 \alpha_*^2}(\delta/2, \kappa) = \kappa^2 \alpha_*^2(\delta/2, \kappa)\}. \tag{14}$$

We summarize the estimation procedure based on the theoretical map $F_2$ in Algorithm 1. An analogous algorithm based on the theoretical map $F_1$ is provided in Appendix A.4. Note that the algorithm requires specifying the value of $\lambda$; in all experiments in this work we use $\lambda = 0.1$, and we conduct a sensitivity analysis in Appendix A.8 to show that the estimation of $\kappa$ is not sensitive to the value of $\lambda$.

We next justify the statistical performance of the estimator $\widehat{\kappa}$. The following identifiability assumption ensures that $\kappa$ can be uniquely recovered from the theoretical maps.

**Assumption 5.4** ($\kappa$ Identifiability). Fix $\delta > 2$ and $\lambda > 0$. Both $F_1$ and $F_2$ are continuously differentiable and strictly monotone on an interval $\mathcal{K} \subset \mathbb{R}$. There exists a constant $c_0 > 0$ such that for the true signal strength $\kappa \in \text{int}(\mathcal{K})$,

$$|F_1'(\kappa)| > c_0 \quad \text{and} \quad |F_2'(\kappa)| > c_0.$$

**Theorem 5.5** (Consistent estimation of $\kappa$). *Under Assumption 5.4 and the conditions of Theorem 5.2, the estimators defined in (13) and (14) are consistent estimators of $\kappa$.*

*Remark* 5.6. The condition $|F_k'(\kappa)| > c_0$ guarantees local identifiability of $\kappa$ through the proxy $F_k(\kappa)$. This assumption is mild: as illustrated in Figure 2, $F_k(\kappa)$ is strictly monotone on the interval $(0, 5)$ with slope bounded away from zero. In particular, when the data are separable (which typically occurs for small $\delta$), both $F_1$ and $F_2$ are highly informative and can accurately identify the true $\kappa$.

## 6. Numerical studies

We focus on settings in which the observed data are mostly linearly separable. Additional experiments are provided in Appendix A.4. Specifically, we fix the aspect ratio $\delta = n/p = 3$ and consider two signal strengths, $\|\boldsymbol{\beta}_*\|_2 \in \{2, 3\}$. The entries of $\boldsymbol{\beta}_*$ are generated independently from the set $\{-\|\boldsymbol{\beta}_*\|_2/\sqrt{p}, \|\boldsymbol{\beta}_*\|_2/\sqrt{p}\}$. We fix $\lambda = 0.1$ in all experiments. For each setting, results are averaged over 100 independent replications. In Appendix A.1, we show that a large proportion of the generated datasets are linearly separable in our experiments.

We consider two covariance structures for the feature vectors: (i) the identity covariance $\boldsymbol{\Sigma} = \mathbf{I}_p$, and (ii) a general covariance matrix $\boldsymbol{\Sigma}$, which is randomly generated as described in Appendix A.4.

To compute the statistic $S_3(\widehat{\boldsymbol{\beta}}_\lambda^{(1)}, \widehat{\boldsymbol{\beta}}_\lambda^{(2)})$ efficiently, we employ a standard leave-one-out (LOO) approximation for $\mathbf{X}_i^\top \widehat{\boldsymbol{\beta}}_{-i,\lambda}^{(1)}$ and $\mathbf{X}_i^\top \widehat{\boldsymbol{\beta}}_{-i,\lambda}^{(2)}$, thereby avoiding repeated model fitting. Details of this approximation are provided in Appendix E.1.

After obtaining the estimate $\widehat{\kappa}$, we plug $(\widehat{\kappa}, \delta, \lambda)$ into the system of equations (5) and solve for $(\widehat{\alpha}_*, \widehat{\sigma}_*, \widehat{\gamma}_*)$. We assess performance using the absolute errors $|\kappa - \widehat{\kappa}|$, $|\alpha_* - \widehat{\alpha}_*|$, and $|\sigma_* - \widehat{\sigma}_*|$. Based on the estimated $\widehat{\alpha}_*$ and $\widehat{\sigma}_*$, we construct 95% confidence intervals using Proposition 3.3 with $\Psi(a, 1.96) = \mathbf{1}\{|a/\sigma_*| \leq 1.96\}$:

$$\widehat{\text{CI}}_j = \left[ \frac{\widehat{\boldsymbol{\beta}}_{\lambda,j} - 1.96\widehat{\sigma}_*/(v_j\sqrt{p})}{\widehat{\alpha}_*}, \frac{\widehat{\boldsymbol{\beta}}_{\lambda,j} + 1.96\widehat{\sigma}_*/(v_j\sqrt{p})}{\widehat{\alpha}_*} \right]. \tag{15}$$

For $\boldsymbol{\Sigma} = \mathbf{I}_p$, we take $v_j = 1$. When the true covariance matrix is unknown, we use $\widehat{v}_j = 1/[\widehat{\Sigma}^{-1}]_{jj}$, following Zhao et al. (2022), where $\widehat{\Sigma}$ denotes the sample covariance matrix.

The experimental results are shown in Figure 3. As the sample size increases, the estimation errors for $\kappa$, $\alpha_*$, and $\sigma_*$ decrease, which is consistent with our theoretical predictions. For the coverage rate of the confidence intervals, the empirical coverage under identity covariance is close to the nominal level of 0.95. Under general covariance, the coverage is slightly inflated due to inaccuracies in estimating $v_j$. When the true $v_j$ is used instead of its estimate, the coverage rate is approximately 0.95 (see Table 2 in Appendix A.4).

## 7. Real Data Analysis

We demonstrate our method on a single-cell RNA sequencing (scRNA-seq) dataset from Hoffman et al. (2020), with the goal of identifying genes associated with the glucocorticoid response (GR).

The dataset comprises $n = 2400$ gene expression profiles: 2000 from glucocorticoid-treated cells and 400 from vehicle-

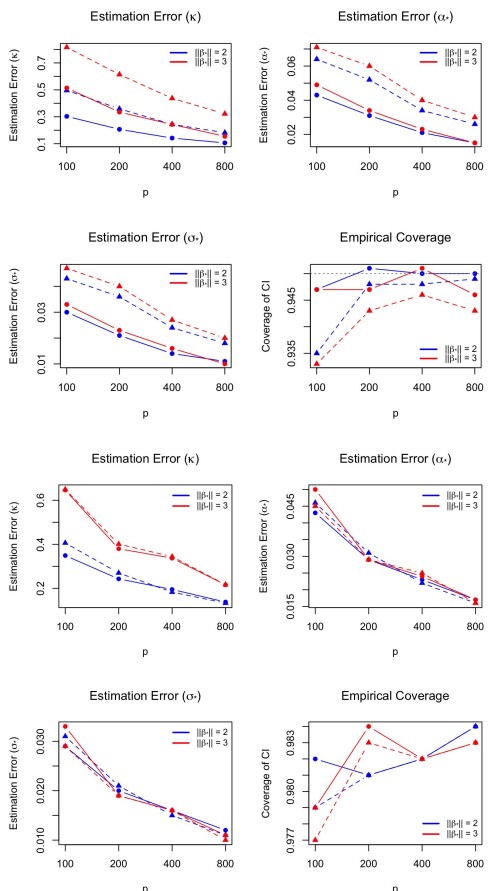

*Figure 3.* The top four panels correspond to the identity covariance case $\mathbf{\Sigma} = \mathbf{I}_p$, while the bottom four panels correspond to the general covariance case. Dashed lines indicate estimates of $\kappa$ obtained using Algorithm 1, whereas solid lines indicate estimates obtained using Algorithm 2.

treated controls. The binary response indicates treatment status (glucocorticoid vs. control). Following the preprocessing pipeline of Dai et al. (2023b), we filter out low-expression genes, center the data, and select the top $p = 600$ most variable genes for downstream analysis. We then standardize each gene to have unit variance.

We apply Algorithm 1 with regularization parameter $\lambda = 0.1$, which yields an estimated aspect ratio correction $\hat{\kappa} = 2.22$. Given $(\hat{\kappa}, \delta) = (2.22, 4)$ with $\delta = n/p = 2400/600$, we construct adjusted confidence intervals using (15) for each coefficient and select genes whose intervals exclude zero.

Our method identifies 23 significant genes in total. Of these, 13 were also reported by Dai et al. (2023b) and are supported by existing literature on glucocorticoid response:

1. **Shared with Dai et al. (2023b) (13 genes):** SER-PINA6, FKBP5, NFKBIA, RPL10, HSPB1, EEF1A1, HSPA1A, BCL6, S100A11, NUPR1, LY6E, DDIT4, and TACSTD2.

2. **Unique to our method (10 genes):** HSPB8, ZNF703, GPSM2, DHCR24, ANXA2, ADIRF, YWHAE, CD9, MARS, and CTSD.

The substantial overlap with prior findings validates our approach, while the novel discoveries may warrant further biological investigation.

## 8. Discussion

We propose an efficient and theoretically sound data-splitting approach for estimating signal strength in moderate-dimensional logistic models, including regimes in which the data are separable. By introducing a non-decomposable generalized ridge regularization weighted by the sample covariance matrix, we establish that the estimator exists with high probability and characterize its asymptotic properties in terms of bias and variance factors. We prove that our data-splitting method yields a consistent estimator of signal strength, and we verify this result through numerical experiments.

Our theoretical analysis is conducted under a Gaussian design assumption. To assess robustness beyond this setting, we numerically investigate universality in Appendix A.3. The results suggest that the proposed methodology is largely insensitive to departures from Gaussianity; however, we leave the development of a rigorous universality theory to future work. Figure 2 reveals that the slope of the mapping between the desired proxy and the signal strength vanishes at both extremes of the signal regime. This indicates that estimating signal strength becomes increasingly difficult near the boundaries, a phenomenon for which we currently lack a theoretical explanation and whose analysis we also defer to future study.

## Acknowledgements

The authors thank the anonymous referees for their insightful and constructive comments, which have led to a significantly improved version of the paper. This work was supported by the Talent Introduction Research Start-up Fund of Tsinghua University under grant number 53330000426.

## Impact Statement

This paper proposes a theoretically grounded estimator for signal strength in logistic regression models. Compared with standard approaches that either ignore dimensionality effects or rely on restrictive asymptotic regimes, our

method improves the reliability of inference when the dimension scales with the sample size, benefiting fields such as high-dimensional statistics and functional estimation of regression coefficients. As a methodological contribution intended for settings where statistical modeling and inference are already routine, we do not expect the proposed approach itself to introduce additional negative societal impacts beyond those associated with logistic regression and high dimensional estimation in general. Our work primarily engages in theoretical analysis and does not directly interact with or manipulate real-world systems. Consequently, it is unlikely to have negative societal consequences.

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

**Contents of Appendix**

# A. Additional numerical experiments

## A.1. Separability issue when $n/p$ is small

In this section, we numerically examine some small-$n$ settings used in our numerical experiments in Section 6 and Section A.4. For each setting, we generate 100 independent datasets and compute the proportion for which the data are separable (i.e., the MLE does not exist).

To generate the covariance matrix $\boldsymbol{\Sigma}$ for each dimension $p$, we proceed as follows: first, we sample a random orthogonal matrix $Q$ uniformly from the Haar distribution on the orthogonal group $\mathcal{O}(p)$; second, we draw eigenvalues $\lambda_1, \ldots, \lambda_p \overset{\text{i.i.d.}}{\sim}$ Uniform$(1, 2)$ and form the true covariance matrix as $\boldsymbol{\Sigma} = Q \operatorname{diag}(\lambda_1, \ldots, \lambda_p) Q^\top$. The covariance matrix is kept fixed for all replications.

The results are shown in Table 1. We find that when $n/p$ is small and the signal strength $\|\boldsymbol{\beta}_*\|_2$ is large, the data are more likely to be separable.

*Table 1.* Proportion of separable case under identity and general covariance structures.

| $\delta$ | $p$ | $\|\boldsymbol{\beta}_*\|_2$ | Identity Covariance | General Covariance |
|---|---|---|---|---|
| 3 | 100 | 2 | 0.38 | 0.79 |
| 3 | 100 | 3 | 0.95 | 1.00 |
| 3 | 100 | 4 | 1.00 | 1.00 |
| 3 | 200 | 2 | 0.25 | 0.85 |
| 3 | 200 | 3 | 0.98 | 1.00 |
| 3 | 200 | 4 | 1.00 | 1.00 |
| 3 | 400 | 2 | 0.28 | 0.91 |
| 3 | 400 | 3 | 1.00 | 1.00 |
| 3 | 400 | 4 | 1.00 | 1.00 |
| 3 | 800 | 2 | 0.13 | 0.98 |
| 3 | 800 | 3 | 1.00 | 1.00 |
| 3 | 800 | 4 | 1.00 | 1.00 |
| 4 | 100 | 2 | 0.00 | 0.02 |
| 4 | 100 | 3 | 0.24 | 0.55 |
| 4 | 100 | 4 | 0.84 | 0.98 |
| 4 | 200 | 2 | 0.00 | 0.01 |
| 4 | 200 | 3 | 0.10 | 0.76 |
| 4 | 200 | 4 | 0.91 | 1.00 |
| 4 | 400 | 2 | 0.00 | 0.00 |
| 4 | 400 | 3 | 0.09 | 0.87 |
| 4 | 400 | 4 | 0.99 | 1.00 |
| 4 | 800 | 2 | 0.00 | 0.00 |
| 4 | 800 | 3 | 0.00 | 0.88 |
| 4 | 800 | 4 | 1.00 | 1.00 |

## A.2. Numerical verification for Theorem 3.1

In Theorem 3.1, under the assumption that $\mathrm{Cov}(X_i) = \mathbf{I}_p$, we have the following asymptotic characterizations:

$$\text{bias scalar: } \frac{\boldsymbol{\beta}_*^\top \widehat{\boldsymbol{\beta}}_\lambda}{\|\boldsymbol{\beta}_*\|^2} \to \alpha_*,$$
$$\text{squared error: } \|\widehat{\boldsymbol{\beta}}_\lambda - \boldsymbol{\beta}_*\|^2 \to (\alpha_* - 1)^2 \kappa_1^2 + \sigma_*^2.$$

We empirically verify these predictions under the following setting: $p = 500$, $n = \delta \cdot p$, and the entries of $\boldsymbol{\beta}_*$ are i.i.d. draws from $N(0, 1/p)$. For each setting, we average the results over 100 independent data-generating replications.

The results are shown in Figure 4. The empirical bias scalars and squared errors closely match their respective theoretical limits across different values of the tuning parameter $\lambda$, providing numerical support for Theorem 3.1.

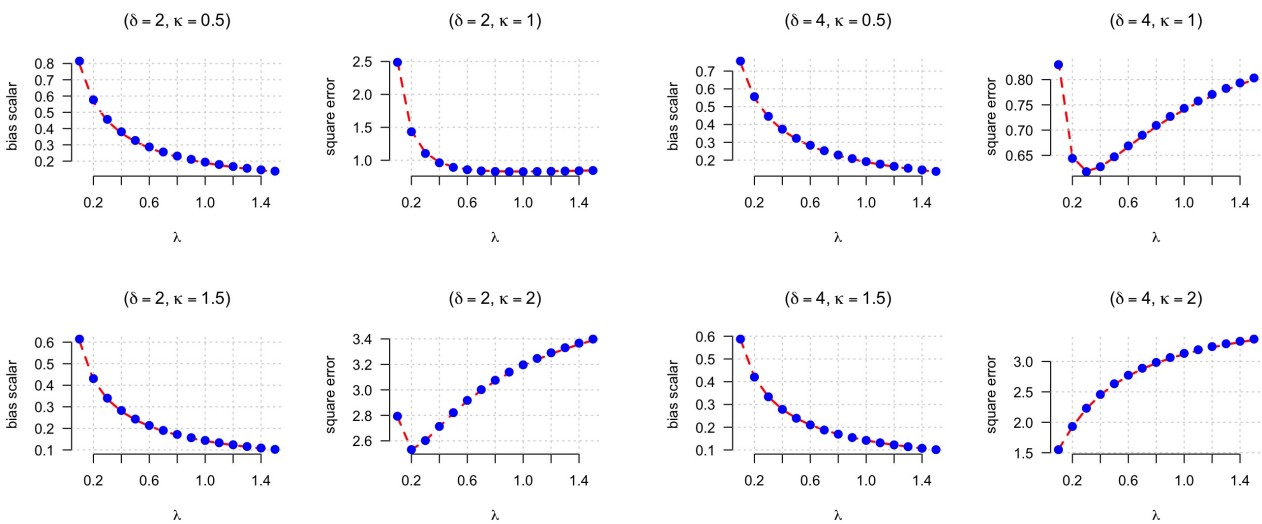

*Figure 4.* Points denote the empirical bias scalar and squared error as functions of the tuning parameter $\lambda$; lines represent the corresponding theoretical limits.

### A.3. Universality experiment for design matrix

In this section, we present several numerical experiments to empirically justify that the Gaussian design assumption can be relaxed.

In the following experiments, the entries of both the observed and synthetic covariate matrices are i.i.d. samples from a Student's $t$-distribution with various degrees of freedom ($df = 3, 4, 5, 6$). The entries are scaled to have mean zero and unit variance, matching the first two moments of the standard Gaussian distribution. The true coefficient vector $\boldsymbol{\beta}_*$ has dimension $p = 500$, with entries drawn independently from $N(0, 1/p)$.

We compare the averaged empirical squared error $\|\widehat{\boldsymbol{\beta}}_\lambda - \boldsymbol{\beta}_*\|^2$ with its theoretical limit $(\alpha_* - 1)^2 \kappa_1^2 + \sigma_*^2$. In Figures 5 , we plot the empirical values (average over 100 independent data-generating replications.) as points and the corresponding theoretical predictions as curves.

We observe that when the degrees of freedom are below 4 (e.g., $df = 2.5$ and 3), the alignment between empirical and theoretical values is bad (see bottom-left panels). However, when the degrees of freedom are 4 or greater, the empirical results align closely with the theoretical predictions. This suggests that our theory on generalized ridge estimator can potentially extend beyond Gaussian designs under some finite fourth-moment condition.

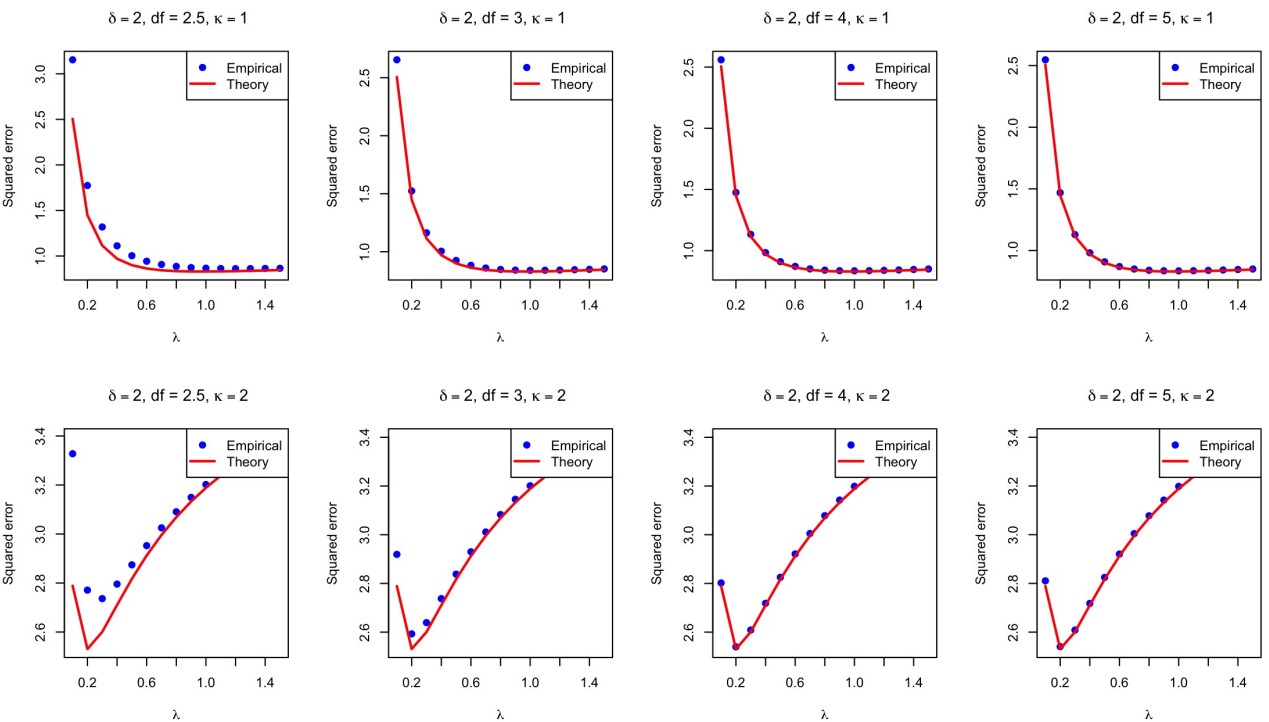

*Figure 5.* From left to right, the degrees of freedom of the $t$-distribution increase.

### A.4. Additional numerical experiment for data splitting

To further illustrate the performance of signal strength estimation via data splitting, we conduct additional experiments across different values of $p$, $\delta = n/p$, and $\|\boldsymbol{\beta}_*\|_2$. For each dimension $p$, the entries of $\boldsymbol{\beta}_*$ are generated independently from $\{-\|\boldsymbol{\beta}_*\|_2/\sqrt{p}, \|\boldsymbol{\beta}_*\|_2/\sqrt{p}\}$. For each setting, we average results over 100 independent replications. For each replication, we average results over 100 random data splitting.

We compare two estimators: one based on Algorithm 1 and the other on Algorithm 2. To distinguish them, we refer to the former as the "$\kappa^2\alpha_*^2$-based" estimator and the latter as the "$\sigma_*^2$-based" estimator. After obtaining the estimate $\widehat{\kappa}$, we plug $(\widehat{\kappa}, \delta, \lambda)$ into the system of equations (5) and solve for $(\widehat{\alpha}_*, \widehat{\sigma}_*, \widehat{\gamma}_*)$. We assess their performance using the following

---

**Algorithm 2** Signal strength estimation via data splitting

---

1: **Input:** Dataset $D = \{(X_i, Y_i)\}_{i=1}^n$; ratio $\delta = n/p$, regularization level $\lambda$.
2: **Step 1 (Theoretical map):** Solve system of equation with varying $\kappa$ and fixed parameter $(\delta/2, \lambda)$. Denote the relationship by $F_1(\kappa) = \sigma_*^2$.
3: **Step 2 (Data splitting):** Randomly split $D$ into two independent subsets $D_1$ and $D_2$ of equal size.
4: **Step 3 (Estimation of proxy):** Compute the empirical estimate of $\sigma_*^2$ based on $D_1$ and $D_2$

$$\widetilde{\sigma_*^2}(\delta/2, \kappa) = \frac{1}{2}[S_1(\widehat{\boldsymbol{\beta}}_\lambda^{(1)}) + S_2(\widehat{\boldsymbol{\beta}}_\lambda^{(2)})] - S_3(\widehat{\boldsymbol{\beta}}_\lambda^{(1)}, \widehat{\boldsymbol{\beta}}_\lambda^{(2)}).$$

5: **Output:** $\widehat{\kappa}$, the root of $F_1(\kappa) - \widetilde{\sigma_*^2}(\delta/2, \kappa) = 0$ using bisection method.

---

metrics:

$$|\kappa - \widehat{\kappa}|, \quad |\alpha_* - \widehat{\alpha}_*|, \quad \text{and} \quad |\sigma_* - \widehat{\sigma}_*|.$$

Based on the estimated $\widehat{\alpha}_*$ and $\widehat{\sigma}_*$, we construct the 95% confidence interval:

$$\widehat{\text{CI}}_j = \left[ \frac{\widehat{\boldsymbol{\beta}}_{\lambda,j} - 1.96\widehat{\sigma}_*/(v_j\sqrt{p})}{\widehat{\alpha}_*}, \frac{\widehat{\boldsymbol{\beta}}_{\lambda,j} + 1.96\widehat{\sigma}_*/(v_j\sqrt{p})}{\widehat{\alpha}_*} \right], \quad j \in [p].$$

For $\boldsymbol{\Sigma} = \mathbf{I}_p$, we take $v_j = 1$. When the true covariance matrix is unknown, we use $\widehat{v}_j = \frac{1}{[\widehat{\Sigma}^{-1}]_{jj}}$, following (Zhao et al., 2022), where $\widehat{\Sigma}$ is the sample covariance matrix.

In the case $\boldsymbol{\Sigma} = \mathbf{I}_p$, the signal strength is $\kappa = \|\boldsymbol{\beta}_*\|_2$, and the corresponding results are presented in Table 3.

For general $\boldsymbol{\Sigma}$, we generate the covariance matrix as follows: first, we sample a random orthogonal matrix $Q$ uniformly from the Haar distribution on the orthogonal group $\mathcal{O}(p)$; second, we draw eigenvalues $\lambda_1, \ldots, \lambda_p \overset{\text{i.i.d.}}{\sim} \text{Uniform}(1, 2)$ and form the true covariance matrix as $\boldsymbol{\Sigma} = Q \operatorname{diag}(\lambda_1, \ldots, \lambda_p) Q^\top$. Due to this randomness, the signal strength $\kappa = \operatorname{Var}(\boldsymbol{X}^\top \boldsymbol{\beta}_*) = \boldsymbol{\beta}_*^\top \boldsymbol{\Sigma} \boldsymbol{\beta}_*$ is not identical across different dimensions $p$, but remains similar. The results are presented in Tables 4 and 5.

We observe that in the general covariance setting, the coverage rate for the 95% confidence interval is inflated. This inflation arises from the additional estimation error induced when estimating the conditional variance $v_j$. To verify this hypothesis, we replace the estimated $\widehat{v}_j$ with the true $v_j$ and present the results in Table 2. As shown in the table, using the true $v_j$ yields coverage rates close to the nominal 95% level. Moreover, for larger sample sizes (e.g., $\delta = 6$ or $\delta = 8$), the coverage rate approaches 0.95 even with estimated $\widehat{v}_j$, which is partly due to the improved accuracy of $\widehat{v}_j$ as more samples become available.

*Table 2.* Empirical coverage of nominal 95% confidence intervals, grouped by $(\delta, \kappa)$ and showing results across $p = 100, 200, 400, 800$. Each cell lists coverage in the order ($p = 100, p = 200, p = 400, p = 800$).

| $\delta$ | $\kappa$ | $\sigma_*$-based | $\kappa^2\alpha_*^2$-based | Oracle $\sigma_*$-based | Oracle $\kappa^2\alpha_*^2$-based |
|---|---|---|---|---|---|
| 3 | 1.226 | (0.983, 0.983, 0.983, 0.984) | (0.982, 0.983, 0.983, 0.984) | (0.950, 0.946, 0.948, 0.950) | (0.948, 0.945, 0.948, 0.950) |
| | 2.434 | (0.982, 0.981, 0.982, 0.984) | (0.979, 0.981, 0.982, 0.984) | (0.944, 0.946, 0.948, 0.949) | (0.940, 0.944, 0.948, 0.949) |
| | 3.590 | (0.979, 0.984, 0.982, 0.983) | (0.977, 0.983, 0.982, 0.983) | (0.941, 0.948, 0.946, 0.949) | (0.940, 0.947, 0.946, 0.948) |
| | 4.746 | (0.976, 0.979, 0.982, 0.984) | (0.974, 0.976, 0.981, 0.983) | (0.938, 0.941, 0.947, 0.949) | (0.931, 0.936, 0.945, 0.948) |
| 4 | 1.226 | (0.972, 0.977, 0.977, 0.975) | (0.973, 0.976, 0.977, 0.975) | (0.944, 0.949, 0.949, 0.948) | (0.944, 0.949, 0.949, 0.948) |
| | 2.434 | (0.975, 0.976, 0.975, 0.977) | (0.975, 0.975, 0.975, 0.977) | (0.946, 0.947, 0.949, 0.951) | (0.944, 0.947, 0.948, 0.951) |
| | 3.590 | (0.972, 0.973, 0.977, 0.975) | (0.970, 0.972, 0.977, 0.975) | (0.942, 0.944, 0.950, 0.949) | (0.940, 0.943, 0.950, 0.949) |
| | 4.746 | (0.974, 0.976, 0.976, 0.976) | (0.974, 0.977, 0.977, 0.976) | (0.945, 0.949, 0.949, 0.950) | (0.946, 0.949, 0.950, 0.949) |
| 6 | 1.226 | (0.973, 0.970, 0.969, 0.968) | (0.972, 0.970, 0.969, 0.968) | (0.955, 0.952, 0.951, 0.950) | (0.954, 0.952, 0.951, 0.950) |
| | 2.434 | (0.964, 0.966, 0.968, 0.968) | (0.964, 0.965, 0.968, 0.968) | (0.943, 0.947, 0.948, 0.949) | (0.944, 0.946, 0.948, 0.949) |
| | 3.590 | (0.963, 0.963, 0.966, 0.968) | (0.963, 0.964, 0.966, 0.968) | (0.941, 0.943, 0.946, 0.950) | (0.941, 0.944, 0.947, 0.950) |
| | 4.746 | (0.958, 0.960, 0.960, 0.964) | (0.956, 0.961, 0.966, 0.965) | (0.939, 0.939, 0.939, 0.944) | (0.938, 0.941, 0.947, 0.947) |
| 8 | 1.226 | (0.963, 0.963, 0.967, 0.964) | (0.964, 0.963, 0.967, 0.964) | (0.949, 0.950, 0.953, 0.950) | (0.949, 0.950, 0.953, 0.949) |
| | 2.434 | (0.961, 0.961, 0.966, 0.963) | (0.960, 0.961, 0.967, 0.964) | (0.947, 0.948, 0.952, 0.950) | (0.948, 0.947, 0.952, 0.950) |
| | 3.590 | (0.958, 0.958, 0.961, 0.961) | (0.960, 0.959, 0.962, 0.962) | (0.941, 0.943, 0.947, 0.947) | (0.943, 0.944, 0.947, 0.948) |
| | 4.746 | (0.948, 0.959, 0.954, 0.959) | (0.948, 0.959, 0.960, 0.960) | (0.931, 0.944, 0.939, 0.943) | (0.931, 0.945, 0.945, 0.946) |

*Table 3.* Estimation error and coverage across ratio ($\delta$), signal strengths ($\kappa$), and dimensions ($p$). Values are mean (standard error). Left: $\sigma_*$-based; Right: $\kappa^2\alpha_*^2$-based estimation.

| $\delta$ | $\kappa$ | $p$ | $\sigma_*$-based | | | | $\kappa^2\alpha_*^2$-based | | | |
|---|---|---|---|---|---|---|---|---|---|---|
| | | | $\|\kappa - \hat{\kappa}\|$ | Coverage | $\|\alpha_* - \hat{\alpha}\|$ | $\|\sigma_* - \hat{\sigma}\|$ | $\|\kappa - \hat{\kappa}\|$ | Coverage | $\|\alpha_* - \hat{\alpha}\|$ | $\|\sigma_* - \hat{\sigma}\|$ |
| | | | | | | $\delta = 3$ | | | | |
| 3 | 2 | 100 | 0.303 (0.027) | 0.947 (0.003) | 0.043 (0.004) | 0.030 (0.002) | 0.496 (0.056) | 0.935 (0.005) | 0.064 (0.006) | 0.043 (0.004) |
| | | 200 | 0.207 (0.015) | 0.951 (0.002) | 0.031 (0.002) | 0.021 (0.002) | 0.359 (0.030) | 0.948 (0.003) | 0.052 (0.004) | 0.036 (0.003) |
| | | 400 | 0.142 (0.010) | 0.950 (0.001) | 0.021 (0.001) | 0.014 (0.001) | 0.244 (0.018) | 0.948 (0.002) | 0.034 (0.002) | 0.024 (0.002) |
| | | 800 | 0.106 (0.008) | 0.950 (0.001) | 0.015 (0.001) | 0.011 (0.001) | 0.182 (0.014) | 0.949 (0.001) | 0.026 (0.002) | 0.018 (0.001) |
| 3 | 3 | 100 | 0.514 (0.042) | 0.947 (0.003) | 0.049 (0.004) | 0.033 (0.002) | 0.816 (0.064) | 0.933 (0.004) | 0.071 (0.005) | 0.047 (0.003) |
| | | 200 | 0.335 (0.025) | 0.947 (0.002) | 0.034 (0.003) | 0.023 (0.002) | 0.614 (0.041) | 0.943 (0.003) | 0.060 (0.004) | 0.040 (0.002) |
| | | 400 | 0.243 (0.018) | 0.951 (0.001) | 0.023 (0.002) | 0.016 (0.001) | 0.437 (0.038) | 0.946 (0.002) | 0.040 (0.003) | 0.027 (0.002) |
| | | 800 | 0.155 (0.012) | 0.946 (0.001) | 0.015 (0.001) | 0.010 (0.001) | 0.322 (0.031) | 0.943 (0.002) | 0.030 (0.003) | 0.020 (0.002) |
| 3 | 4 | 100 | 0.678 (0.038) | 0.944 (0.003) | 0.046 (0.003) | 0.029 (0.002) | 0.848 (0.044) | 0.935 (0.003) | 0.057 (0.004) | 0.036 (0.003) |
| | | 200 | 0.513 (0.034) | 0.950 (0.002) | 0.035 (0.003) | 0.022 (0.002) | 0.811 (0.042) | 0.942 (0.002) | 0.059 (0.004) | 0.037 (0.003) |
| | | 400 | 0.369 (0.027) | 0.949 (0.002) | 0.024 (0.002) | 0.015 (0.001) | 0.597 (0.037) | 0.944 (0.002) | 0.039 (0.003) | 0.025 (0.002) |
| | | 800 | 0.244 (0.018) | 0.947 (0.001) | 0.016 (0.001) | 0.010 (0.001) | 0.507 (0.034) | 0.943 (0.001) | 0.033 (0.002) | 0.020 (0.001) |
| | | | | | | $\delta = 4$ | | | | |
| 4 | 2 | 100 | 0.264 (0.022) | 0.948 (0.003) | 0.037 (0.003) | 0.020 (0.002) | 0.370 (0.043) | 0.938 (0.004) | 0.047 (0.004) | 0.026 (0.002) |
| | | 200 | 0.163 (0.013) | 0.948 (0.002) | 0.023 (0.002) | 0.012 (0.001) | 0.269 (0.023) | 0.944 (0.002) | 0.036 (0.003) | 0.020 (0.002) |
| | | 400 | 0.130 (0.010) | 0.948 (0.001) | 0.018 (0.001) | 0.010 (0.001) | 0.186 (0.014) | 0.946 (0.002) | 0.025 (0.002) | 0.014 (0.001) |
| | | 800 | 0.102 (0.007) | 0.950 (0.001) | 0.014 (0.001) | 0.008 (0.001) | 0.127 (0.010) | 0.950 (0.001) | 0.018 (0.001) | 0.010 (0.001) |
| 4 | 3 | 100 | 0.370 (0.034) | 0.946 (0.003) | 0.035 (0.003) | 0.019 (0.002) | 0.625 (0.052) | 0.932 (0.005) | 0.055 (0.004) | 0.029 (0.002) |
| | | 200 | 0.259 (0.017) | 0.945 (0.002) | 0.025 (0.002) | 0.013 (0.001) | 0.450 (0.043) | 0.937 (0.003) | 0.039 (0.003) | 0.021 (0.002) |
| | | 400 | 0.198 (0.015) | 0.946 (0.001) | 0.018 (0.001) | 0.010 (0.001) | 0.296 (0.026) | 0.943 (0.002) | 0.027 (0.002) | 0.014 (0.001) |
| | | 800 | 0.134 (0.009) | 0.951 (0.001) | 0.013 (0.001) | 0.007 (0.000) | 0.209 (0.016) | 0.949 (0.001) | 0.020 (0.001) | 0.011 (0.001) |
| 4 | 4 | 100 | 0.518 (0.034) | 0.945 (0.003) | 0.034 (0.002) | 0.017 (0.001) | 0.706 (0.041) | 0.935 (0.004) | 0.047 (0.003) | 0.024 (0.002) |
| | | 200 | 0.384 (0.027) | 0.944 (0.002) | 0.024 (0.002) | 0.012 (0.001) | 0.606 (0.034) | 0.936 (0.002) | 0.038 (0.002) | 0.019 (0.001) |
| | | 400 | 0.277 (0.022) | 0.946 (0.001) | 0.017 (0.001) | 0.009 (0.001) | 0.492 (0.031) | 0.940 (0.002) | 0.031 (0.002) | 0.015 (0.001) |
| | | 800 | 0.228 (0.018) | 0.948 (0.001) | 0.015 (0.001) | 0.007 (0.001) | 0.433 (0.028) | 0.944 (0.002) | 0.027 (0.002) | 0.014 (0.001) |
| | | | | | | $\delta = 6$ | | | | |
| 6 | 2 | 100 | 0.205 (0.016) | 0.947 (0.003) | 0.027 (0.002) | 0.011 (0.001) | 0.246 (0.019) | 0.943 (0.003) | 0.032 (0.002) | 0.013 (0.001) |
| | | 200 | 0.127 (0.009) | 0.949 (0.002) | 0.017 (0.001) | 0.007 (0.001) | 0.143 (0.013) | 0.947 (0.002) | 0.019 (0.002) | 0.008 (0.001) |
| | | 400 | 0.088 (0.006) | 0.948 (0.001) | 0.012 (0.001) | 0.005 (0.000) | 0.094 (0.007) | 0.947 (0.001) | 0.012 (0.001) | 0.005 (0.000) |
| | | 800 | 0.072 (0.006) | 0.950 (0.001) | 0.009 (0.001) | 0.004 (0.000) | 0.083 (0.006) | 0.950 (0.001) | 0.011 (0.001) | 0.004 (0.000) |
| 6 | 3 | 100 | 0.315 (0.024) | 0.946 (0.003) | 0.029 (0.002) | 0.012 (0.001) | 0.466 (0.032) | 0.936 (0.004) | 0.042 (0.003) | 0.017 (0.001) |
| | | 200 | 0.206 (0.015) | 0.944 (0.002) | 0.019 (0.001) | 0.008 (0.001) | 0.283 (0.028) | 0.940 (0.003) | 0.025 (0.002) | 0.010 (0.001) |
| | | 400 | 0.115 (0.009) | 0.948 (0.001) | 0.011 (0.001) | 0.004 (0.000) | 0.160 (0.013) | 0.946 (0.001) | 0.015 (0.001) | 0.006 (0.000) |
| | | 800 | 0.102 (0.009) | 0.947 (0.001) | 0.009 (0.001) | 0.004 (0.000) | 0.151 (0.011) | 0.946 (0.001) | 0.014 (0.001) | 0.006 (0.000) |
| 6 | 4 | 100 | 0.452 (0.031) | 0.938 (0.003) | 0.028 (0.002) | 0.011 (0.001) | 0.644 (0.035) | 0.927 (0.003) | 0.041 (0.002) | 0.016 (0.001) |
| | | 200 | 0.307 (0.023) | 0.945 (0.002) | 0.019 (0.001) | 0.007 (0.001) | 0.450 (0.034) | 0.936 (0.003) | 0.028 (0.002) | 0.011 (0.001) |
| | | 400 | 0.205 (0.016) | 0.947 (0.001) | 0.013 (0.001) | 0.005 (0.000) | 0.325 (0.026) | 0.942 (0.002) | 0.020 (0.002) | 0.008 (0.001) |
| | | 800 | 0.157 (0.012) | 0.948 (0.001) | 0.010 (0.001) | 0.004 (0.000) | 0.275 (0.021) | 0.945 (0.001) | 0.017 (0.001) | 0.007 (0.000) |
| | | | | | | $\delta = 8$ | | | | |
| 8 | 2 | 100 | 0.178 (0.014) | 0.949 (0.002) | 0.023 (0.002) | 0.008 (0.001) | 0.169 (0.014) | 0.947 (0.003) | 0.022 (0.002) | 0.008 (0.001) |
| | | 200 | 0.107 (0.009) | 0.949 (0.002) | 0.014 (0.001) | 0.005 (0.000) | 0.105 (0.007) | 0.948 (0.002) | 0.014 (0.001) | 0.005 (0.000) |
| | | 400 | 0.080 (0.005) | 0.950 (0.001) | 0.011 (0.001) | 0.004 (0.000) | 0.072 (0.005) | 0.949 (0.001) | 0.009 (0.001) | 0.003 (0.000) |
| | | 800 | 0.065 (0.004) | 0.950 (0.001) | 0.009 (0.001) | 0.003 (0.000) | 0.057 (0.004) | 0.950 (0.001) | 0.008 (0.001) | 0.003 (0.000) |
| 8 | 3 | 100 | 0.254 (0.018) | 0.943 (0.003) | 0.023 (0.002) | 0.008 (0.001) | 0.339 (0.029) | 0.937 (0.004) | 0.029 (0.002) | 0.010 (0.001) |
| | | 200 | 0.175 (0.014) | 0.944 (0.002) | 0.016 (0.001) | 0.005 (0.000) | 0.186 (0.015) | 0.944 (0.002) | 0.017 (0.001) | 0.006 (0.000) |
| | | 400 | 0.125 (0.009) | 0.949 (0.001) | 0.011 (0.001) | 0.004 (0.000) | 0.153 (0.010) | 0.947 (0.001) | 0.014 (0.001) | 0.005 (0.000) |
| | | 800 | 0.093 (0.007) | 0.950 (0.001) | 0.008 (0.001) | 0.003 (0.000) | 0.109 (0.008) | 0.949 (0.001) | 0.010 (0.001) | 0.003 (0.000) |
| 8 | 4 | 100 | 0.352 (0.027) | 0.944 (0.003) | 0.022 (0.002) | 0.007 (0.001) | 0.512 (0.033) | 0.928 (0.004) | 0.032 (0.002) | 0.010 (0.001) |
| | | 200 | 0.259 (0.021) | 0.943 (0.002) | 0.016 (0.001) | 0.005 (0.000) | 0.344 (0.028) | 0.939 (0.002) | 0.021 (0.002) | 0.007 (0.001) |
| | | 400 | 0.176 (0.012) | 0.948 (0.001) | 0.011 (0.001) | 0.004 (0.000) | 0.274 (0.021) | 0.942 (0.002) | 0.017 (0.001) | 0.005 (0.000) |
| | | 800 | 0.125 (0.010) | 0.948 (0.001) | 0.008 (0.001) | 0.003 (0.000) | 0.205 (0.016) | 0.944 (0.001) | 0.013 (0.001) | 0.004 (0.000) |

## A.5. Comparison with other estimation methods

When the data are not separable, we compare the performance of three methods in estimating $\kappa$ :

(i) our method in Algorithm 1, denoted as DS; (ii) SLOE, the MLE-based approach of Yadlowsky et al. (2021); and (iii) ProbeFrontier, the MLE-based procedure of Sur & Candès (2019).

Since ProbeFrontier relies on repeated subsampling to detect the frontier of the phase transition curve, this process is computationally expensive so we only repeat each experiment setting 50 times.

We set $\boldsymbol{\Sigma} = \mathbf{I}_p$ and use the same data-generating scheme as described in Appendix A.4. After generating the samples, we numerically verify that all simulated datasets in our experimental settings are non-separable.

The results are presented in Table 6. All three methods perform comparably well, and their estimation accuracy improves as the sample size increases. Together with the experimental results in the regime where the MLE does not exist (i.e., when

*Table 4.* Estimation error and coverage across $\delta$, signal strength ($\kappa$), and dimension ($p$). Values are mean (standard error). Left: $\sigma_*$-based; Right: $\kappa^2\alpha_*^2$-based estimation. For each $(\delta, \kappa)$, results for $p = 100, 200, 400, 800$ are listed vertically.

| $\delta$ | $\kappa$ | $p$ | $\sigma_*$-based | | | | $\kappa^2\alpha_*^2$-based | | | |
|---|---|---|---|---|---|---|---|---|---|---|
| | | | $\|\kappa - \hat{\kappa}\|$ | Coverage | $\|\alpha_* - \hat{\alpha}\|$ | $\|\sigma_* - \hat{\sigma}\|$ | $\|\kappa - \hat{\kappa}\|$ | Coverage | $\|\alpha_* - \hat{\alpha}\|$ | $\|\sigma_* - \hat{\sigma}\|$ |
| | | | | | | $\delta = 3$ | | | | |
| 3 | 1.23 | 100 | 0.234 (0.019) | 0.983 (0.002) | 0.042 (0.003) | 0.027 (0.002) | 0.227 (0.020) | 0.982 (0.002) | 0.039 (0.003) | 0.026 (0.002) |
| | 1.23 | 200 | 0.164 (0.013) | 0.983 (0.001) | 0.029 (0.002) | 0.019 (0.001) | 0.156 (0.013) | 0.983 (0.001) | 0.027 (0.002) | 0.018 (0.001) |
| | 1.23 | 400 | 0.097 (0.008) | 0.983 (0.001) | 0.017 (0.001) | 0.011 (0.001) | 0.098 (0.008) | 0.983 (0.001) | 0.017 (0.001) | 0.011 (0.001) |
| | 1.23 | 800 | 0.076 (0.006) | 0.984 (0.001) | 0.014 (0.001) | 0.009 (0.001) | 0.071 (0.005) | 0.984 (0.001) | 0.013 (0.001) | 0.008 (0.001) |
| 3 | 2.43 | 100 | 0.349 (0.029) | 0.982 (0.002) | 0.043 (0.003) | 0.029 (0.002) | 0.406 (0.036) | 0.979 (0.002) | 0.046 (0.004) | 0.031 (0.002) |
| | 2.45 | 200 | 0.243 (0.023) | 0.981 (0.001) | 0.029 (0.002) | 0.020 (0.002) | 0.270 (0.026) | 0.981 (0.001) | 0.031 (0.003) | 0.021 (0.002) |
| | 2.46 | 400 | 0.195 (0.014) | 0.982 (0.001) | 0.023 (0.002) | 0.016 (0.001) | 0.183 (0.014) | 0.982 (0.001) | 0.022 (0.002) | 0.015 (0.001) |
| | 2.43 | 800 | 0.138 (0.009) | 0.984 (0.001) | 0.017 (0.001) | 0.012 (0.001) | 0.133 (0.010) | 0.984 (0.000) | 0.016 (0.001) | 0.011 (0.001) |
| 3 | 3.59 | 100 | 0.646 (0.061) | 0.979 (0.002) | 0.050 (0.004) | 0.033 (0.002) | 0.649 (0.069) | 0.977 (0.002) | 0.045 (0.004) | 0.029 (0.002) |
| | 3.69 | 200 | 0.380 (0.028) | 0.984 (0.001) | 0.029 (0.002) | 0.019 (0.001) | 0.401 (0.029) | 0.983 (0.001) | 0.029 (0.002) | 0.019 (0.001) |
| | 3.68 | 400 | 0.337 (0.028) | 0.982 (0.001) | 0.024 (0.002) | 0.016 (0.001) | 0.344 (0.029) | 0.982 (0.001) | 0.025 (0.002) | 0.016 (0.001) |
| | 3.63 | 800 | 0.216 (0.015) | 0.983 (0.001) | 0.017 (0.001) | 0.011 (0.001) | 0.217 (0.016) | 0.983 (0.001) | 0.016 (0.001) | 0.010 (0.001) |
| 3 | 4.75 | 100 | 0.915 (0.068) | 0.976 (0.002) | 0.046 (0.003) | 0.027 (0.002) | 1.064 (0.072) | 0.974 (0.002) | 0.049 (0.003) | 0.027 (0.002) |
| | 4.94 | 200 | 0.776 (0.059) | 0.979 (0.001) | 0.036 (0.003) | 0.021 (0.001) | 0.849 (0.069) | 0.976 (0.002) | 0.037 (0.003) | 0.020 (0.002) |
| | 4.97 | 400 | 0.541 (0.043) | 0.982 (0.001) | 0.025 (0.002) | 0.014 (0.001) | 0.606 (0.048) | 0.981 (0.001) | 0.027 (0.002) | 0.015 (0.001) |
| | 4.86 | 800 | 0.430 (0.032) | 0.984 (0.001) | 0.020 (0.001) | 0.012 (0.001) | 0.442 (0.037) | 0.983 (0.001) | 0.021 (0.002) | 0.012 (0.001) |
| | | | | | | $\delta = 4$ | | | | |
| 4 | 1.23 | 100 | 0.221 (0.016) | 0.972 (0.002) | 0.037 (0.003) | 0.019 (0.001) | 0.199 (0.015) | 0.973 (0.002) | 0.033 (0.002) | 0.017 (0.001) |
| | 1.23 | 200 | 0.107 (0.008) | 0.977 (0.001) | 0.018 (0.001) | 0.009 (0.001) | 0.101 (0.008) | 0.976 (0.001) | 0.017 (0.001) | 0.008 (0.001) |
| | 1.23 | 400 | 0.092 (0.007) | 0.977 (0.001) | 0.015 (0.001) | 0.008 (0.001) | 0.093 (0.006) | 0.977 (0.001) | 0.015 (0.001) | 0.008 (0.001) |
| | 1.23 | 800 | 0.066 (0.005) | 0.975 (0.001) | 0.012 (0.001) | 0.006 (0.000) | 0.063 (0.005) | 0.975 (0.001) | 0.011 (0.001) | 0.006 (0.000) |
| 4 | 2.43 | 100 | 0.265 (0.022) | 0.975 (0.002) | 0.032 (0.003) | 0.017 (0.001) | 0.261 (0.023) | 0.975 (0.002) | 0.031 (0.003) | 0.017 (0.001) |
| | 2.45 | 200 | 0.192 (0.016) | 0.976 (0.001) | 0.023 (0.002) | 0.012 (0.001) | 0.181 (0.016) | 0.975 (0.001) | 0.021 (0.002) | 0.012 (0.001) |
| | 2.46 | 400 | 0.157 (0.011) | 0.975 (0.001) | 0.018 (0.001) | 0.010 (0.001) | 0.159 (0.011) | 0.975 (0.001) | 0.018 (0.001) | 0.010 (0.001) |
| | 2.43 | 800 | 0.119 (0.009) | 0.977 (0.001) | 0.014 (0.001) | 0.008 (0.001) | 0.102 (0.008) | 0.977 (0.001) | 0.012 (0.001) | 0.007 (0.001) |
| 4 | 3.59 | 100 | 0.472 (0.038) | 0.972 (0.002) | 0.035 (0.002) | 0.018 (0.001) | 0.475 (0.040) | 0.970 (0.002) | 0.033 (0.002) | 0.017 (0.001) |
| | 3.69 | 200 | 0.389 (0.029) | 0.973 (0.001) | 0.028 (0.002) | 0.014 (0.001) | 0.399 (0.031) | 0.972 (0.001) | 0.028 (0.002) | 0.014 (0.001) |
| | 3.68 | 400 | 0.246 (0.016) | 0.977 (0.001) | 0.018 (0.001) | 0.009 (0.001) | 0.213 (0.017) | 0.977 (0.001) | 0.016 (0.001) | 0.008 (0.001) |
| | 3.63 | 800 | 0.231 (0.019) | 0.975 (0.001) | 0.017 (0.001) | 0.009 (0.001) | 0.186 (0.014) | 0.975 (0.001) | 0.014 (0.001) | 0.007 (0.001) |
| 4 | 4.75 | 100 | 0.434 (0.035) | 0.974 (0.002) | 0.024 (0.002) | 0.012 (0.001) | 0.393 (0.033) | 0.974 (0.002) | 0.021 (0.002) | 0.010 (0.001) |
| | 4.94 | 200 | 0.320 (0.036) | 0.976 (0.001) | 0.017 (0.002) | 0.008 (0.001) | 0.254 (0.031) | 0.977 (0.001) | 0.013 (0.002) | 0.006 (0.001) |
| | 4.97 | 400 | 0.204 (0.026) | 0.976 (0.001) | 0.010 (0.001) | 0.005 (0.001) | 0.169 (0.022) | 0.977 (0.001) | 0.008 (0.001) | 0.004 (0.000) |
| | 4.86 | 800 | 0.214 (0.019) | 0.976 (0.001) | 0.011 (0.001) | 0.005 (0.001) | 0.180 (0.016) | 0.976 (0.001) | 0.009 (0.001) | 0.004 (0.000) |

the data are separable), these findings suggest that our method (DS) performs robustly—delivering accurate estimates of $\kappa$ regardless of whether the data are separable or not.

*Table 5.* Estimation error and coverage across $\delta$, signal strength ($\kappa$), and dimension ($p$). Values are mean (standard error). Left: $\sigma_*$-based; Right: $\kappa^2\alpha_*^2$-based estimation. For each $(\delta, \kappa)$, results for $p = 100, 200, 400, 800$ are listed vertically.

| $\delta$ | $\kappa$ | $p$ | $\sigma_*$-based | | | | $\kappa^2\alpha_*^2$-based | | | |
|---|---|---|---|---|---|---|---|---|---|---|
| | | | $|\kappa - \hat{\kappa}|$ | Coverage | $|\alpha_* - \hat{\alpha}|$ | $|\sigma_* - \hat{\sigma}|$ | $|\kappa - \hat{\kappa}|$ | Coverage | $|\alpha_* - \hat{\alpha}|$ | $|\sigma_* - \hat{\sigma}|$ |
| | | | | | | $\delta = 6$ | | | | |
| 6 | 1.23 | 100 | 0.163 (0.011) | 0.973 (0.002) | 0.027 (0.002) | 0.010 (0.001) | 0.134 (0.010) | 0.972 (0.002) | 0.022 (0.002) | 0.008 (0.001) |
| | 1.23 | 200 | 0.106 (0.007) | 0.970 (0.001) | 0.017 (0.001) | 0.006 (0.000) | 0.089 (0.007) | 0.970 (0.001) | 0.014 (0.001) | 0.005 (0.000) |
| | 1.23 | 400 | 0.073 (0.005) | 0.969 (0.001) | 0.013 (0.001) | 0.005 (0.000) | 0.057 (0.004) | 0.969 (0.001) | 0.010 (0.001) | 0.003 (0.000) |
| | 1.23 | 800 | 0.078 (0.006) | 0.968 (0.001) | 0.012 (0.001) | 0.005 (0.000) | 0.047 (0.004) | 0.968 (0.001) | 0.008 (0.001) | 0.003 (0.000) |
| 6 | 2.43 | 100 | 0.225 (0.018) | 0.964 (0.002) | 0.026 (0.002) | 0.011 (0.001) | 0.207 (0.016) | 0.964 (0.002) | 0.024 (0.002) | 0.010 (0.001) |
| | 2.45 | 200 | 0.167 (0.015) | 0.966 (0.002) | 0.018 (0.002) | 0.008 (0.001) | 0.164 (0.014) | 0.965 (0.002) | 0.018 (0.001) | 0.008 (0.001) |
| | 2.46 | 400 | 0.114 (0.009) | 0.968 (0.001) | 0.013 (0.001) | 0.006 (0.000) | 0.107 (0.008) | 0.968 (0.001) | 0.012 (0.001) | 0.005 (0.000) |
| | 2.43 | 800 | 0.103 (0.008) | 0.968 (0.001) | 0.012 (0.001) | 0.005 (0.000) | 0.069 (0.006) | 0.968 (0.001) | 0.008 (0.001) | 0.003 (0.000) |
| 6 | 3.59 | 100 | 0.411 (0.036) | 0.963 (0.002) | 0.029 (0.002) | 0.011 (0.001) | 0.394 (0.034) | 0.963 (0.002) | 0.027 (0.002) | 0.011 (0.001) |
| | 3.69 | 200 | 0.285 (0.020) | 0.963 (0.002) | 0.020 (0.001) | 0.008 (0.000) | 0.235 (0.019) | 0.964 (0.002) | 0.016 (0.001) | 0.006 (0.000) |
| | 3.68 | 400 | 0.213 (0.016) | 0.966 (0.001) | 0.015 (0.001) | 0.006 (0.000) | 0.179 (0.014) | 0.966 (0.001) | 0.012 (0.001) | 0.005 (0.000) |
| | 3.63 | 800 | 0.182 (0.014) | 0.968 (0.001) | 0.013 (0.001) | 0.005 (0.000) | 0.137 (0.011) | 0.968 (0.001) | 0.010 (0.001) | 0.004 (0.000) |
| 6 | 4.75 | 100 | 0.608 (0.050) | 0.958 (0.003) | 0.029 (0.002) | 0.010 (0.001) | 0.567 (0.053) | 0.956 (0.003) | 0.026 (0.002) | 0.009 (0.001) |
| | 4.94 | 200 | 0.542 (0.037) | 0.960 (0.002) | 0.024 (0.002) | 0.009 (0.001) | 0.450 (0.035) | 0.961 (0.002) | 0.020 (0.001) | 0.007 (0.001) |
| | 4.97 | 400 | 0.559 (0.045) | 0.960 (0.002) | 0.024 (0.002) | 0.008 (0.001) | 0.319 (0.025) | 0.966 (0.001) | 0.014 (0.001) | 0.005 (0.000) |
| | 4.86 | 800 | 0.321 (0.022) | 0.964 (0.001) | 0.015 (0.001) | 0.006 (0.000) | 0.205 (0.015) | 0.965 (0.001) | 0.010 (0.001) | 0.004 (0.000) |
| | | | | | | $\delta = 8$ | | | | |
| 8 | 1.23 | 100 | 0.131 (0.010) | 0.963 (0.002) | 0.020 (0.002) | 0.006 (0.000) | 0.105 (0.008) | 0.964 (0.002) | 0.017 (0.001) | 0.005 (0.000) |
| | 1.23 | 200 | 0.084 (0.007) | 0.963 (0.001) | 0.013 (0.001) | 0.004 (0.000) | 0.069 (0.006) | 0.963 (0.001) | 0.011 (0.001) | 0.003 (0.000) |
| | 1.23 | 400 | 0.067 (0.005) | 0.967 (0.001) | 0.012 (0.001) | 0.004 (0.000) | 0.051 (0.004) | 0.967 (0.001) | 0.009 (0.001) | 0.003 (0.000) |
| | 1.23 | 800 | 0.085 (0.007) | 0.964 (0.001) | 0.013 (0.001) | 0.004 (0.000) | 0.044 (0.003) | 0.964 (0.001) | 0.007 (0.001) | 0.002 (0.000) |
| 8 | 2.43 | 100 | 0.166 (0.014) | 0.961 (0.002) | 0.019 (0.002) | 0.007 (0.001) | 0.148 (0.012) | 0.960 (0.002) | 0.016 (0.001) | 0.006 (0.000) |
| | 2.45 | 200 | 0.148 (0.011) | 0.961 (0.001) | 0.017 (0.001) | 0.006 (0.000) | 0.118 (0.010) | 0.961 (0.001) | 0.013 (0.001) | 0.005 (0.000) |
| | 2.46 | 400 | 0.158 (0.012) | 0.966 (0.001) | 0.018 (0.001) | 0.006 (0.000) | 0.091 (0.006) | 0.967 (0.001) | 0.010 (0.001) | 0.004 (0.000) |
| | 2.43 | 800 | 0.104 (0.008) | 0.963 (0.001) | 0.012 (0.001) | 0.004 (0.000) | 0.063 (0.005) | 0.964 (0.001) | 0.008 (0.001) | 0.003 (0.000) |
| 8 | 3.59 | 100 | 0.313 (0.028) | 0.958 (0.002) | 0.023 (0.002) | 0.008 (0.001) | 0.277 (0.022) | 0.960 (0.002) | 0.020 (0.002) | 0.007 (0.001) |
| | 3.69 | 200 | 0.247 (0.021) | 0.958 (0.002) | 0.017 (0.001) | 0.006 (0.000) | 0.215 (0.016) | 0.959 (0.002) | 0.015 (0.001) | 0.005 (0.000) |
| | 3.68 | 400 | 0.228 (0.018) | 0.961 (0.001) | 0.016 (0.001) | 0.005 (0.000) | 0.174 (0.012) | 0.962 (0.001) | 0.012 (0.001) | 0.004 (0.000) |
| | 3.63 | 800 | 0.172 (0.012) | 0.961 (0.001) | 0.013 (0.001) | 0.004 (0.000) | 0.116 (0.008) | 0.962 (0.001) | 0.009 (0.001) | 0.003 (0.000) |
| 8 | 4.75 | 100 | 0.548 (0.048) | 0.948 (0.003) | 0.026 (0.002) | 0.008 (0.001) | 0.510 (0.046) | 0.948 (0.003) | 0.023 (0.002) | 0.007 (0.001) |
| | 4.94 | 200 | 0.376 (0.028) | 0.959 (0.002) | 0.017 (0.001) | 0.005 (0.000) | 0.326 (0.027) | 0.959 (0.002) | 0.015 (0.001) | 0.004 (0.000) |
| | 4.97 | 400 | 0.457 (0.040) | 0.954 (0.002) | 0.020 (0.002) | 0.006 (0.000) | 0.243 (0.017) | 0.960 (0.001) | 0.011 (0.001) | 0.003 (0.000) |
| | 4.86 | 800 | 0.257 (0.021) | 0.959 (0.001) | 0.012 (0.001) | 0.004 (0.000) | 0.153 (0.011) | 0.960 (0.001) | 0.007 (0.001) | 0.002 (0.000) |

*Table 6.* Estimation error across $\delta$, signal strength ($\kappa$), and dimension ($p$). Values are mean (standard error) of absolute estimation error $|\hat{\kappa} - \kappa|$. Three methods compared: DS, SLOE, and ProbeFrontier.

| $\delta$ | $\kappa$ | $p$ | DS | SLOE | ProbeFrontier |
|---|---|---|---|---|---|
| | | | | $\delta = 6$ | |
| 6 | 1.0 | 100 | 0.157(0.016) | 0.151(0.014) | 0.132(0.014) |
| | | 200 | 0.111(0.011) | 0.101(0.009) | 0.110(0.011) |
| | | 400 | 0.063(0.007) | 0.058(0.006) | 0.055(0.006) |
| | | 800 | 0.041(0.005) | 0.043(0.004) | 0.047(0.005) |
| 6 | 1.5 | 100 | 0.201(0.019) | 0.171(0.017) | 0.157(0.015) |
| | | 200 | 0.122(0.013) | 0.106(0.010) | 0.101(0.009) |
| | | 400 | 0.072(0.008) | 0.067(0.009) | 0.069(0.009) |
| | | 800 | 0.052(0.005) | 0.049(0.005) | 0.056(0.005) |
| 6 | 2.0 | 100 | 0.281(0.031) | 0.208(0.025) | 0.179(0.021) |
| | | 200 | 0.159(0.021) | 0.130(0.014) | 0.139(0.013) |
| | | 400 | 0.093(0.010) | 0.088(0.008) | 0.089(0.009) |
| | | 800 | 0.075(0.007) | 0.066(0.007) | 0.076(0.007) |
| 6 | 2.5 | 100 | 0.340(0.031) | 0.218(0.024) | 0.211(0.020) |
| | | 200 | 0.223(0.028) | 0.150(0.019) | 0.154(0.016) |
| | | 400 | 0.120(0.013) | 0.088(0.011) | 0.104(0.009) |
| | | 800 | 0.092(0.010) | 0.079(0.009) | 0.090(0.009) |
| | | | | $\delta = 8$ | |
| 8 | 1.0 | 100 | 0.107(0.012) | 0.099(0.012) | 0.101(0.013) |
| | | 200 | 0.076(0.008) | 0.072(0.008) | 0.074(0.008) |
| | | 400 | 0.056(0.006) | 0.051(0.005) | 0.068(0.007) |
| | | 800 | 0.035(0.003) | 0.034(0.003) | 0.043(0.004) |
| 8 | 1.5 | 100 | 0.128(0.014) | 0.111(0.012) | 0.112(0.015) |
| | | 200 | 0.084(0.010) | 0.082(0.009) | 0.094(0.010) |
| | | 400 | 0.059(0.007) | 0.054(0.006) | 0.066(0.007) |
| | | 800 | 0.045(0.005) | 0.041(0.005) | 0.050(0.005) |
| 8 | 2.0 | 100 | 0.160(0.022) | 0.144(0.013) | 0.139(0.014) |
| | | 200 | 0.113(0.011) | 0.101(0.011) | 0.116(0.011) |
| | | 400 | 0.073(0.008) | 0.057(0.007) | 0.078(0.008) |
| | | 800 | 0.059(0.006) | 0.053(0.005) | 0.078(0.008) |
| 8 | 2.5 | 100 | 0.211(0.024) | 0.154(0.016) | 0.156(0.017) |
| | | 200 | 0.154(0.014) | 0.144(0.015) | 0.155(0.017) |
| | | 400 | 0.097(0.009) | 0.072(0.009) | 0.090(0.013) |
| | | 800 | 0.079(0.009) | 0.066(0.007) | 0.092(0.010) |

## A.6. Efficiency gain by multiple splits

In Section 5, we propose a data-splitting method to estimate the variance factor $\kappa^2 \alpha_*^2(\delta/2, \kappa)$, which is then used in Algorithm 1 to estimate the signal strength $\kappa$. To improve statistical efficiency, we leverage the full data information by averaging the estimates across multiple independent data splits. Specifically, we generate 100 independent datasets, and for each dataset we compute the estimate $\hat{\kappa}$ based on different numbers of splits.

We compare the efficiency of our proposed multiple data splitting (MDS) based method across varying numbers of splits. Over 100 independent experiments, we compute the mean squared error (MSE) $\mathbb{E}(\hat{\kappa} - \kappa)^2$. The relative efficiency is measured as the ratio of the variance of the estimate $\hat{\kappa}$ compared to the case with 100 splits.

*Table 7.* Efficiency of the MDS-based method across different numbers of splits.

| Number of splits | 1 | 2 | 4 | 8 | 16 | 32 | 64 | 100 |
|---|---|---|---|---|---|---|---|---|
| MSE | 0.132 | 0.082 | 0.074 | 0.067 | 0.062 | 0.056 | 0.053 | 0.053 |
| Efficiency | 0.401 | 0.651 | 0.722 | 0.803 | 0.851 | 0.944 | 1.000 | 1.000 |

The results are shown in Table 7. We find that using a single split yields only about 40% of the efficiency achieved with many splits, indicating a substantial loss of information. The performance improves monotonically as the number of splits increases, with efficiency reaching 80% at 8 splits and approximately 95% at 32 splits. The gains plateau beyond 50 splits, suggesting that around 32 to 64 splits are sufficient to achieve near-optimal efficiency while maintaining computational tractability.

## A.7. Comparison with $L_1$ and $L_2$ Regularization

We compare our proposed covariance-adaptive regularization with standard $L_1$ (lasso) and $L_2$ (ridge) regularization. When $\boldsymbol{\Sigma} = \mathrm{Cov}(\boldsymbol{X}) = \mathbf{I}_p$, all regularizations perform similarly in terms of estimation error; we therefore omit these results for brevity and focus on the more challenging case where $\boldsymbol{\Sigma}$ deviates significantly from the identity.

To investigate the effect of heterogeneous covariance structure, we set

$$\boldsymbol{\Sigma} = \mathrm{diag}(\lambda_1, \lambda_2, \ldots, \lambda_p),$$

where the eigenvalues $\lambda_i$ are linearly spaced between $1/4$ and $10$. The true coefficient vector is chosen to be inversely proportional to the standard deviations:

$$\boldsymbol{\beta}_* = c \cdot (\lambda_1^{-1}, \lambda_2^{-1}, \ldots, \lambda_p^{-1})^\top,$$

where the scalar $c$ is chosen to control $\|\boldsymbol{\beta}_*\|_2$.

We consider four regularization schemes:

1. **Adaptive (empirical):** $0.1\|\widehat{\boldsymbol{\Sigma}}^{1/2}\boldsymbol{\beta}\|_2^2$ — uses the sample covariance;

2. **Adaptive (oracle):** $0.1\|\boldsymbol{\Sigma}^{1/2}\boldsymbol{\beta}\|_2^2$ — uses the true covariance;

3. **$L_1$:** $0.1\|\boldsymbol{\beta}\|_1$ — standard lasso;

4. **$L_2$:** $0.1\|\boldsymbol{\beta}\|_2^2$ — standard ridge.

The first two regularizations are scale-invariant and adapt to the covariance structure: they apply stronger shrinkage in high-variance directions (where the true coefficients are small) and weaker shrinkage in low-variance directions (where the true coefficients are large). This aligns with the data-generating process, whereas standard $L_2$ applies uniform shrinkage regardless of predictor variance.

Table 8 reports the estimation error ($\|\widehat{\boldsymbol{\beta}} - \boldsymbol{\beta}_*\|_2$) and classification error (CE) for each method across different aspect ratios $\delta = n/p$ and signal strengths $\|\boldsymbol{\beta}_*\|_2$.

We find: The covariance-adaptive regularizations (columns $\widehat{\boldsymbol{\Sigma}}$ and $\boldsymbol{\Sigma}$) consistently outperform $L_1$ and $L_2$ in estimation error, with the oracle version ($\boldsymbol{\Sigma}$) achieving the lowest error across all settings. The performance gap is most pronounced at higher

*Table 8.* Comparison of regularization methods under heterogeneous covariance. Columns 3–6 report estimation error; columns 7–10 report classification error (CE).

| $\delta$ | $\|\boldsymbol{\beta}_*\|_2$ | Estimation Error | | | | Classification Error | | | |
|---|---|---|---|---|---|---|---|---|---|
| | | $\widehat{\boldsymbol{\Sigma}}$ | $\boldsymbol{\Sigma}$ | $L_1$ | $L_2$ | $\widehat{\boldsymbol{\Sigma}}$ | $\boldsymbol{\Sigma}$ | $L_1$ | $L_2$ |
| 2 | 1.0 | 1.01 | 0.765 | 1.68 | 1.14 | 0.437 | 0.423 | 0.444 | 0.447 |
| 2 | 1.5 | 1.08 | 0.944 | 1.78 | 1.33 | 0.396 | 0.369 | 0.400 | 0.411 |
| 2 | 2.0 | 1.27 | 1.23 | 1.85 | 1.62 | 0.354 | 0.326 | 0.368 | 0.379 |
| 3 | 1.0 | 0.725 | 0.645 | 1.00 | 0.881 | 0.408 | 0.404 | 0.417 | 0.425 |
| 3 | 1.5 | 0.855 | 0.836 | 1.11 | 1.11 | 0.357 | 0.348 | 0.368 | 0.379 |
| 3 | 2.0 | 1.10 | 1.13 | 1.20 | 1.42 | 0.314 | 0.302 | 0.326 | 0.341 |
| 4 | 1.0 | 0.603 | 0.573 | 0.773 | 0.763 | 0.400 | 0.392 | 0.408 | 0.416 |
| 4 | 1.5 | 0.775 | 0.782 | 0.876 | 1.01 | 0.337 | 0.331 | 0.349 | 0.361 |
| 4 | 2.0 | 1.07 | 1.10 | 0.995 | 1.35 | 0.294 | 0.292 | 0.307 | 0.325 |

signal strengths ($\|\boldsymbol{\beta}_*\|_2 = 2.0$), where the adaptive shrinkage correctly accounts for the heterogeneous variance structure. For classification error, all methods perform more similarly, though the adaptive regularizations maintain a slight advantage, particularly in high-signal regimes.

### A.8. Sensitivity Analysis with Respect to $\lambda$

In this section, we investigate the sensitivity of the proposed estimator with respect to the tuning parameter $\lambda$. To this end, we conduct a systematic sensitivity analysis by considering a range of values $\lambda \in \{0.1, 0.2, 0.4, 0.8\}$ under different aspect ratios $\delta = n/p$. Throughout this analysis, we fix the dimension $p = 500$ and the true signal strength $\kappa = 2$. For each configuration, we perform 100 independent runs and report the mean, median, and the first and third quartiles of the estimated signal strength $\hat{\kappa}$. The results are summarized in Table 9. As can be seen from the table, the estimator exhibits only minor variation across the different choices of $\lambda$, suggesting that the estimation procedure is relatively insensitive to the specific value of this tuning parameter.

*Table 9.* Sensitivity of $\hat{\kappa}$ with respect to $\lambda$ under different aspect ratios $\delta = n/p$. The results are based on 100 independent runs with $p = 500$ and $\kappa = 2$.

| $\delta$ | $\lambda$ | $\kappa_{\text{mean}}$ | $\kappa_{\text{median}}$ | $\kappa_{\text{q25}}$ | $\kappa_{\text{q75}}$ |
|---|---|---|---|---|---|
| 4 | 0.1 | 2.16 | 2.16 | 1.99 | 2.31 |
| 4 | 0.2 | 2.10 | 2.10 | 1.93 | 2.26 |
| 4 | 0.4 | 2.08 | 2.07 | 1.91 | 2.24 |
| 4 | 0.8 | 2.07 | 2.07 | 1.90 | 2.24 |
| 8 | 0.1 | 2.08 | 2.07 | 2.02 | 2.15 |
| 8 | 0.2 | 2.04 | 2.03 | 1.98 | 2.10 |
| 8 | 0.4 | 2.01 | 2.00 | 1.95 | 2.07 |
| 8 | 0.8 | 2.00 | 1.99 | 1.94 | 2.06 |

To further assess the robustness of the estimation procedure, we also report the mean and quantiles of the absolute estimation error $|\hat{\kappa} - \kappa|$ in Table 10. These results corroborate the findings above and confirm that the estimation accuracy remains stable across different values of $\lambda$. We observe qualitatively similar behavior for the other estimated quantities, namely $\alpha_*$ and $\sigma_*$.

Finally, regarding the practical selection of $\lambda$ in applications where a default value may not be available, we have included a data-driven procedure in the revision. Specifically, using the leave-one-out (LOO) approximation described in Appendix E.1, we approximate $\boldsymbol{X}_i^\top \widehat{\boldsymbol{\beta}}_{-i,\lambda}$, compute the corresponding deviance, and select the value of $\lambda$ that yields the best fit.

## B. Proof of Theorem 3.1 part 1: boundedness of $\|\widehat{\boldsymbol{\beta}}_\lambda\|$

We first rewrite the objective function as follows.

$$\widehat{\boldsymbol{\beta}}_\lambda = \arg\min_{\boldsymbol{\beta}} \frac{1}{n} \sum_{i=1}^{n} \log\left(1 + \exp(-(2Y_i - 1)\boldsymbol{X}_i^\top \boldsymbol{\beta})\right) + \frac{\lambda}{2}\boldsymbol{\beta}^\top \left(\frac{1}{n}\sum_{i=1}^{n} \boldsymbol{X}_i \boldsymbol{X}_i^\top\right)\boldsymbol{\beta}$$

*Table 10.* Sensitivity of the absolute error $|\hat{\kappa} - \kappa|$ with respect to $\lambda$. The results are based on 100 independent runs with $p = 500$ and $\kappa = 2$.

| $\delta$ | $\lambda$ | mean | median | 25% | 75% |
|---|---|---|---|---|---|
| 4 | 0.1 | 0.2130 | 0.1790 | 0.0725 | 0.3140 |
| 4 | 0.2 | 0.1940 | 0.1560 | 0.0899 | 0.2760 |
| 4 | 0.4 | 0.1920 | 0.1450 | 0.0838 | 0.2740 |
| 4 | 0.8 | 0.1920 | 0.1550 | 0.0779 | 0.2700 |
| 8 | 0.1 | 0.1010 | 0.0807 | 0.0440 | 0.1550 |
| 8 | 0.2 | 0.0795 | 0.0655 | 0.0272 | 0.1290 |
| 8 | 0.4 | 0.0752 | 0.0613 | 0.0243 | 0.1150 |
| 8 | 0.8 | 0.0761 | 0.0633 | 0.0261 | 0.1160 |

The objective function evaluated at $\widehat{\boldsymbol{\beta}}_\lambda$ is necessarily smaller than that evaluated at $\boldsymbol{\beta} = \mathbf{0}$, which is $\log(2)$. Next, note that both part of objective function is positive, we have

$$\frac{\lambda}{2}\|\widehat{\boldsymbol{\beta}}_\lambda\|^2 \lambda_{min}\left(\frac{1}{n}\sum_{i=1}^n \boldsymbol{X}_i\boldsymbol{X}_i^\top\right) \le \frac{\lambda}{2}\widehat{\boldsymbol{\beta}}_\lambda^\top\left(\frac{1}{n}\sum_{i=1}^n \boldsymbol{X}_i\boldsymbol{X}_i^\top\right)\widehat{\boldsymbol{\beta}}_\lambda$$

$$\le \frac{1}{n}\sum_{i=1}^n \log\left(1 + \exp(-(2Y_i - 1)\boldsymbol{X}_i^\top\widehat{\boldsymbol{\beta}}_\lambda)\right) + \frac{\lambda}{2}\widehat{\boldsymbol{\beta}}_\lambda^\top\left(\frac{1}{n}\sum_{i=1}^n \boldsymbol{X}_i\boldsymbol{X}_i^\top\right)\widehat{\boldsymbol{\beta}}_\lambda$$

$$\le \log(2).$$

It remains to bound the minimum eigenvalue of sample covariance matrix, following (Vershynin, 2010), we have

$$\mathbb{P}\left(\lambda_{\min}(\widehat{\Sigma}) \le \lambda_{\min}(\Sigma)\left(1 - \sqrt{\frac{p}{n}} - \sqrt{\frac{2t}{n}}\right)^2\right) \le e^{-t}.$$

The proof is completed by taking $t = \frac{1}{8}(1 - \sqrt{1/\delta})^2 n$.

## C. Proof of Theorem 3.1 part 2: exact asymptotic of $\widehat{\boldsymbol{\beta}}_\lambda$

We prove the case where $\mathrm{Cov}(\boldsymbol{X}_i) = \mathbf{I}_p$. The extension to general covariance follows from a similar argument as in (Zhao et al., 2022).

We recall the distributional conditions and streamline the notations. The observed covariates are $\{\boldsymbol{X}_i\}_{i=1}^n \overset{\text{i.i.d.}}{\sim} N(\mathbf{0}, \mathbf{I}_p)$. Additionally, the observed responses are $Y_i \sim \mathrm{Bern}(\rho'(\boldsymbol{X}_i^\top\boldsymbol{\beta}_*))$, we aim to analyze following estimator:

$$\widehat{\boldsymbol{\beta}} = \arg\min_{\boldsymbol{\beta}} \frac{1}{n}\sum_{i=1}^n [\rho(\boldsymbol{X}^\top\boldsymbol{\beta}) - Y_i\boldsymbol{X}_i^\top\boldsymbol{\beta}] + \frac{\lambda}{2n}\left(\sum_{i=1}^n (\boldsymbol{X}_i^\top\boldsymbol{\beta})^2\right) \tag{16}$$

Our analysis is based on the Convex Gaussian Min-max Theorem (CGMT), which we will briefly review here; detailed theory and application can be found in (Thrampoulidis et al., 2015; Thrampoulidis, 2016; Thrampoulidis et al., 2018). This technique connects a Primary Optimization (PO) problem with an Auxiliary Optimization (AO) problem, which is easy to analyze yet allows studying various aspects of the PO. Specifically, we define the PO and AO problems as follows:

$$(\text{PO}) \quad \Phi(\mathbf{G}) := \min_{\mathbf{w}\in\mathcal{S}_\mathbf{w}} \max_{\mathbf{u}\in\mathcal{S}_\mathbf{u}} \mathbf{u}^T\mathbf{G}\mathbf{w} + \psi(\mathbf{u}, \mathbf{w}) \tag{17}$$

$$(\text{AO}) \quad \phi(\mathbf{g}, \mathbf{h}) := \min_{\mathbf{w}\in\mathcal{S}_\mathbf{w}} \max_{\mathbf{u}\in\mathcal{S}_\mathbf{u}} \|\mathbf{w}\|\mathbf{g}^T\mathbf{u} - \|\mathbf{u}\|\mathbf{h}^T\mathbf{w} + \psi(\mathbf{u}, \mathbf{w}) \tag{18}$$

where $\mathbf{G} \in \mathbb{R}^{m\times n}, \mathbf{g} \in \mathbb{R}^m, \mathbf{h} \in \mathbb{R}^n, \mathcal{S}_\mathbf{w} \subset \mathbb{R}^n, \mathcal{S}_\mathbf{u} \subset \mathbb{R}^m$ and $\psi : \mathbb{R}^n \times \mathbb{R}^m \to \mathbb{R}$. Denote by $\mathbf{w}_\Phi := \mathbf{w}_\Phi(\mathbf{G})$ and $\mathbf{w}_\phi := \mathbf{w}_\phi(\mathbf{g}, \mathbf{h})$ any optimal minimizers in (17) and (18), respectively.

**Lemma C.1** ((Thrampoulidis, 2016)). *Let $\mathcal{S}_\mathbf{w}$ and $\mathcal{S}_\mathbf{u}$ be two convex and compact sets. Assume the function $\psi(\cdot, \cdot)$ is convex-concave on $\mathcal{S}_\mathbf{w} \times \mathcal{S}_\mathbf{u}$. Also assume that $\mathbf{G}$, $\mathbf{g}$, and $\mathbf{h}$ all have entries i.i.d. standard normal. Then for all $\mu \in \mathbb{R}$, and $t > 0$,*

$$\mathbb{P}(|\Phi(\mathbf{G}) - \mu| > t) \leq 2\mathbb{P}(|\phi(\mathbf{g}, \mathbf{h}) - \mu| \geq t)$$

*The probabilities are taken with respect to the randomness in $\mathbf{G}$, $\mathbf{g}$, and $\mathbf{h}$.*

**Lemma C.2** (Asymptotic CGMT (Thrampoulidis, 2016)). *Let $\mathcal{S}$ be an arbitrary open subset of $\mathcal{S}_\mathbf{w}$ and $\mathcal{S}^c := \mathcal{S}_\mathbf{w}/\mathcal{S}$. Denote $\Phi_{\mathcal{S}^c}(\mathbf{G})$ and $\phi_{\mathcal{S}^c}(\mathbf{g}, \mathbf{h})$ be the optimal costs of the optimizations in (17) and (18), respectively, when the minimization over $\mathbf{w}$ is now constrained over $\mathbf{w} \in \mathcal{S}^c$. Suppose that there exists constants $\bar{\phi} < \bar{\phi}_{\mathcal{S}^c}$ such that $\phi(\mathbf{g}, \mathbf{h}) \xrightarrow{\mathbb{P}} \bar{\phi}$, and $\phi_{\mathcal{S}^c}(\mathbf{g}, \mathbf{h}) \longrightarrow \bar{\phi}_{\mathcal{S}^c}$. Then, $\lim_{n \to \infty} \mathbb{P}(\mathbf{w}_\Phi(\mathbf{G}) \in \mathcal{S}) = 1$.*

To make use of CGMT, we shall first rewrite our generalized ridge estimator in the form of PO.

### C.1. Reformulation and transformation

The goal of this subsection is to reformulate the optimization for the estimator (3) into a PO problem and define the associated AO problem. We starts with rewriting the optimization in (3) as

$$\min_{\boldsymbol{\beta} \in \mathbb{R}^p} \left\{ \frac{1}{n} \mathbf{1}^T \rho(\mathbf{H}\boldsymbol{\beta}) - \frac{1}{n} \mathbf{Y}^\top \mathbf{H}\boldsymbol{\beta} + \frac{\lambda}{2n}(\mathbf{H}\boldsymbol{\beta})^\top (\mathbf{H}\boldsymbol{\beta}) \right\}$$

where the action of function $\rho(\cdot)$ on a vector is considered entry-wise, $\mathbf{Y} \in \mathbb{R}^n$ is the vector of observed responses, $\mathbf{H} \in \mathbb{R}^{n \times p}$ is $[\boldsymbol{X}_1, \ldots, \boldsymbol{X}_n]^\top$. Note the entries of $\mathbf{H}$ are i.i.d. standard normal variables.

Introducing a new variable $\mathbf{u}$, we further rewrite the optimiztion as

$$\min_{\boldsymbol{\beta} \in \mathbb{R}^p, \mathbf{u} \in \mathbb{R}^n} \left( \frac{1}{n} \mathbf{1}^T \rho(\mathbf{u}) - \frac{1}{n} \mathbf{Y}^\top \mathbf{u} + \frac{\lambda}{2n} \mathbf{u}^\top \mathbf{u} \right)$$
$$\text{s.t. } \mathbf{u} = \mathbf{H}\boldsymbol{\beta}.$$

Using a Lagrange multiplier, we rewrite the above optimization as a min-max optimization

$$\min_{\boldsymbol{\beta} \in \mathbb{R}^p, \mathbf{u} \in \mathbb{R}^n} \max_{\boldsymbol{v} \in \mathbb{R}^n} \left( \frac{1}{n} \mathbf{1}^T \rho(\mathbf{u}) - \frac{1}{n} \mathbf{Y}^T \mathbf{u} + \frac{\lambda}{2n} \mathbf{u}^\top \mathbf{u} + \frac{1}{\sqrt{n}} \boldsymbol{v}^T (\mathbf{u} - \mathbf{H}\boldsymbol{\beta}) \right) \tag{19}$$

We reformulate the original loss function into a new form specifically designed to facilitate the application of the Convex Gaussian Minimax Theorem (CGMT). This is motivated by the fact that the current min–max optimization problem is affine in the Gaussian matrix $\mathbf{H}$.

To apply CGMT, we further restrict the feasible sets of $\boldsymbol{\beta}$, $\mathbf{u}$ and $\boldsymbol{v}$ in (19) to be compact and convex. The compactness requirement is essential, it satisfies a key technical condition that justifies interchanging the order of minimization and maximization in the minimax theorem.

**Feasible set for optimization** We denote by $(\widehat{\boldsymbol{\beta}}, \widehat{\mathbf{u}}, \widehat{\boldsymbol{v}})$ the solution to the unconstrained min–max problem in (19).

By Part 1 of Theorem 3.1, we can restrict the feasible set $\mathcal{S}_{\boldsymbol{\beta}}$ to a Euclidean ball centered at the origin with a fixed (dimension-independent) radius in $\mathbb{R}^p$ for all $p$.

Moreover, we will show that the norm $\frac{1}{\sqrt{n}}\|\mathbf{u}\|$ can be bounded above and below by universal constants without affecting the value of the original optimization problem.

Note that the first order optimality condition with respect $\boldsymbol{v}$ implies

$$\|\widehat{\mathbf{u}}\| = \left\|\mathbf{H}\widehat{\boldsymbol{\beta}}\right\| \leq \|\mathbf{H}\|_{op}\|\widehat{\boldsymbol{\beta}}\|_2$$

To show $\frac{1}{\sqrt{n}}\|\widehat{\mathbf{u}}\|$ is bounded by some universal constants, it suffices to show $\frac{1}{\sqrt{n}}\|\mathbf{H}\|_{op}$ is bounded by some universal constant for all sufficiently large sample sizes. Using the standard upper bound on the operator norm of Gaussian random

matrices (Vershynin, 2010, Corollary 5.35), we have $\mathbb{P}(\|\mathbf{H}\|_{op} > \sqrt{n} + \sqrt{p} + \sqrt{2n}) \leq 2\exp(-n)$. Recalling that $n/p = \delta$, we have

$$\sum_{n=1}^{\infty} \mathbb{P}\left(\frac{1}{\sqrt{n}}\|\mathbf{H}\|_{op} > 1 + \sqrt{\frac{1}{\delta}} + \sqrt{2}\right) \leq 2\sum_{n=1}^{\infty} \exp(-n) < \infty.$$

By Borel–Cantelli lemma, we conclude that

$$\mathbb{P}\left(\left\{\frac{1}{\sqrt{n}}\|\mathbf{H}\|_{op} > 1 + \sqrt{\frac{1}{\delta}} + \sqrt{2}\right\} \text{ happens infinitely many times}\right) = 0. \tag{20}$$

Thus, it is safe to constrain the feasible sets of $\mathbf{u}$ to be some closed balls with diverging radii $C\sqrt{n}$ for some sufficiently large constant $C$, which are denoted by $\mathcal{S}_{\mathbf{u}}$.

Furthermore, based on the first order optimality of the min-max optimization in (19), we have

$$\mathbf{Y} - \rho'(\widehat{\mathbf{u}}) + \frac{\lambda}{n}\widehat{\mathbf{u}} = \sqrt{n}\widehat{v},$$

where $\widehat{v}$ denotes the maximizer of the inner problem. Since the entries of $\rho'(\widehat{\mathbf{u}}_1), \rho'(\widehat{\mathbf{u}}_2)$ are bounded by 1 and the entries of $\mathbf{Y}$ are either 0 or 1, it follows that $\|\widehat{v}\|^2 \leq 1 + \lambda^2 C^2$. Consequently, we can reduce the feasible set of $v$ to be a closed ball centered at origin with a constant radius. In the following section, we denote these restricted feasible sets by $\mathcal{S}_{\boldsymbol{\beta}}, \mathcal{S}_{\mathbf{u}}$ and $\mathcal{S}_v$.

**Formulations of PO and AO** In order to define the PO and AO problems, we need to decompose $\boldsymbol{\beta}$ into a "signal part" and a "noise part".

Denoted by $S$ the space spanned by $\boldsymbol{\beta}_*$. Let $\mathbf{P}$ be the projection matrix onto $S$ and let $\mathbf{P}^{\perp} := \mathbf{I}_p - \mathbf{P}$ be the projection matrices onto the orthogonal complement of $S$. We use these projections to decompose $\boldsymbol{\beta}$ as the sum of $\boldsymbol{\beta}_S := \mathbf{P}\boldsymbol{\beta}$ and $\boldsymbol{\beta}_{S^{\perp}} := \mathbf{P}^{\perp}\boldsymbol{\beta}$. Since the length and the direction of $\mathbf{P}\boldsymbol{\beta}$ and those of $\mathbf{P}^{\perp}\boldsymbol{\beta}$ are independent with each other, the optimization can be conducted over these directions and lengths separately. Besides, since the feasible set $\mathcal{S}_{\boldsymbol{\beta}}$ we defined earlier is a ball centered at the origin, the images of projections, $\mathbf{P}\mathcal{S}_{\boldsymbol{\beta}}$ and $\mathbf{P}^{\perp}\mathcal{S}_{\boldsymbol{\beta}}$, are convex, compact, and bounded sets. In light of these observations, the optimization can be rewritten as

$$\min_{\substack{\boldsymbol{\beta}_S \in \mathbf{P}\mathcal{S}_{\boldsymbol{\beta}}, \boldsymbol{\beta}_{S^{\perp}} \in \mathbf{P}^{\perp}\mathcal{S}_{\boldsymbol{\beta}} \\ \mathbf{u} \in \mathcal{S}_{\mathbf{u}}}} \max_{v \in \mathbb{R}^n} \left(\frac{1}{n}\mathbf{1}^T\rho(\mathbf{u}) - \frac{1}{n}\mathbf{Y}^T\mathbf{u} + \frac{\lambda}{2n}\mathbf{u}^{\top}\mathbf{u} + \frac{1}{\sqrt{n}}v^T(\mathbf{u} - \mathbf{H}\boldsymbol{\beta}_S) - \frac{1}{\sqrt{n}}v^T\mathbf{H}\boldsymbol{\beta}_{S^{\perp}}\right) \tag{21}$$

In addition, the objective function is jointly convex with respect to $\left(\boldsymbol{\beta}_S, \boldsymbol{\beta}_S^{\perp}, \mathbf{u}\right)$, and is concave with respect to $v$. Based on Sion's minimax theorem and the compactness of all the feasible sets, we can rewrite (21) by flipping the min and max signs as follows

$$\min_{\boldsymbol{\beta}_{S^{\perp}} \in \mathbf{P}^{\perp}\mathcal{S}_{\boldsymbol{\beta}}} \max_{v \in \mathbb{R}^n} \min_{\substack{\boldsymbol{\beta}_S \in \mathbf{P}\mathcal{S}_{\boldsymbol{\beta}} \\ \mathbf{u} \in \mathcal{S}_{\mathbf{u}}}} \left(\frac{1}{n}\mathbf{1}^T\rho(\mathbf{u}) - \frac{1}{n}\mathbf{Y}^T\mathbf{u} + \frac{\lambda}{2n}\mathbf{u}^{\top}\mathbf{u} + \frac{1}{\sqrt{n}}v^T(\mathbf{u} - \mathbf{H}\boldsymbol{\beta}_S) - \frac{1}{\sqrt{n}}v^T\mathbf{H}\boldsymbol{\beta}_{S^{\perp}}\right)$$

It is important to note that the vector of observed responses, $\mathbf{Y}$, is independent of $\mathbf{H}\mathbf{P}^{\perp}$. This independence arises because $\mathbf{H}\boldsymbol{\beta}_* = \mathbf{H}\mathbf{P}\boldsymbol{\beta}_*$. Given that $\mathbf{H}\mathbf{P}$ and $\mathbf{H}\mathbf{P}^{\perp}$ are independent to each other, and considering that $\mathbf{H}\mathbf{P}^{\perp}$ has the same distribution as $\tilde{\mathbf{H}}\mathbf{P}^{\perp}$, where $\tilde{\mathbf{H}}$ denotes an independent copy of $\mathbf{H}$, we can conclude that the solution to the optimization problem above follows the same distribution of the solution to the following

$$\min_{\boldsymbol{\beta}_{S^{\perp}} \in \mathbf{P}^{\perp}\mathcal{S}_{\boldsymbol{\beta}}} \max_{v \in \mathbb{R}^n} \min_{\substack{\boldsymbol{\beta}_S \in \mathbf{P}\mathcal{S}_{\boldsymbol{\beta}} \\ \mathbf{u} \in \mathcal{S}_{\mathbf{u}}}} \left(\frac{1}{n}\mathbf{1}^T\rho(\mathbf{u}) - \frac{1}{n}\mathbf{Y}^T\mathbf{u} + \frac{\lambda}{2n}\mathbf{u}^{\top}\mathbf{u} + \frac{1}{\sqrt{n}}v^T(\mathbf{u} - \mathbf{H}\boldsymbol{\beta}_S) - \frac{1}{\sqrt{n}}v^T\tilde{\mathbf{H}}\boldsymbol{\beta}_{S^{\perp}}\right)$$

We are ready to define the PO problem as

$$\text{PO:} \quad \min_{\boldsymbol{\beta}_{S^{\perp}} \in \mathbf{P}^{\perp}\mathcal{S}_{\boldsymbol{\beta}}} \max_{v \in \mathcal{S}_v} \left\{\frac{-1}{\sqrt{n}}v^{\top}\tilde{\mathbf{H}}\boldsymbol{\beta}_{S^{\perp}} + \psi(\boldsymbol{\beta}_{S^{\perp}}, v)\right\}, \tag{22}$$

where $\psi(\boldsymbol{\beta}_{S^\perp}, \boldsymbol{v})$ is defined as

$$\psi(\boldsymbol{\beta}_{S^\perp}, \boldsymbol{v}) := \min_{\substack{\boldsymbol{\beta}_S \in \mathbf{P}\mathcal{S}_{\boldsymbol{\beta}} \\ \mathbf{u} \in \mathcal{S}_{\mathbf{u}}}} \left( \frac{1}{n}\mathbf{1}^T \rho(\mathbf{u}) - \frac{1}{n}\mathbf{Y}^T \mathbf{u} + \frac{\lambda}{2n}\mathbf{u}^\top \mathbf{u} + \frac{1}{\sqrt{n}}\boldsymbol{v}^T (\mathbf{u} - \mathbf{H}\boldsymbol{\beta}_S) \right).$$

It is easy to see the objective function in (22) is jointly convex with respect to $\left( \boldsymbol{\beta}_S, \boldsymbol{\beta}_S^\perp, \boldsymbol{u} \right)$, and is concave with respect to $\boldsymbol{v}$.

Furthermore, we define the AO problem as follows

$$\text{AO:} \quad \phi(\mathbf{g}, \mathbf{h}) = \min_{\boldsymbol{\beta}_{S^\perp} \in \mathbf{P}^\perp \mathcal{S}_{\boldsymbol{\beta}}} \max_{\boldsymbol{v} \in \mathcal{S}_{\boldsymbol{v}}} \left\{ -\frac{1}{\sqrt{n}} \left( \boldsymbol{v}^T \mathbf{h} \|\boldsymbol{\beta}_{S^\perp}\| + \|\boldsymbol{v}\| \mathbf{g}^T \boldsymbol{\beta}_{S^\perp} \right) + \psi(\boldsymbol{\beta}_{S^\perp}, \boldsymbol{v}) \right\}, \tag{23}$$

where $\mathbf{h} \in \mathbb{R}^{n+M}$ and $\mathbf{g} \in \mathbb{R}^p$ have i.i.d. standard normal entries and are independent with $\mathbf{H}$.

### C.2. Analyzing the auxiliary optimization

Since the objective function in Eq.(23) is concave with respect to $\boldsymbol{v}$, and the objective function in the definition of $\psi(\boldsymbol{\beta}_{S^\perp}, \boldsymbol{v})$ is jointly convex with respect to $(\boldsymbol{\beta}_S, \boldsymbol{u})$, and all the feasible sets of $\boldsymbol{\beta}_S, \boldsymbol{v}$ and $\mathbf{u}$ are compact and convex, we apply Sion's minimax theorem to rewrite (23) by flipping the $\min_{\boldsymbol{\beta}_S, \mathbf{u}}$ and $\max_{\boldsymbol{v}}$:

$$\min_{\substack{\boldsymbol{\beta}_S \in \mathbf{P}\mathcal{S}_{\boldsymbol{\beta}}, \boldsymbol{\beta}_{S^\perp} \in \mathbf{P}^\perp \mathcal{S}_{\boldsymbol{\beta}} \\ \mathbf{u} \in \mathcal{S}_{\mathbf{u}}}} \max_{\boldsymbol{v} \in \mathcal{S}_{\boldsymbol{v}}} \left( \frac{1}{n}\mathbf{1}^T \rho(\mathbf{u}) - \frac{1}{n}\mathbf{Y}^T \mathbf{u} + \frac{\lambda}{2n}\mathbf{u}^\top \mathbf{u} + \frac{1}{\sqrt{n}}\boldsymbol{v}^T (\mathbf{u} - \mathbf{H}\boldsymbol{\beta}_S) \right.$$

$$\left. -\frac{1}{\sqrt{n}} \left( \boldsymbol{v}^T \mathbf{h} \|\boldsymbol{\beta}_{S^\perp}\| + \|\boldsymbol{v}\| \mathbf{g}^T \boldsymbol{\beta}_{S^\perp} \right) \right). \tag{24}$$

Ideally, we would like to solve the optimization in (24) with respect to the directions of the vectors while fixing the norms of the vectors, so that we get a scalar optimization. We first perform the maximization with respect to the direction of $\boldsymbol{v}$. The maximization with respect to $\boldsymbol{v}$ in (24) can be rewritten as

$$\max_{\boldsymbol{v} \in \mathcal{S}_{\boldsymbol{v}}} \frac{1}{\sqrt{n}} \|\boldsymbol{v}\| \mathbf{g}^T \boldsymbol{\beta}_{S^\perp} + \frac{1}{\sqrt{n}}\boldsymbol{v}^T (\mathbf{u} - \mathbf{H}\boldsymbol{\beta}_S - \|\boldsymbol{\beta}_{S^\perp}\|\mathbf{h}).$$

For this maximization, we choose the direction of $\boldsymbol{v}$ to be the same as the direction of the vector that it is multiplied to and introduce a variable $r := \|\boldsymbol{v}\|$ to denote the length of $\boldsymbol{v}$. Additionally, the feasible set of $r$ is $(0, V]$ where $V$ comes from the compact set $\mathcal{S}_v$. The maximization then becomes

$$\max_{r \in (0, V]} \frac{r}{\sqrt{n}} \left( \mathbf{g}^T \boldsymbol{\beta}_{S^\perp} + \|\mathbf{u} - \mathbf{H}\boldsymbol{\beta}_S - \|\boldsymbol{\beta}_{S^\perp}\|\mathbf{h}\| \right)$$

The AO is now given by

$$\min_{\substack{\boldsymbol{\beta}_S \in \mathbf{P}\mathcal{S}_{\boldsymbol{\beta}}, \boldsymbol{\beta}_{S^\perp} \in \mathbf{P}^\perp \mathcal{S}_{\boldsymbol{\beta}} \\ \mathbf{u} \in \mathbb{R}^n}} \max_{r \in (0, V]} \left\{ \frac{1}{n}\mathbf{1}^T \rho(\mathbf{u}) - \frac{1}{n}\mathbf{Y}^T \mathbf{u} + \frac{\lambda}{2n}\mathbf{u}^\top \mathbf{u} \right.$$

$$\left. + \frac{r}{\sqrt{n}} \left( \mathbf{g}^T \boldsymbol{\beta}_{S^\perp} + \|\mathbf{u} - \mathbf{H}\boldsymbol{\beta}_S - \|\boldsymbol{\beta}_{S^\perp}\|\mathbf{h}\| \right) \right\} \tag{25}$$

For the subsequent analysis, we need an explicit expression for the projection matrix $\mathbf{P}$. It is worth mentioning that in the literature, the projection matrix is often equal to $\frac{\boldsymbol{\beta}_* \boldsymbol{\beta}_*^\top}{\|\boldsymbol{\beta}_*\|^2}$, which has rank 1.

For any candidate $\boldsymbol{\beta}$ in (25), since the length and the direction of $\mathbf{P}\boldsymbol{\beta}$ and those of $\mathbf{P}^\perp\boldsymbol{\beta}$ are independent with each other, this decoupling allows us to optimize over their lengths and orientations separately. To see how this works, we decompose $\boldsymbol{\beta}$ as follows:

$$\boldsymbol{\beta} = \mathbf{P}\boldsymbol{\beta} + \mathbf{P}^{\perp}\boldsymbol{\beta}$$
$$= (\frac{\boldsymbol{\beta}_*^T \boldsymbol{\beta}}{\|\boldsymbol{\beta}_*\|^2})\boldsymbol{\beta}_* + \|\mathbf{P}^{\perp}\boldsymbol{\beta}\| \cdot \text{direction}(\mathbf{P}^{\perp}\boldsymbol{\beta}). \tag{26}$$

For the estimator $\widehat{\boldsymbol{\beta}}$, the two scalar quantities $\frac{\boldsymbol{\beta}_*^T \boldsymbol{\beta}}{\|\boldsymbol{\beta}_*\|^2}$, $\|\mathbf{P}^{\perp}\widehat{\boldsymbol{\beta}}\|$ will be tracked in the asymptotics with a system of equations. Using the above decomposition, we interpret $\boldsymbol{\beta}_*$ as the true signal, and $\mathbf{P}^{\perp}\widehat{\boldsymbol{\beta}}$ as the noise, which will be approximated by a standard Gaussian vector. The essential of the application of CGMT is to characterize the asymptotic behaviors of the scalar quantities aforementioned.

To be concrete, we introduce the scalars $\alpha := \frac{\boldsymbol{\beta}_*^T \boldsymbol{\beta}}{\|\boldsymbol{\beta}_*\|\kappa}$, $\sigma := \|\mathbf{P}^{\perp}\boldsymbol{\beta}\|$ and let $\boldsymbol{\theta}$ be the direction of $\mathbf{P}^{\perp}\boldsymbol{\beta}$. In the following, we drop the feasible sets to ease the notation whenever there is no ambiguity. The AO problem is now written as

$$\min_{\substack{\sigma > 0 \\ \mathbf{u} \in \mathbb{R}^n \\ \alpha \in \mathbb{R}}} \min_{\|\boldsymbol{\theta}\|_2 = 1} \max_{r \in (0, V]} \left( \frac{1}{n}\mathbf{1}^T \rho(\mathbf{u}) - \frac{1}{n}\mathbf{Y}^T \mathbf{u} + \frac{\lambda}{2n}\mathbf{u}^{\top}\mathbf{u} \right.$$
$$\left. + \frac{r}{\sqrt{n}} \left( \sigma \mathbf{g}^T \boldsymbol{\theta} + \|\mathbf{u} - \kappa\alpha\boldsymbol{q} - \sigma\mathbf{h}\| \right) \right),$$

where $\boldsymbol{q} := \mathbf{H}\boldsymbol{\beta}_*/\|\boldsymbol{\beta}_*\|$. Notice that $\boldsymbol{q}$ have i.i.d. standard normal entries (recall that $\mathbf{H}$ has i.i.d. standard normal entries). In the next step, we exchange the order of the $\min_{\|\boldsymbol{\theta}\|=1}$ and $\max_{r \in [0, V]}$ in the above problem. This flipping is based on lemma C.5.

The AO problem can be reformulated as

$$\min_{\substack{\sigma > 0 \\ \mathbf{u} \in \mathbb{R}^n \\ \alpha \in \mathbb{R}}} \max_{r \in (0, V]} \min_{\|\boldsymbol{\theta}\|_2 = 1} \left( \frac{1}{n}\mathbf{1}^T \rho(\mathbf{u}) - \frac{1}{n}\mathbf{Y}^T \mathbf{u} + \frac{\lambda}{2n}\mathbf{u}^{\top}\mathbf{u} \right.$$
$$\left. + \frac{r}{\sqrt{n}} \left( \sigma \mathbf{g}^T \boldsymbol{\theta} + \|\mathbf{u} - \kappa\alpha\boldsymbol{q} - \sigma\mathbf{h}\| \right) \right), \tag{27}$$

Optimizing this problem with respect to the direction of $\boldsymbol{\theta}$ ($\boldsymbol{\theta}_{optimal} = \text{direction}(\mathbf{P}^{\perp}\mathbf{g})$) yields the following

$$\min_{\substack{\sigma > 0 \\ \mathbf{u} \in \mathbb{R}^n \\ \alpha \in \mathbb{R}}} \max_{r \in (0, V]} \left( \frac{1}{n}\mathbf{1}^T \rho(\mathbf{u}) - \frac{1}{n}\mathbf{Y}^T \mathbf{u} + \frac{\lambda}{2n}\mathbf{u}^{\top}\mathbf{u} - \frac{r\sigma}{\sqrt{n}}\|\mathbf{P}^{\perp}\mathbf{g}\| \right.$$
$$\left. + r\frac{1}{\sqrt{n}}\|\mathbf{u} - \kappa\alpha\boldsymbol{q} - \sigma\mathbf{h}\| \right).$$

Next, we use the identity that $\|\mathbf{a}\| = \min_{\tilde{\nu} > 0}\left(\frac{1}{2\tilde{\nu}}\|\mathbf{a}\|^2 + \frac{\tilde{\nu}}{2}\right)$, with optima $\widehat{\nu} = \|\mathbf{a}\|$, to replace the norm in the last display by a squared term:

$$\min_{\substack{\sigma > 0 \\ \mathbf{u} \in \mathbb{R}^n \\ \alpha \in \mathbb{R}}} \max_{r \in (0, V]} \min_{\tilde{\nu} > 0} \left( \frac{1}{n}\mathbf{1}^T \rho(\mathbf{u}) - \frac{1}{n}\mathbf{Y}^T \mathbf{u} + \frac{\lambda}{2n}\mathbf{u}^{\top}\mathbf{u} - \frac{r\sigma}{\sqrt{n}}\|\mathbf{P}^{\perp}\mathbf{g}\| \right.$$
$$\left. + \frac{r\tilde{\nu}}{2} + \frac{r}{2\tilde{\nu}}\left\| \frac{1}{\sqrt{n}}\mathbf{u} - \frac{1}{\sqrt{n}}\kappa\alpha\boldsymbol{q} - \frac{1}{\sqrt{n}}\sigma\mathbf{h} \right\|^2 \right) \tag{28}$$

We shall show the above objective function is jointly convex in $(\mathbf{u}, \alpha, \sigma, \tilde{\nu})$ and concave in $r$. The concavity is easy since the objective function is linear in $r$. To show the joint convexity, we first note that the function $h_1(\tilde{\boldsymbol{\theta}}) := 1 + \left\| \frac{1}{\sqrt{n}}\mathbf{u} - \frac{1}{\sqrt{n}}\kappa\alpha\boldsymbol{q} - \frac{1}{\sqrt{n}}\sigma\mathbf{h} \right\|^2$ is jointly convex in $\tilde{\boldsymbol{\theta}} := (\mathbf{u}, \alpha, \sigma)$ since $h_1$ is quadratic over some linear functions. We then

note that the perspective function of $h_1(\tilde{\boldsymbol{\theta}})$ is

$$
\begin{aligned}
g_1(\tilde{\boldsymbol{\theta}}, \tilde{\nu}) &:= \tilde{\nu} + \frac{1}{\tilde{\nu}} \left\| \frac{1}{\sqrt{n}}\mathbf{u} - \frac{1}{\sqrt{n}}\kappa\alpha\boldsymbol{q} - \frac{1}{\sqrt{n}}\sigma\mathbf{h} \right\|^2 \\
&= \tilde{\nu}\left( 1 + \frac{1}{\tilde{\nu}^2} \left\| \frac{1}{\sqrt{n}}\mathbf{u} - \frac{1}{\sqrt{n}}\kappa\alpha\boldsymbol{q} - \frac{1}{\sqrt{n}}\sigma\mathbf{h} \right\|^2 \right) \\
&= \tilde{\nu}h_1\left(\frac{\tilde{\boldsymbol{\theta}}}{\tilde{\nu}}\right),
\end{aligned}
$$

which is jointly convex in $(\tilde{\boldsymbol{\theta}}, \tilde{\nu})$ since $h_1$ is convex in $\tilde{\boldsymbol{\theta}}$. The joint convexity of the objective function follows from the joint convexity of $g_1(\tilde{\boldsymbol{\theta}}, \tilde{\nu})$ and the convexity of $\rho(\cdot)$. To perform minization over $\mathbf{u}$, we use Sion's minimax theorem to swap the order of minimization and maximization, arrive at

$$
\min_{\substack{\sigma>0,\tilde{\nu}>0 \\ \alpha\in\mathbb{R}}} \max_{r\in(0,V]} \min_{\mathbf{u}\in\mathbb{R}^n} \left( \frac{1}{n}\mathbf{1}^T\rho(\mathbf{u}) - \frac{1}{n}\mathbf{Y}^T\mathbf{u} + \frac{\lambda}{2n}\mathbf{u}^\top\mathbf{u} - \frac{r\sigma}{\sqrt{n}}\left\|\mathbf{P}^\perp\boldsymbol{g}\right\| \right.
$$

$$
\left. + \frac{r\tilde{\nu}}{2} + \frac{r}{2\tilde{\nu}}\left\| \frac{1}{\sqrt{n}}\mathbf{u} - \frac{1}{\sqrt{n}}\kappa\alpha\boldsymbol{q} - \frac{1}{\sqrt{n}}\sigma\mathbf{h} \right\|^2 \right)
$$

**Minimization over u:**  We now focus on the optimization over $\mathbf{u}\in\mathbb{R}^n$. Specifically, we analyze the following problem:

$$
\min_{\mathbf{u}\in\mathbb{R}^n} \left( \frac{1}{n}\mathbf{1}^T\rho(\mathbf{u}) - \frac{1}{n}\mathbf{Y}^T\mathbf{u} + \frac{\lambda}{2n}\mathbf{u}^\top\mathbf{u} + \frac{r}{2\tilde{\nu}}\left\| \frac{1}{\sqrt{n}}\mathbf{u} - \frac{1}{\sqrt{n}}\kappa\alpha\boldsymbol{q} - \frac{1}{\sqrt{n}}\sigma\mathbf{h} \right\|^2 \right) \tag{29}
$$

For the terms involving $\mathbf{Y}$ and $\mathbf{u}$, we use the following completion of squares:

$$
\begin{aligned}
&-\frac{1}{n}\mathbf{Y}^T\mathbf{u} + \frac{\lambda}{2n}\mathbf{u}^\top\mathbf{u} + \frac{r}{2\tilde{\nu}}\left\| \frac{1}{\sqrt{n}}\mathbf{u} - \frac{1}{\sqrt{n}}\kappa\alpha\boldsymbol{q} - \frac{1}{\sqrt{n}}\sigma\mathbf{h} \right\|^2 \\
&= \frac{\lambda+\frac{r}{\tilde{\nu}}}{2n}\left\| \mathbf{u} - \frac{\mathbf{Y}+\frac{r}{\tilde{\nu}}(\kappa\alpha\boldsymbol{q}+\sigma\mathbf{h})}{\lambda+\frac{r}{\tilde{\nu}}} \right\|^2 + \frac{r}{2\tilde{\nu}n}\|\kappa\alpha\boldsymbol{q}+\sigma\mathbf{h}\|^2 - \frac{1}{2n\left(\lambda+\frac{r}{\tilde{\nu}}\right)}\left\| \mathbf{Y}+\frac{r}{\tilde{\nu}}(\kappa\alpha\boldsymbol{q}+\sigma\mathbf{h}) \right\|^2 .
\end{aligned} \tag{30}
$$

Eq.(28) can be rewritten as

$$
\begin{aligned}
\min_{\substack{\sigma>0,\tilde{\nu}>0 \\ \alpha\in\mathbb{R}}} \max_{r\in(0,V]} \min_{\mathbf{u}\in\mathbb{R}^n} &\left( \frac{1}{n}\mathbf{1}^\top\rho(\mathbf{u}) + \frac{\lambda+\frac{r}{\tilde{\nu}}}{2n}\left\| \mathbf{u} - \frac{\mathbf{Y}+\frac{r}{\tilde{\nu}}(\kappa\alpha\boldsymbol{q}+\sigma\mathbf{h})}{\lambda+\frac{r}{\tilde{\nu}}} \right\|^2 \right. \\
&+ \frac{r}{2\tilde{\nu}n}\|\kappa\alpha\boldsymbol{q}+\sigma\mathbf{h}\|^2 - \frac{1}{2n\left(\lambda+\frac{r}{\tilde{\nu}}\right)}\left\| \mathbf{Y}+\frac{r}{\tilde{\nu}}(\kappa\alpha\boldsymbol{q}+\sigma\mathbf{h}) \right\|^2 \\
&\left. - \frac{\sigma r}{\sqrt{n}}\left\|\mathbf{P}^\perp\boldsymbol{g}\right\| + \frac{r\tilde{\nu}}{2} \right).
\end{aligned}
$$

Now we can perform the minimization over $\mathbf{u}$. Based on the definition of the Moreau envelope, we can express the minimization over $\mathbf{u}$ as

$$
\begin{aligned}
\min_{\mathbf{u}\in\mathbb{R}^n} \frac{1}{n}\mathbf{1}^\top\rho(\mathbf{u}) + \frac{\lambda+\frac{r}{\tilde{\nu}}}{2n}\left\| \mathbf{u} - \frac{\mathbf{Y}+\frac{r}{\tilde{\nu}}(\kappa\alpha\boldsymbol{q}+\sigma\mathbf{h})}{\lambda+\frac{r}{\tilde{\nu}}} \right\|^2 \\
= \frac{1}{n}M_{\rho(\cdot)}\left( \frac{\mathbf{Y}+\frac{r}{\tilde{\nu}}(\kappa\alpha\boldsymbol{q}+\sigma\mathbf{h})}{\lambda+\frac{r}{\tilde{\nu}}}, 1/(\lambda+\frac{r}{\tilde{\nu}}) \right),
\end{aligned}
$$

As a result, Eq.(28) can be simplified as

$$
\min_{\substack{\sigma>0,\tilde{\nu}>0 \\ \alpha\in\mathbb{R}}} \max_{r>0} \quad \mathcal{R}_n(\sigma, r, \tilde{\nu}, \alpha), \tag{31}
$$

where

$$\mathcal{R}_n(\sigma, r, \tilde{\nu}, \alpha) := \frac{1}{n} M_{\rho(\cdot)} \left( \frac{\mathbf{Y} + \frac{r}{\tilde{\nu}}(\kappa\alpha\mathbf{q} + \sigma\mathbf{h})}{\lambda + \frac{r}{\tilde{\nu}}}, 1/(\lambda + \frac{r}{\tilde{\nu}}) \right)$$
$$+ \frac{r}{2\tilde{\nu}n} \|\kappa\alpha\mathbf{q} + \sigma\mathbf{h}\|^2 - \frac{1}{2n\left(\lambda + \frac{r}{\tilde{\nu}}\right)} \left\| \mathbf{Y} + \frac{r}{\tilde{\nu}}(\kappa\alpha\mathbf{q} + \sigma\mathbf{h}) \right\|^2$$
$$- \frac{\sigma r}{\sqrt{n}} \left\| \mathbf{P}^\perp \mathbf{g} \right\| + \frac{r\tilde{\nu}}{2} \right).$$

Since the partial minimization of a convex function over a convex feasible set preserves the convexity, the objective function $\mathcal{R}_n$ is jointly convex in $(\sigma, \tilde{\nu}, \alpha)$ for any $r$. By Danskin's theorem (Danskin, 1966), $\mathcal{R}_n$ is concave in $r$ for any $(\sigma, \tilde{\nu}, \alpha)$. In the following, we aim to find the limit of $\mathcal{R}_n$ and then show that the solution to $\mathcal{R}_n$ converges to the solution to the limit.

**Limit of $\mathcal{R}_n(\sigma, r, \tilde{\nu}, \alpha)$**  Fix any $(\sigma, r, \tilde{\nu}, \alpha)$. Using SLLN, we have as $n \to \infty$,

$$\frac{1}{n} M_{\rho(\cdot)} \left( \frac{\mathbf{Y} + \frac{r}{\tilde{\nu}}(\kappa\alpha\mathbf{q} + \sigma\mathbf{h})}{\lambda + \frac{r}{\tilde{\nu}}}, 1/(\lambda + \frac{r}{\tilde{\nu}}) \right) \xrightarrow{a.s} \mathbb{E} \left( M_{\rho(\cdot)} \left( \frac{r/\tilde{\nu}}{\lambda + r/\tilde{\nu}} \kappa\alpha Z_1 + \frac{r/\tilde{\nu}}{\lambda + r/\tilde{\nu}} \sigma Z_2 + \frac{Ber(\rho'(\kappa Z_1))}{\lambda + r/\tilde{\nu}}, \frac{1}{\lambda + r/\tilde{\nu}} \right) \right)$$

(32)

We can further simplify RHS according to Bernoulli distribution if we consider logistic regression:

$$\mathbb{E} \left( M_{\rho(\cdot)} \left( \frac{r/\tilde{\nu}}{\lambda + r/\tilde{\nu}} \kappa\alpha Z_1 + \frac{r/\tilde{\nu}}{\lambda + r/\tilde{\nu}} \sigma Z_2 + \frac{Ber(\rho'(\kappa Z_1))}{\lambda + r/\tilde{\nu}}, \frac{1}{\lambda + r/\tilde{\nu}} \right) \right)$$
$$= \mathbb{E} \left( \rho'(-\kappa Z_1) M_{\rho(\cdot)} \left( \frac{r/\tilde{\nu}}{\lambda + r/\tilde{\nu}} \kappa\alpha Z_1 + \frac{r/\tilde{\nu}}{\lambda + r/\tilde{\nu}} \sigma Z_2, \frac{1}{\lambda + r/\tilde{\nu}} \right) \right)$$
$$+ \mathbb{E} \left( \rho'(\kappa Z_1) M_{\rho(\cdot)} \left( \frac{r/\tilde{\nu}}{\lambda + r/\tilde{\nu}} \kappa\alpha Z_1 + \frac{r/\tilde{\nu}}{\lambda + r/\tilde{\nu}} \sigma Z_2 + \frac{1}{\lambda + r/\tilde{\nu}}, \frac{1}{\lambda + r/\tilde{\nu}} \right) \right)$$

Recall that $\mathbf{Y} = Ber(\rho'(\mathbf{H}\boldsymbol{\beta}_*)) = Ber(\rho'(\kappa\boldsymbol{q}))$, we have

$$\frac{1}{n} \mathbf{Y}^\top \boldsymbol{q} = \frac{1}{n} \sum_{i=1}^n y_i q_i = \frac{1}{n} \sum_{i=1}^n Ber \left( \rho'(\kappa q_i) \right) q_i \xrightarrow{a.s} \mathbb{E}_Z [Z \cdot \rho'(\kappa Z)] = \kappa \mathbb{E}_Z [\rho''(\kappa Z)],$$

$$\text{and} \quad \frac{1}{n} \|\mathbf{Y}\|^2 = \frac{1}{n} \sum_{i=1}^n y_i^2 \xLeftrightarrow[n\to\infty]{\text{SLLN}} \mathbb{E} [y_i^2] = \mathbb{E} [y_i] = \mathbb{E}_Z [\rho'(\kappa Z)] = \frac{1}{2},$$

which implies that

$$\frac{r}{2\tilde{\nu}n} \|\kappa\alpha\mathbf{q} + \sigma\mathbf{h}\|^2 - \frac{1}{2n\left(\lambda + \frac{r}{\tilde{\nu}}\right)} \left\| \mathbf{Y} + \frac{r}{\tilde{\nu}}(\kappa\alpha\mathbf{q} + \sigma\mathbf{h}) \right\|^2$$
$$\xrightarrow{a.s} \frac{\lambda r/\tilde{\nu}}{2(\lambda + r/\tilde{\nu})} (\kappa^2\alpha^2 + \sigma^2) - \frac{1}{4(\lambda + r/\tilde{\nu})} - \frac{r/\tilde{\nu}}{\lambda + r/\tilde{\nu}} \kappa^2 \alpha \mathbb{E}_Z [\rho''(\kappa Z)]$$

For the term $\frac{\sigma r}{\sqrt{n}} \left\| \mathbf{P}^\perp \mathbf{g} \right\|$, since $\mathbf{g} \in \mathbb{R}^p$ has i.i.d. standard normal entries, we can approximate $\frac{\sigma r}{\sqrt{n}} \left\| \mathbf{P}^\perp \mathbf{g} \right\|$ with $\frac{\sigma r}{\sqrt{\delta}}$ by SLLN for any fixed $(\sigma, r)$, where $\delta := \frac{n}{p}$ is the oversampling ratio.

Putting all these together, the point-wise limit of the objective function $\mathcal{R}_n(\sigma, r, \tilde{\nu}, \alpha)$, denoted by $\mathcal{R}(\sigma, r, \tilde{\nu}, \alpha)$, can be expressed as follows:

$$
\begin{aligned}
\mathcal{R}&(\sigma, r, \tilde{\nu}, \alpha) \\
&:= \lim_{n\to\infty} \mathcal{R}_n(\sigma, r, v, \alpha) \\
&= \left\{ -\frac{r\sigma}{\sqrt{\delta}} + \frac{r\tilde{\nu}}{2} - \frac{1}{4(\lambda + r/\tilde{\nu})} - \frac{r/\tilde{\nu}}{\lambda + r/\tilde{\nu}}\kappa^2\alpha\mathbb{E}_Z\left[\rho''(\kappa Z)\right] + \frac{\lambda r/\tilde{\nu}}{2(\lambda + r/\tilde{\nu})}(\kappa^2\alpha^2 + \sigma^2) \right. \\
&\quad + \mathbb{E}\left(\rho'(-\kappa Z_1)M_{\rho(\cdot)}\left(\frac{r/\tilde{\nu}}{\lambda + r/\tilde{\nu}}\kappa\alpha Z_1 + \frac{r/\tilde{\nu}}{\lambda + r/\tilde{\nu}}\sigma Z_2, \frac{1}{\lambda + r/\tilde{\nu}}\right)\right) \\
&\quad \left. + \mathbb{E}\left(\rho'(\kappa Z_1)M_{\rho(\cdot)}\left(\frac{r/\tilde{\nu}}{\lambda + r/\tilde{\nu}}\kappa\alpha Z_1 + \frac{r/\tilde{\nu}}{\lambda + r/\tilde{\nu}}\sigma Z_2 + \frac{1}{\lambda + r/\tilde{\nu}}, \frac{1}{\lambda + r/\tilde{\nu}}\right)\right) \right\}.
\end{aligned}
$$

Since taking point-wise limit preserves the convexity and the concavity, we know that $\mathcal{R}(\sigma, r, \tilde{\nu}, \alpha)$ is concave in $r$ and jointly convex in $(\sigma, \tilde{\nu}, \alpha)$.

Define an scalar optimization based on $\mathcal{R}(\sigma, r, \tilde{\nu}, \alpha)$

$$
\bar{\phi} := \min_{\substack{\sigma>0, \tilde{\nu}>0 \\ \alpha\in\mathbb{R}}} \max_{r\in(0,V]} \mathcal{R}(\sigma, r, \tilde{\nu}, \alpha), \tag{33}
$$

and let $(\sigma_*, r_*, \tilde{\nu}_*, \alpha_*)$ be the solution to the optimization in (33). We will show below that optima of (31) will converge to $(\sigma_*, r_*, \tilde{\nu}_*, \alpha_*)$.

**Converge of the optima** In order to justify the convergence of the optima of $\mathcal{R}_n$, we should show that the domain for $(\sigma, r, \tilde{\nu}, \alpha)$ is uniformly bounded in the following sense:

$$
\begin{aligned}
\sigma &= \left\| \mathbf{P}^\perp\boldsymbol{\beta} \right\| \le \|\boldsymbol{\beta}\| \le c_1, \\
|\alpha| &= \left| \frac{\boldsymbol{\beta}_*^T\boldsymbol{\beta}}{\|\boldsymbol{\beta}_*\|\kappa} \right| \le \frac{\|\boldsymbol{\beta}\|}{\kappa} \le c_1/\kappa, \\
r &= \|\boldsymbol{v}\| \le V
\end{aligned} \tag{34}
$$

The first two inequalities in (34) follow from the fact that the feasible set of $\boldsymbol{\beta}$ is a closed ball centered at the origin and has a constant radius, as proved in part 1 of Theorem 3.1. The last inequality regarding $r$ follows from the fact that the feasible set for the variable $\boldsymbol{v}$ is a closed ball with a constant radius. For the scalar variable $\tilde{\nu}$, we recall its definition in

$$
\frac{1}{\sqrt{n}}\left\| \mathbf{u} - \kappa\alpha\boldsymbol{q} - \sigma\mathbf{h} \right\| = \min_{\tilde{\nu}>0}\left\{ \frac{\tilde{\nu}}{2} + \frac{1}{2\tilde{\nu}}\left\| \frac{1}{\sqrt{n}}\mathbf{u} - \frac{1}{\sqrt{n}}\kappa\alpha\boldsymbol{q} - \frac{1}{\sqrt{n}}\sigma\mathbf{h} \right\|^2 \right\}, \tag{35}
$$

where the optimal $\widehat{\nu}$ is equal to $\left\| \frac{1}{\sqrt{n}}\mathbf{u} - \frac{1}{\sqrt{n}}\kappa\alpha\boldsymbol{q} - \frac{1}{\sqrt{n}}\sigma\mathbf{h} \right\|$. Therefore, we can, without changing the formulation, restrict the feasible set of $\tilde{\nu}$ to be an interval with the right end larger than $\widehat{\nu}$. Since we have already shown $\|\mathbf{u}\| \le C\sqrt{n}$ for large enough sample size $n$ in (20), by the triangle inequality, it suffices to bound $\frac{1}{\sqrt{n}}\|\kappa\alpha\boldsymbol{q} + +\sigma\mathbf{h}\|$. Recall $\boldsymbol{q}$ and $\mathbf{h}$ are random vectors with independent standard Gaussian random variable as entries. By lemma C.3 and (34), we have

$$
\begin{aligned}
\mathbb{P}(\|\kappa\alpha\boldsymbol{q}\| > 2c_1\sqrt{n}) &\le \exp(-n/2), \\
P(\|\sigma\mathbf{h}\| > 2c_1\sqrt{n}) &\le \exp(-n/2).
\end{aligned}
$$

By union bound and Borel Cantelli lemma, we have

$$
\mathbb{P}\left( \left\{ \frac{1}{\sqrt{n}}\|\kappa\alpha\boldsymbol{q} + \sigma\mathbf{h}\| > 4c_1 \right\} \text{ happens infinitely many times} \right) = 0 \tag{36}
$$

Therefore, we can constrain the feasible set of $\tilde{\nu}$ to be bounded.

Up to this point, we have shown that the objective function in (28) converges point-wise to the objective function $\mathcal{R}(\sigma, r, \tilde{\nu}, \alpha)$. Furthermore, we've established that both objective functions are joint convex with respect to $(\sigma, \tilde{\nu}, \alpha)$ and concave with

respect to $r$, within a compact domain for these parameters. Drawing on similar reasoning as presented in the proof of Dai et al. (2023b, Lemma A.1) and in Javanmard & Soltanolkotabi (2022, Appendix B.3.3), which in turn make use of arguments from Thrampoulidis et al. (2018, Lemma A.5), we can conclude that the optimal solutions in (28), denoted as $(\hat{\sigma}, \hat{r}, \hat{\tilde{\nu}}, \hat{\alpha})$, will uniformly converge to the optimal solution $(\sigma_*, r_*, \tilde{\nu}_*, \alpha_*)$ in (33).

### C.3. Finding the optimality condition of the limiting scalar optimization

We characterize the solution to the optimization in (33). To facilitate the analysis in the following, we reparametrize $\tilde{\nu}$ by introducing $v = 1/\tilde{\nu}$. The original scalar optimization become:

$$\min_{\substack{\sigma>0,v>0 \\ \alpha\in\mathbb{R}}} \max_{r\in(0,V]} \quad \left\{ -\frac{r\sigma}{\sqrt{\delta}} + \frac{r}{2v} - \frac{1}{4(\lambda+r/\tilde{\nu})} - \frac{rv}{\lambda+rv}\kappa^2\alpha\mathbb{E}_Z\left[\rho''(\kappa Z)\right] + \frac{\lambda rv}{2(\lambda+rv)}(\kappa^2\alpha^2 + \sigma^2) \right.$$

$$+ \mathbb{E}\left(\rho'(-\kappa Z_1)M_{\rho(\cdot)}\left(\frac{rv}{\lambda+rv}\kappa\alpha Z_1 + \frac{rv}{\lambda+rv}\sigma Z_2, \frac{1}{\lambda+rv}\right)\right) \quad (37)$$

$$\left. + \mathbb{E}\left(\rho'(\kappa Z_1)M_{\rho(\cdot)}\left(\frac{rv}{\lambda+rv}\kappa\alpha Z_1 + \frac{rv}{\lambda+rv}\sigma Z_2 + \frac{1}{\lambda+rv}, \frac{1}{\lambda+rv}\right)\right) \right\}.$$

Let $C(r, v, \sigma, \alpha)$ denote the objective function in (37), we aim to analyze the optima of $C(\cdot)$, i.e., $(r^*, v^*, \sigma^*, \alpha^*)$. Since the objective function is smooth, they should satisfy the first order optimality condition, i.e., $\nabla C = 0$. We will show that $\nabla C = 0$ will reduce to our system of nonlinear equations in (5). In the subsequent section, we will frequently use Lemma C.8 to compute the derivative of Moreau envelope and dominated convergence theorem to interchange derivatives and expectation without further statement.

Define $\tilde{r} = \frac{1}{r+\lambda v}$, and let $Q_1(\tilde{r}) := (1-\lambda\tilde{r})[\kappa\alpha Z_1 + \sigma Z_2] + \tilde{r}, Q_2(\tilde{r}) := (1-\lambda\tilde{r})[\kappa\alpha Z_1 + \sigma Z_2]$. The we can rewrite last two functions in (37):

$$F_1(Q_2(\tilde{r}), \tilde{r}) := \mathbb{E}\left(\rho'(-\kappa Z_1)M_{\rho(\cdot)}\left(Q_2(\tilde{r}), \tilde{r}\right)\right)$$
$$F_2(Q_1(\tilde{r}), \tilde{r}) = \mathbb{E}\left(\rho'(\kappa Z_1)M_{\rho(\cdot)}\left(Q_1(\tilde{r}), \tilde{r}\right)\right)$$

Let $F(\alpha, \sigma, \tilde{r}) := F_1(Q_2(\tilde{r}), \tilde{r}) + F_2(Q_2(\tilde{r}), \tilde{r})$, we first compute the derivative w.r.t $\tilde{r}$, this is important for simplification in later steps.

$$\frac{\partial F_1}{\partial \tilde{r}} = \frac{\partial F_1}{\partial Q_2}\frac{\partial Q_2}{\partial \tilde{r}} + \frac{\partial F_1}{\partial \tilde{r}}\frac{\partial \tilde{r}}{\partial \tilde{r}}$$

$$= \mathbb{E}\left[\rho'(-\kappa Z_1)\frac{1}{\tilde{r}}(Q_2 - \text{Prox}_{\tilde{r}\rho}(Q_2))[-\lambda(\kappa\alpha Z_1 + \sigma Z_2)]\right] - \mathbb{E}\left[\rho'(-\kappa Z_1)\frac{1}{2\tilde{r}^2}(Q_2 - \text{Prox}_{\tilde{r}\rho}(Q_2))^2\right]$$

$$\frac{\partial F_2}{\partial \tilde{r}} = \frac{\partial F_2}{\partial Q_1}\frac{\partial Q_1}{\partial \tilde{r}} + \frac{\partial F_2}{\partial \tilde{r}}\frac{\partial \tilde{r}}{\partial \tilde{r}}$$

$$= \mathbb{E}\left[\rho'(\kappa Z_1)\frac{1}{\tilde{r}}(Q_1 - \text{Prox}_{\tilde{r}\rho}(Q_1))[-\lambda(\kappa\alpha Z_1 + \sigma Z_2) + 1]\right] - \mathbb{E}\left[\rho'(\kappa Z_1)\frac{1}{2\tilde{r}^2}(Q_1 - \text{Prox}_{\tilde{r}\rho}(Q_1))^2\right]$$

To further simplfy, we note that $\text{Prox}_{\tilde{r}\rho}(Q_1) = \text{Prox}_{\tilde{r}\rho}(Q_2 + \tilde{r}) = -\text{Prox}_{\tilde{r}\rho}(Q_2)$ and $Q_1 - \text{Prox}_{\tilde{r}\rho}(Q_1) = Q_2 + \tilde{r} + \text{Prox}_{\tilde{r}\rho}(-Q_2)$, therefore we can simplify the last expectation term as follows.

$$\mathbb{E}\left[\rho'(\kappa Z_1)\frac{1}{2\tilde{r}^2}(Q_1 - \text{Prox}_{\tilde{r}\rho}(Q_1))^2\right] = \mathbb{E}\left[\rho'(\kappa Z_1)\frac{1}{2\tilde{r}^2}(Q_2 + \tilde{r} + \text{Prox}_{\tilde{r}\rho}(-Q_2))^2\right]$$

$$= \mathbb{E}\left[\rho'(\kappa Z_1)\frac{1}{2\tilde{r}^2}(Q_2 + \text{Prox}_{\tilde{r}\rho}(-Q_2))^2\right] + \mathbb{E}\left[\rho'(\kappa Z_1)\frac{1}{2}\right]$$

$$+ \mathbb{E}\left[\rho'(\kappa Z_1)\frac{1}{\tilde{r}}(Q_2 + \text{Prox}_{\tilde{r}\rho}(-Q_2))\right]$$

$$= \mathbb{E}\left[\rho'(-\kappa Z_1)\frac{1}{2\tilde{r}^2}(Q_2 - \text{Prox}_{\tilde{r}\rho}(Q_2))^2\right] + \frac{1}{4} - \mathbb{E}\left[\rho'(-\kappa Z_1)\frac{1}{\tilde{r}}(Q_2 - \text{Prox}_{\tilde{r}\rho}(Q_2))\right]$$

where last equality follows from that $(Z_1, Z_2) \stackrel{d}{=} ((-Z_1, -Z_2))$. For the first expectation term in $\frac{\partial F_2}{\partial \tilde{r}}$, we have

$$\mathbb{E}\left[\rho'(\kappa Z_1)\frac{1}{\tilde{r}}(Q_1 - \mathrm{Prox}_{\tilde{r}\rho}(Q_1))[-\lambda(\kappa\alpha Z_1 + \sigma Z_2) + 1]\right]$$

$$=\mathbb{E}\left[\rho'(\kappa Z_1)\frac{1}{\tilde{r}}(Q_2 + \tilde{r} + \mathrm{Prox}_{\tilde{r}\rho}(-Q_2))[-\lambda(\kappa\alpha Z_1 + \sigma Z_2) + 1]\right]$$

$$=\mathbb{E}\left[\rho'(\kappa Z_1)\frac{1}{\tilde{r}}(-(-Q_2) + \mathrm{Prox}_{\tilde{r}\rho}(-Q_2))[-\lambda(\kappa\alpha Z_1 + \sigma Z_2) + 1]\right] + \mathbb{E}\left[\rho'(\kappa Z_1)[-\lambda(\kappa\alpha Z_1 + \sigma Z_2) + 1]\right]$$

$$=\frac{1}{2} - \lambda\kappa^2\alpha\mathbb{E}(\rho''(\kappa Z_1)) - \mathbb{E}\left[\rho'(-\kappa Z_1)\frac{1}{\tilde{r}}(Q_2 - \mathrm{Prox}_{\tilde{r}\rho}(Q_2))\right]$$

$$- \mathbb{E}\left[\rho'(-\kappa Z_1)\frac{1}{\tilde{r}}(Q_2 - \mathrm{Prox}_{\tilde{r}\rho}(Q_2))[\lambda(\kappa\alpha Z_1 + \sigma Z_2)]\right].$$

Based on two simplifications in $\frac{\partial F_2}{\partial \tilde{r}}$, we can compute the partial derivative $\frac{\partial F}{\partial \tilde{r}}$:

$$\frac{\partial F(\alpha, \sigma, \tilde{r})}{\partial \tilde{r}} = \frac{\partial F_1}{\partial \tilde{r}} + \frac{\partial F_2}{\partial \tilde{r}} = \frac{1}{4} - \lambda\kappa^2\alpha\mathbb{E}(\rho''(\kappa Z_1)) - \frac{1}{\tilde{r}^2}\mathbb{E}\left[\rho'(-\kappa Z_1)(Q_2 - \mathrm{Prox}_{\tilde{r}\rho}(Q_2))^2\right]$$

$$- \frac{2}{\tilde{r}}\mathbb{E}\left[\rho'(-\kappa Z_1)(Q_2 - \mathrm{Prox}_{\tilde{r}\rho}(Q_2))[\lambda(\kappa\alpha Z_1 + \sigma Z_2)]\right]$$

Next we compute the derivative w.r.t $\sigma$, this is important for simplification in later steps.

$$\frac{\partial F(\alpha, \sigma, \tilde{r})}{\partial \sigma} = \mathbb{E}\left[\rho'(-\kappa Z_1)\frac{1}{\tilde{r}}(Q_2 - \mathrm{Prox}_{\tilde{r}\rho}(Q_2))[1 - \lambda\tilde{r}]Z_2\right] + \mathbb{E}\left[\rho'(\kappa Z_1)\frac{1}{\tilde{r}}(Q_2 + \mathrm{Prox}_{\tilde{r}\rho}(-Q_2))[1 - \lambda\tilde{r}]Z_2\right]$$

$$= \frac{2}{\tilde{r}}(1 - \lambda\tilde{r})\mathbb{E}\left[\rho'(-\kappa Z_1)(Q_2 - \mathrm{Prox}_{\tilde{r}\rho}(Q_2))Z_2\right]$$

$$= \frac{2}{\tilde{r}}(1 - \lambda\tilde{r})\mathbb{E}\left[\rho'(-\kappa Z_1)\left((1 - \lambda\tilde{r})[\kappa\alpha Z_1 + \sigma Z_2] - \mathrm{Prox}_{\tilde{r}\rho}((1 - \lambda\tilde{r})[\kappa\alpha Z_1 + \sigma Z_2])\right)Z_2\right]$$

$$= \frac{(1 - \lambda\tilde{r})^2\sigma}{\tilde{r}} - \frac{2(1 - \lambda\tilde{r})}{\tilde{r}}\mathbb{E}\left[Z_2\rho'(-\kappa Z_1)\mathrm{Prox}_{\tilde{r}\rho}((1 - \lambda\tilde{r})[\kappa\alpha Z_1 + \sigma Z_2])\right]$$

$$= \frac{(1 - \lambda\tilde{r})^2\sigma}{\tilde{r}} - \frac{2(1 - \lambda\tilde{r})}{\tilde{r}}\mathbb{E}\left(\frac{(1 - \lambda\tilde{r})\sigma\rho'(-\kappa Z_1)}{1 + \tilde{r}\rho''(\mathrm{Prox}_{\tilde{r}\rho}((1 - \lambda\tilde{r})[\kappa\alpha Z_1 + \sigma Z_2]))}\right)$$

$$= \frac{(1 - \lambda\tilde{r})^2\sigma}{\tilde{r}}\left(1 - 2\mathbb{E}\left(\frac{\rho'(-\kappa Z_1)}{1 + \tilde{r}\rho''(\mathrm{Prox}_{\tilde{r}\rho}((1 - \lambda\tilde{r})[\kappa\alpha Z_1 + \sigma Z_2]))}\right)\right)$$

Similarly, for derivative w.r.t $\alpha$, we have

$$\frac{\partial F(\alpha, \sigma, \tilde{r})}{\partial \alpha} = \mathbb{E}\left[\rho'(-\kappa Z_1)\frac{1}{\tilde{r}}(Q_2 - \mathrm{Prox}_{\tilde{r}\rho}(Q_2))[1 - \lambda\tilde{r}]\kappa Z_1\right] + \mathbb{E}\left[\rho'(\kappa Z_1)\frac{1}{\tilde{r}}(Q_2 + \tilde{r} + \mathrm{Prox}_{\tilde{r}\rho}(-Q_2))[1 - \lambda\tilde{r}]\kappa Z_1\right]$$

$$= (1 - \lambda\tilde{r})\kappa^2\mathbb{E}(\rho''(\kappa Z_1)) + \frac{2\kappa(1 - \lambda\tilde{r})}{\tilde{r}}\mathbb{E}\left[Z_1\rho'(-\kappa Z_1)(Q_2 - \mathrm{Prox}_{\tilde{r}\rho}(Q_2))\right]$$

$$= (1 - \lambda\tilde{r})\kappa^2\mathbb{E}(\rho''(\kappa Z_1)) + \frac{2\kappa(1 - \lambda\tilde{r})}{\tilde{r}}\mathbb{E}\left[Z_1\rho'(-\kappa Z_1)((1 - \lambda\tilde{r})[\kappa\alpha Z_1 + \sigma Z_2] - \mathrm{Prox}_{\tilde{r}\rho}(Q_2))\right]$$

$$= (1 - \lambda\tilde{r})\kappa^2\mathbb{E}(\rho''(\kappa Z_1)) + \frac{2\kappa(1 - \lambda\tilde{r})}{\tilde{r}}\left(\frac{(1 - \lambda\tilde{r})\kappa\alpha}{2} + \kappa\mathbb{E}[\rho''(-\kappa Z_1)\mathrm{Prox}_{\tilde{r}\rho}(Q_2)] - \mathbb{E}\left(\frac{(1 - \lambda\tilde{r})\kappa\alpha\rho'(-\kappa Z_1)}{1 + \tilde{r}\rho''(\mathrm{Prox}_{\tilde{r}\rho}(Q_2))}\right)\right)$$

$$= (1 - \lambda\tilde{r})\kappa^2\mathbb{E}(\rho''(\kappa Z_1)) + \frac{(1 - \lambda\tilde{r})^2\kappa^2\alpha}{\tilde{r}} + \frac{2\kappa^2(1 - \lambda\tilde{r})}{\tilde{r}}\mathbb{E}[\rho''(-\kappa Z_1)\mathrm{Prox}_{\tilde{r}\rho}(Q_2)]$$

$$+ \frac{2(1 - \lambda\tilde{r})^2\kappa^2\alpha}{\tilde{r}}\mathbb{E}\left(\frac{\rho'(-\kappa Z_1)}{1 + \tilde{r}\rho''(\mathrm{Prox}_{\tilde{r}\rho}(Q_2))}\right).$$

Next we can compute the partial derivative $\frac{\partial C}{\partial v}, \frac{\partial C}{\partial \alpha}$, using chain rule and derived $\frac{\partial F(\alpha, \sigma, \tilde{r})}{\partial \tilde{r}}$, we have

$$\begin{aligned}
\frac{\partial C}{\partial v} &= -\frac{r}{2v^2} + \frac{r}{4(rv+\lambda)} - \frac{r\lambda}{(rv+\lambda)^2}\kappa^2\alpha\mathbb{E}(\rho''(\kappa Z_1)) + \frac{\lambda(\kappa^2\alpha^2+\sigma^2)r\lambda}{2(rv+\lambda)^2} - \frac{r}{(rv+\lambda)^2}\frac{\partial F}{\partial \tilde{r}} \\
&= -\frac{r}{2v^2} + \frac{\tilde{r}^2\lambda^2 r^2(\kappa^2\alpha^2+\sigma^2)}{2} + r\mathbb{E}\left[\rho'(-\kappa Z_1)(Q_2 - \operatorname{Prox}_{\tilde{r}\rho}(Q_2))^2\right] \\
&\quad + 2r\tilde{r}\mathbb{E}\left[\rho'(-\kappa Z_1)(Q_2 - \operatorname{Prox}_{\tilde{r}\rho}(Q_2))[\lambda(\kappa\alpha Z_1 + \sigma Z_2)]\right]
\end{aligned}$$

and

$$\frac{\partial C}{\partial r} = -\frac{\sigma}{\sqrt{\delta}} + \frac{1}{2v} + \frac{v}{4(rv+\lambda)^2} - \frac{v\lambda}{(rv+\lambda)^2}\kappa^2\alpha\mathbb{E}(\rho''(\kappa Z_1)) + \frac{\lambda^2 v(\kappa^2\alpha^2+\sigma^2)}{2(rv+\lambda)^2} - \frac{v}{(rv+\lambda)^2}\frac{\partial F}{\partial \tilde{r}}$$

Setting both $\frac{\partial C}{\partial r} = 0$ and $\frac{\partial C}{\partial v} = 0$ we can get $\frac{1}{v} = \frac{\sigma}{\sqrt{\delta}}$ and

$$\begin{aligned}
\frac{\sigma^2}{2\delta} &= \frac{\tilde{r}^2\lambda^2(\kappa^2\alpha^2+\sigma^2)}{2} + \mathbb{E}\left[\rho'(-\kappa Z_1)(Q_2 - \operatorname{Prox}_{\tilde{r}\rho}(Q_2))^2\right] \\
&\quad + 2\tilde{r}\mathbb{E}\left[\rho'(-\kappa Z_1)(Q_2 - \operatorname{Prox}_{\tilde{r}\rho}(Q_2))[\lambda(\kappa\alpha Z_1 + \sigma Z_2)]\right].
\end{aligned} \tag{38}$$

Next we compute $\frac{\partial C}{\partial \sigma}, \frac{\partial C}{\partial r}$ based on already derived $\frac{\partial F(\alpha, \sigma, \tilde{r})}{\partial \alpha}$ and $\frac{\partial F(\alpha, \sigma, \tilde{r})}{\partial \sigma}$:

$$\begin{aligned}
\frac{\partial C}{\partial \alpha} &= \frac{rv\kappa^2}{(rv+\lambda)}E(\rho''(\kappa Z_1)) + \frac{rv\lambda}{(rv+\lambda)}\kappa^2\alpha + \frac{\partial F}{\partial \alpha} \\
&= \frac{rv\lambda}{(rv+\lambda)}\kappa^2\alpha + \frac{(1-\lambda\tilde{r})^2\kappa^2\alpha}{\tilde{r}} + \frac{2\kappa^2(1-\lambda\tilde{r})}{\tilde{r}}\mathbb{E}[\rho''(-\kappa Z_1)\operatorname{Prox}_{\tilde{r}\rho}(Q_2)] + \frac{2(1-\lambda\tilde{r})^2\kappa^2\alpha}{\tilde{r}}\mathbb{E}\left(\frac{\rho'(-\kappa Z_1)}{1+\tilde{r}\rho''(\operatorname{Prox}_{\tilde{r}\rho}(Q_2))}\right) \\
\frac{\partial C}{\partial \sigma} &= \frac{-r}{\sqrt{\delta}} + \frac{rv\lambda\sigma}{(rv+\lambda)} + \frac{\partial F}{\partial \sigma} \\
&= \frac{-r}{\sqrt{\delta}} + \frac{rv\lambda\sigma}{(rv+\lambda)} + \frac{(1-\lambda\tilde{r})^2\sigma}{\tilde{r}}\left(1 - 2\mathbb{E}\left(\frac{\rho'(-\kappa Z_1)}{1+\tilde{r}\rho''(\operatorname{Prox}_{\tilde{r}\rho}((1-\lambda\tilde{r})[\kappa\alpha Z_1 + \sigma Z_2]))}\right)\right)
\end{aligned}$$

Setting both $\frac{\partial C}{\partial \sigma} = 0$ and $\frac{\partial C}{\partial \alpha} = 0$ and use $\frac{1}{v} = \frac{\sigma}{\sqrt{\delta}}$ and $\tilde{r} = \frac{1}{r+\lambda v}$, we can get

$$-\frac{\alpha}{2\delta} = \mathbb{E}[\rho''(-\kappa Z_1)\operatorname{Prox}_{\tilde{r}\rho}(Q_2)]. \tag{39}$$

$$1 - \frac{1}{\delta} = 2(1-\lambda\tilde{r})E\left(\frac{\rho'(-\kappa Z_1)}{1+\tilde{r}\rho''(\operatorname{Prox}_{\tilde{r}\rho}(Q_2))}\right). \tag{40}$$

Combine (38),(39) and (40), we have

$$\begin{cases}
\frac{\sigma^2}{2\delta} = \frac{\tilde{r}^2\lambda^2(\kappa^2\alpha^2+\sigma^2)}{2} + \mathbb{E}\left[\rho'(-\kappa Z_1)\left((1-\lambda\tilde{r})[\kappa\alpha Z_1 + \sigma Z_2] - \operatorname{Prox}_{\tilde{r}\rho}((1-\lambda\tilde{r})[\kappa\alpha Z_1 + \sigma Z_2])\right)^2\right] \\
\quad + 2\lambda\tilde{r}\mathbb{E}\left[\rho'(-\kappa Z_1)\left((1-\lambda\tilde{r})[\kappa\alpha Z_1 + \sigma Z_2] - \operatorname{Prox}_{\tilde{r}\rho}((1-\lambda\tilde{r})[\kappa\alpha Z_1 + \sigma Z_2])\right)[(\kappa\alpha Z_1 + \sigma Z_2)]\right] \\
-\frac{\alpha}{2\delta} = \mathbb{E}[\rho''(-\kappa Z_1)\operatorname{Prox}_{\tilde{r}\rho}((1-\lambda\tilde{r})[\kappa\alpha Z_1 + \sigma Z_2])] \\
1 - \frac{1}{\delta} = 2(1-\lambda\tilde{r})E\left(\frac{\rho'(-\kappa Z_1)}{1+\tilde{r}\rho''(\operatorname{Prox}_{\tilde{r}\rho}((1-\lambda\tilde{r})[\kappa\alpha Z_1 + \sigma Z_2]))}\right)
\end{cases} \tag{41}$$

## C.4. Uniqueness of the population optima

The system of equations derived in the previous section provides a way to compute the constants $(\alpha_*, \sigma_*)$. However, to establish a well-defined relationship between the proxy quantities and the signal strength $\kappa$, we must ensure that this solution is unique. In this section, we prove the uniqueness of the optimizer for the scalar optimization problem in (33).

Although the objective function $\mathcal{R}(\sigma, r, \tilde{\nu}, \alpha)$ is jointly convex in $(\sigma, \tilde{\nu}, \alpha)$ and concave in $r$. To guarantee unique optimal solution $(\sigma_*, r_*, \tilde{\nu}_*, \alpha_*)$, we shall verify that (1) for fixed $r$, $\mathcal{R}(\sigma, r, \tilde{\nu}, \alpha)$ is jointly **strictly** convex in $(\sigma, \tilde{\nu}, \alpha)$ (2) for fixed $(\sigma, \tilde{\nu}, \alpha)$, $\mathcal{R}(\sigma, r, \tilde{\nu}, \alpha)$ is **strictly** concave in $r$.

We begin with proof of strict convexity, the objective function can be decomposed into three parts:

$$(\textbf{part one}) \quad F_1(\sigma, \tilde{\nu}, \alpha) = -\frac{r\sigma}{\sqrt{\delta}} + \frac{r\tilde{\nu}}{2} + \frac{\lambda r/\tilde{\nu}}{2(\lambda + r/\tilde{\nu})}(\kappa^2\alpha^2 + \sigma^2),$$

$$(\textbf{part two}) \quad F_2(\sigma, \tilde{\nu}, \alpha) = \mathbb{E}\left(\rho'(-\kappa Z_1)M_{\rho(\cdot)}\left(\frac{r/\tilde{\nu}}{\lambda + r/\tilde{\nu}}\kappa\alpha Z_1 + \frac{r/\tilde{\nu}}{\lambda + r/\tilde{\nu}}\sigma Z_2, \frac{1}{\lambda + r/\tilde{\nu}}\right)\right),$$

$$(\textbf{part three}) \quad F_3(\sigma, \tilde{\nu}, \alpha) = \mathbb{E}\left(\rho'(\kappa Z_1)M_{\rho(\cdot)}\left(\frac{r/\tilde{\nu}}{\lambda + r/\tilde{\nu}}\kappa\alpha Z_1 + \frac{r/\tilde{\nu}}{\lambda + r/\tilde{\nu}}\sigma Z_2 + \frac{1}{\lambda + r/\tilde{\nu}}, \frac{1}{\lambda + r/\tilde{\nu}}\right)\right)$$
$$- \frac{1}{4(\lambda + r/\tilde{\nu})} - \frac{r/\tilde{\nu}}{\lambda + r/\tilde{\nu}}\kappa^2\alpha\mathbb{E}_Z\left[\rho''(\kappa Z)\right]. \tag{42}$$

We begin by simplifying $F_3(\sigma, \tilde{\nu}, \alpha)$,

$$F_3(\sigma, \tilde{\nu}, \alpha) = \mathbb{E}\left(\rho'(\kappa Z_1)M_{\rho(\cdot)}\left(\frac{r/\tilde{\nu}}{\lambda + r/\tilde{\nu}}\kappa\alpha Z_1 + \frac{r/\tilde{\nu}}{\lambda + r/\tilde{\nu}}\sigma Z_2 + \frac{1}{\lambda + r/\tilde{\nu}}, \frac{1}{\lambda + r/\tilde{\nu}}\right)\right)$$
$$- \frac{1}{4(\lambda + r/\tilde{\nu})} - \frac{r/\tilde{\nu}}{\lambda + r/\tilde{\nu}}\kappa^2\alpha\mathbb{E}_Z\left[\rho''(\kappa Z)\right]$$
$$= \mathbb{E}\left(\rho'(\kappa Z_1)\min_t\left[\rho(t) + \frac{\lambda + r/\tilde{\nu}}{2}\left(\frac{r/\tilde{\nu}}{\lambda + r/\tilde{\nu}}\kappa\alpha Z_1 + \frac{r/\tilde{\nu}}{\lambda + r/\tilde{\nu}}\sigma Z_2 + \frac{1}{\lambda + r/\tilde{\nu}} - t\right)^2\right]\right)$$
$$- \frac{1}{4(\lambda + r/\tilde{\nu})} - \frac{r/\tilde{\nu}}{\lambda + r/\tilde{\nu}}\kappa^2\alpha\mathbb{E}_Z\left[\rho''(\kappa Z)\right]$$
$$= \mathbb{E}\left(\rho'(\kappa Z_1)\min_t\left[\rho(t) - t + \frac{\lambda + r/\tilde{\nu}}{2}\left(\frac{r/\tilde{\nu}}{\lambda + r/\tilde{\nu}}\kappa\alpha Z_1 + \frac{r/\tilde{\nu}}{\lambda + r/\tilde{\nu}}\sigma Z_2 - t\right)^2\right]\right)$$
$$+ \mathbb{E}\left(\rho'(\kappa Z_1)\left[\frac{1}{2(\lambda + r/\tilde{\nu})} + \left(\frac{r/\tilde{\nu}}{\lambda + r/\tilde{\nu}}\kappa\alpha Z_1 + \frac{r/\tilde{\nu}}{\lambda + r/\tilde{\nu}}\sigma Z_2\right)\right]\right)$$
$$- \frac{1}{4(\lambda + r/\tilde{\nu})} - \frac{r/\tilde{\nu}}{\lambda + r/\tilde{\nu}}\kappa^2\alpha\mathbb{E}_Z\left[\rho''(\kappa Z)\right]$$

$$\overset{(1)}{=} \mathbb{E}\left(\rho'(\kappa Z_1)\min_t\left[\rho(t) - t + \frac{\lambda + r/\tilde{\nu}}{2}\left(\frac{r/\tilde{\nu}}{\lambda + r/\tilde{\nu}}\kappa\alpha Z_1 + \frac{r/\tilde{\nu}}{\lambda + r/\tilde{\nu}}\sigma Z_2 - t\right)^2\right]\right)$$
$$\overset{(2)}{=} \mathbb{E}\left(\rho'(\kappa Z_1)\min_t\left[\rho(t) - t + \frac{\lambda}{2}t^2 + \frac{r}{2\tilde{\nu}}(t - (\kappa\alpha Z_1 + \sigma Z_2))^2\right]\right)$$
$$+ \mathbb{E}\left(\rho'(\kappa Z_1)\left[\frac{r^2/\tilde{\nu}^2}{2(\lambda + r/\tilde{\nu})}(\kappa\alpha Z_1 + \sigma Z_2)^2 - \frac{r}{2\tilde{\nu}}(\kappa\alpha Z_1 + \sigma Z_2)^2\right]\right)$$
$$= \mathbb{E}\left(\rho'(\kappa Z_1)\min_t\left[\rho(t) - t + \frac{\lambda}{2}t^2 + \frac{r}{2\tilde{\nu}}(t - (\kappa\alpha Z_1 + \sigma Z_2))^2\right]\right)$$
$$+ \frac{r^2/\tilde{\nu}^2}{2(\lambda + r/\tilde{\nu})}\frac{(\kappa^2\alpha^2 + \sigma^2)}{2} - \frac{r}{2\tilde{\nu}}\frac{(\kappa^2\alpha^2 + \sigma^2)}{2}$$
$$= \mathbb{E}\left(\rho'(\kappa Z_1)\min_t\left[\rho(t) - t + \frac{\lambda}{2}t^2 + \frac{r}{2\tilde{\nu}}(t - (\kappa\alpha Z_1 + \sigma Z_2))^2\right]\right) - \frac{r\lambda\left(\kappa^2\alpha^2 + \sigma^2\right)}{4(\lambda\tilde{\nu} + r)}$$

where (1) we use $\mathbb{E}(\rho'(\kappa Z_1)) = 1/2$ and stein identity $\mathbb{E}(\rho'(\kappa Z_1)Z_1) = \kappa\mathbb{E}_Z\left[\rho''(\kappa Z)\right]$ and (2) we use the following identity:

$$\frac{\lambda + a}{2}\left(t - \frac{a}{\lambda + a}w\right)^2 + \frac{a}{2}w^2 - \frac{a^2}{2(\lambda + a)}w^2 = \frac{\lambda t^2}{2} + \frac{a}{2}(t - w)^2$$

Similarly, we can show that

$$F_2(\sigma, \tilde{\nu}, \alpha) = \mathbb{E}\left(\rho'(-\kappa Z_1) M_{\rho(\cdot)}\left(\frac{r/\tilde{\nu}}{\lambda + r/\tilde{\nu}}\kappa\alpha Z_1 + \frac{r/\tilde{\nu}}{\lambda + r/\tilde{\nu}}\sigma Z_2, \frac{1}{\lambda + r/\tilde{\nu}}\right)\right)$$

$$= \mathbb{E}\left(\rho'(-\kappa Z_1)\min_t\left[\rho(t) + \frac{\lambda + r/\tilde{\nu}}{2}\left(\frac{r/\tilde{\nu}}{\lambda + r/\tilde{\nu}}\kappa\alpha Z_1 + \frac{r/\tilde{\nu}}{\lambda + r/\tilde{\nu}}\sigma Z_2 - t\right)^2\right]\right)$$

$$= \mathbb{E}\left(\rho'(-\kappa Z_1)\min_t\left[\rho(t) + \frac{\lambda}{2}t^2 + \frac{r}{2\tilde{\nu}}(t - (\kappa\alpha Z_1 + \sigma Z_2))^2\right]\right) - \frac{r\lambda\left(\kappa^2\alpha^2 + \sigma^2\right)}{4(\lambda\tilde{\nu} + r)}$$

Therefore, we can simplify the $\mathcal{R}(\sigma, r, \tilde{\nu}, \alpha)$ as follows: For fixed $r > 0$,

$$\mathcal{R}(\sigma, r, \tilde{\nu}, \alpha) = F_1(\sigma, \tilde{\nu}, \alpha) + F_2(\sigma, \tilde{\nu}, \alpha) + F_3(\sigma, \tilde{\nu}, \alpha)$$

$$= -\frac{r\sigma}{\sqrt{\delta}} + \frac{r\tilde{\nu}}{2} + \mathbb{E}\left(\rho'(-\kappa Z_1)\min_t\left[\rho(t) + \frac{\lambda}{2}t^2 + \frac{r}{2\tilde{\nu}}(t - (\kappa\alpha Z_1 + \sigma Z_2))^2\right]\right)$$

$$+ \mathbb{E}\left(\rho'(\kappa Z_1)\min_t\left[\rho(t) - t + \frac{\lambda}{2}t^2 + \frac{r}{2\tilde{\nu}}(t - (\kappa\alpha Z_1 + \sigma Z_2))^2\right]\right)$$

## Step (1): strict convexity

To show the strict convexity of $\mathcal{R}(\sigma, r, \tilde{\nu}, \alpha)$ for fixed $r > 0$, it suffices to show that the hessian of following function is jointly strictly convex in $(\sigma, \tilde{\nu}, \alpha)$.

1. $G_1(\sigma, \tilde{\nu}, \alpha) = \mathbb{E}\left(\rho'(-\kappa Z_1)\min_t\left[\rho(t) + \frac{\lambda}{2}t^2 + \frac{r}{2\tilde{\nu}}(t - (\kappa\alpha Z_1 + \sigma Z_2))^2\right]\right)$

2. $G_2(\sigma, \tilde{\nu}, \alpha) = \mathbb{E}\left(\rho'(\kappa Z_1)\min_t\left[\rho(t) - t + \frac{\lambda}{2}t^2 + \frac{r}{2\tilde{\nu}}(t - (\kappa\alpha Z_1 + \sigma Z_2))^2\right]\right)$

Since $\rho(t)$ is a convex function, and the term $\frac{r}{\tilde{\nu}}\left(t - \kappa\alpha Z_1 - \sigma Z_2\right)^2$ can be written as $\frac{r}{\tilde{\nu}}\|(t, -\kappa\alpha Z_1, -\sigma Z_2)\|^2$, which is the perspective function of the squared norm, known to be jointly convex in $(t, \alpha, \sigma, \tilde{\nu})$ when $\tilde{\nu} > 0$. It suffices to show that $G_1(\sigma, \tilde{\nu}, \alpha)$ is strictly convex in $(\sigma, \tilde{\nu}, \alpha)$.

By Lemma C.7 and the fact that $\rho'(t) > 0$ for all $t \in \mathbb{R}$, it suffices to show that

$$\mathbb{E}\left(\min_t\left[\rho(t) + \frac{\lambda}{2}t^2 + \frac{r}{2\tilde{\nu}}(t - (\kappa\alpha Z_1 + \sigma Z_2))^2\right]\right) \tag{43}$$

is strictly convex in $(\sigma, \tilde{\nu}, \alpha)$. Define $q(\alpha, \sigma) = \sqrt{\kappa^2\alpha^2 + \sigma^2}$, then we have

$$\mathbb{E}\left(\min_t\left[\rho(t) + \frac{\lambda}{2}t^2 + \frac{r}{2\tilde{\nu}}(t - (\kappa\alpha Z_1 + \sigma Z_2))^2\right]\right) = \mathbb{E}\left(\min_t\left[\rho(t) + \frac{\lambda}{2}t^2 + \frac{r}{2\tilde{\nu}}(t - q(\alpha, \sigma)Z)^2\right]\right).$$

We first show that

$$L(q, \tilde{\nu}) := \mathbb{E}\left[\min_t\left\{\rho(t) + \frac{\lambda}{2}t^2 + \frac{r}{2\tilde{\nu}}(t - qZ)^2\right\}\right] := \mathbb{E}[M_{\tilde{\rho}(\cdot)}(qZ, \frac{\tilde{\nu}}{r})]$$

is strictly convex in $(q, \tilde{\nu})$, where $\tilde{\rho}(t) := \rho(t) + \frac{\lambda}{2}t^2$ is a strictly convex function. The proof is adapted from Lemma C.1 in (Thrampoulidis et al., 2018), we present the modified version here for completeness.

For any $q > 0, \tilde{\nu} > 0$, it suffices to show that

$$\Gamma(x, y) := L(q + x, \tilde{\nu} + y) - L(q, \tilde{\nu}) - L_1(q, \tilde{\nu})x - L_2(q, \tilde{\nu})y > 0, \quad \text{for all } x > -q, y > -\tilde{\nu},$$

where $L_1 = \partial L/\partial q$ and $L_2 = \partial L/\partial \tilde{\nu}$. First note that $M_{\tilde{\rho}}(a, b)$ is jointly convex in $(a, b)$, this implies that $\Gamma(x, y)$ is jointly convex in $(x, y)$. Moreover, $\Gamma(0, 0) = 0$, by mean value theorem, there exist $t^* \in (0, 1)$ s.t.,

$$\Gamma(x,y) - \Gamma(0,0) = [\nabla\Gamma(t^*x, t^*y) - \nabla\Gamma(0,0)]^\top (x,y)$$
$$= \frac{1}{t^*} \nabla\Gamma(t^*x, t^*y)^\top (t^*x, t^*y)$$

Here we use

$$\nabla\Gamma(0,0) = \left[ \begin{array}{c} \frac{\partial\Gamma}{\partial x}(0,0) \\ \frac{\partial\Gamma}{\partial y}(0,0) \end{array} \right] = \left[ \begin{array}{c} L_1(q+0, \tilde\nu + 0) - L_1(q, \tilde\nu) \\ L_2(q+0, \tilde\nu + 0) - L_2(q, \tilde\nu) \end{array} \right] = \left[ \begin{array}{c} 0 \\ 0 \end{array} \right].$$

So it suffices to show for any $x > -q, y > -\tilde\nu$

$$\nabla\Gamma(x,y)^\top (x,y) > 0 \text{ for all } (x,y) \neq (0,0).$$

This is equivalent to show

$$[L_1(q+x, \tilde\nu + y) - L_1(q, \tilde\nu)] x + [L_2(q+x, \tilde\nu + y) - L_2(q, \tilde\nu)] y > 0$$

Using dominated convergence to interchange derivatives and expectation, it is equivalent to show

$$[L_1(q+x, \tilde\nu + y) - L_1(q, \tilde\nu)] x + [L_2(q+x, \tilde\nu + y) - L_2(q, \tilde\nu)] y$$
$$= x\mathbb{E}\left[ Z\tilde\rho'\left(Prox_{\tilde\rho}((q+x)Z; \frac{\tilde\nu + y}{r})\right) - Z\tilde\rho'\left(Prox_{\tilde\rho}(qZ; \frac{\tilde\nu}{r})\right) \right]$$
$$+ y\mathbb{E}\left[ \frac{-1}{2r}\tilde\rho'\left(Prox_{\tilde\rho}((q+x)Z; \frac{\tilde\nu + y}{r})\right)^2 + \frac{1}{2r}\tilde\rho'\left(Prox_{\tilde\rho}(qZ; \frac{\tilde\nu}{r})\right)^2 \right]$$
$$= \mathbb{E}\left\{ \left[ \tilde\rho'\left(Prox_{\tilde\rho}((q+x)Z; \frac{\tilde\nu + y}{r})\right) - \tilde\rho'\left(Prox_{\tilde\rho}(qZ; \frac{\tilde\nu}{r})\right) \right] \times \right.$$
$$\left. \left[ xZ - \frac{y}{2r}\left( \tilde\rho'\left(Prox_{\tilde\rho}((q+x)Z; \frac{\tilde\nu + y}{r})\right) + \tilde\rho'\left(Prox_{\tilde\rho}(qZ; \frac{\tilde\nu}{r})\right) \right) \right] \right\}$$

$$= \left( \frac{\tilde\nu}{r} + \frac{y}{2r} \right) \mathbb{E}\left[ \tilde\rho'\left(Prox_{\tilde\rho}((q+x)Z; \frac{\tilde\nu + y}{r})\right) - \tilde\rho'\left(Prox_{\tilde\rho}(qZ; \frac{\tilde\nu}{r})\right) \right]^2$$
$$+ \mathbb{E}\left[ \tilde\rho'\left(Prox_{\tilde\rho}\left((q+x)Z; \frac{\tilde\nu + y}{r}\right)\right) - \tilde\rho'\left(Prox_{\tilde\rho}\left(qZ; \frac{\tilde\nu}{r}\right)\right) \right]$$
$$\times \left[ Prox_{\tilde\rho}\left((q+x)Z; \frac{\tilde\nu + y}{r}\right) - Prox_{\tilde\rho}\left(qZ; \frac{\tilde\nu}{r}\right) \right] > 0$$

where we use identity $z - Prox_{\tilde\rho}(z; b) = b\tilde\rho'(Prox_{\tilde\rho}(z; b))$ in last equation. The proof is completed by observing that

(1) $\left( \frac{\tilde\nu}{r} + \frac{y}{2r} \right) > 0,$

(2) $\tilde\rho''(t) > 0 \implies [\tilde\rho'(t_1) - \tilde\rho'(t_2)](t_1 - t_2) > 0,$

(3) $\mathbb{E}\left[ \tilde\rho'\left(Prox_{\tilde\rho}((q+x)Z; \frac{\tilde\nu + y}{r})\right) - \tilde\rho'\left(Prox_{\tilde\rho}(qZ; \frac{\tilde\nu}{r})\right) \right]^2 > 0.$

We have show that $L(q, \tilde\nu) := \mathbb{E}[M_{\tilde\rho(\cdot)}(qZ, \frac{\tilde\nu}{r})]$ is jointly strictly convex in $(q, \tilde\nu)$. Next we will establish that $\mathbb{E}[M_{\tilde\rho(\cdot)}(\kappa\alpha Z_1 + \sigma Z_2, \frac{\tilde\nu}{r})]$ is strictly convex in $(\sigma, \tilde\nu, \alpha)$, where $Z_i \overset{i.i.d}{\sim} N(0,1)$.

By dominated convergence theorem and Lemma C.8, for fixed $\tilde\nu > 0$ we have

$$\frac{\partial L}{\partial q} = \mathbb{E}\left[ \frac{q\tilde\rho''\left(Prox_{\tilde\rho}(qZ; \frac{\tilde\nu}{r})\right)}{1 + \frac{\tilde\nu}{r}\tilde\rho''\left(Prox_{\tilde\rho}(qZ; \frac{\tilde\nu}{r})\right)} \right] > 0, \tag{44}$$

which suggests that for fixed $\tilde{\nu}$, $L(q, \tilde{\nu})$ is strictly increasing and strictly convex in $q$.

Now fix $\tilde{\nu} > 0$. Write $\xi = (\alpha, \sigma)$ and let $\lambda \in (0, 1)$. Define the diagonal matrix $D = \mathrm{diag}\,(\kappa, 1)$, so that $q(\xi) = \|D\xi\|_2$. Take two distinct vectors $\xi_1 = \left(\alpha^{(1)}, \sigma^{(1)}\right) \neq \xi_2 = \left(\alpha^{(2)}, \sigma^{(2)}\right)$. We will prove strict convexity of the map $\xi \mapsto L(q(\xi), \tilde{\nu})$ by the definition of strict convexity.

- **Case 1**: $\xi_1$ is not parallel to $\xi_2$. Then, by strict convexity of the Euclidean norm, we have

$$
\begin{aligned}
q((1 - \lambda)\xi_1 + \lambda\xi_2) &= \|D[(1 - \lambda)\xi_1 + \lambda\xi_2)]\|_2 \\
&< (1 - \lambda)\|D\xi_1\|_2 + \lambda\|D\xi_2\|_2 \\
&= (1 - \lambda)q(\xi_1) + \lambda q(\xi_2)
\end{aligned}
$$

Let us denote the convex combination of components as

$$
\tilde{\alpha} = (1 - \lambda)\alpha^{(1)} + \lambda\alpha^{(2)}, \quad \tilde{\sigma} = (1 - \lambda)\sigma^{(1)} + \lambda\sigma^{(2)}.
$$

Then we have

$$
\begin{aligned}
&\mathbb{E}\left[M_{\tilde{\rho}(\cdot)}\left(\kappa\tilde{\alpha}Z_1 + \tilde{\sigma}Z_2, \frac{\tilde{\nu}}{r}\right)\right] \\
&= \mathbb{E}\left[M_{\tilde{\rho}(\cdot)}\left(\sqrt{\kappa^2\tilde{\alpha}^2 + \tilde{\sigma}^2}Z, \frac{\tilde{\nu}}{r}\right)\right] \\
&= \mathbb{E}\left[M_{\tilde{\rho}(\cdot)}\left(q((1 - \lambda)\xi_1 + \lambda\xi_2)Z, \frac{\tilde{\nu}}{r}\right)\right] \\
&< \mathbb{E}\left[M_{\tilde{\rho}(\cdot)}\left((1 - \lambda)q(\xi_1)Z + \lambda q(\xi_2)Z, \frac{\tilde{\nu}}{r}\right)\right] \\
&< (1 - \lambda)\mathbb{E}\left[M_{\tilde{\rho}(\cdot)}\left(q(\xi_1)Z, \frac{\tilde{\nu}}{r}\right)\right] + \lambda\mathbb{E}\left[M_{\tilde{\rho}(\cdot)}\left(q(\xi_2)Z, \frac{\tilde{\nu}}{r}\right)\right] \\
&= (1 - \lambda)\mathbb{E}\left[M_{\tilde{\rho}(\cdot)}\left(\kappa\alpha^{(1)}Z_1 + \sigma^{(1)}Z_2, \frac{\tilde{\nu}}{r}\right)\right] \\
&\quad + \lambda\mathbb{E}\left[M_{\tilde{\rho}(\cdot)}\left(\kappa\alpha^{(2)}Z_1 + \sigma^{(2)}Z_2, \frac{\tilde{\nu}}{r}\right)\right],
\end{aligned}
$$

where the first inequality follows from the strictly increasing property, and the second inequality follows from strict convexity(it is possible that $q(\xi_1) = q(\xi_2)$).

- **Case 2**: when $\xi_1$ is parallel to $\xi_2$ but $\|\xi_1\| \neq \|\xi_2\|$, we have $q((1 - \lambda)\xi_1 + \lambda\xi_2) = (1 - \lambda)q(\xi_1) + \lambda q(\xi_2)$. But $q(\xi_1) \neq q(\xi_2)$, then by strict convexity, we have

$$
\begin{aligned}
&\mathbb{E}\left[M_{\tilde{\rho}(\cdot)}\left(q((1 - \lambda)\xi_1 + \lambda\xi_2)Z, \frac{\tilde{\nu}}{r}\right)\right] \\
&= \mathbb{E}\left[M_{\tilde{\rho}(\cdot)}\left((1 - \lambda)q(\xi_1)Z + \lambda q(\xi_2)Z, \frac{\tilde{\nu}}{r}\right)\right] \\
&< (1 - \lambda)\mathbb{E}\left[M_{\tilde{\rho}(\cdot)}\left(q(\xi_1)Z, \frac{\tilde{\nu}}{r}\right)\right] + \lambda\mathbb{E}\left[M_{\tilde{\rho}(\cdot)}\left(q(\xi_2)Z, \frac{\tilde{\nu}}{r}\right)\right]
\end{aligned}
$$

Based on these two cases, we conclude that for fixed $\tilde{\nu}$, $\mathbb{E}\left[M_{\tilde{\rho}(\cdot)}\left(\kappa_1\alpha Z_1 + \sigma Z_2, \frac{\tilde{\nu}}{r}\right)\right]$ is strictly convex in $(\alpha, \alpha, \sigma)$.

For joint convexity including $\tilde{\nu}$, let $(\alpha^{(1)}, \sigma^{(1)}, \tilde{\nu}^{(1)}) \neq (\alpha^{(2)}, , \sigma^{(2)}, \tilde{\nu}^{(2)})$, and define $\tilde{\alpha} = (1 - \lambda)\alpha^{(1)} + \lambda\alpha^{(2)}$, $\tilde{\sigma}, \tilde{\tilde{\nu}}$ defined similarly.

- **Scenario 1**: If $(\alpha^{(1)}, \sigma^{(1)}) = (\alpha^{(2)}, \sigma^{(2)}), \tilde{\nu}^{(1)} \neq \tilde{\nu}^{(2)}$

$$\mathbb{E}\left[M_{\tilde{\rho}(\cdot)}\left(\kappa\tilde{\alpha}Z_1 + \tilde{\sigma}Z_2, \frac{(1-\lambda)\tilde{\nu}^{(1)} + \lambda\tilde{\nu}^{(2)}}{r}\right)\right]$$

$$= \mathbb{E}\left[M_{\tilde{\rho}(\cdot)}\left(\sqrt{\kappa^2\tilde{\alpha}^2 + \tilde{\sigma}^2}Z, \frac{(1-\lambda)\tilde{\nu}^{(1)} + \lambda\tilde{\nu}^{(2)}}{r}\right)\right]$$

$$= \mathbb{E}\left[M_{\tilde{\rho}(\cdot)}\left((1-\lambda+\lambda)\sqrt{\kappa^2\tilde{\alpha}^2 + \tilde{\sigma}^2}Z, \frac{(1-\lambda)\tilde{\nu}^{(1)} + \lambda\tilde{\nu}^{(2)}}{r}\right)\right]$$

$$< (1-\lambda)\mathbb{E}\left[M_{\tilde{\rho}(\cdot)}\left(\sqrt{\kappa^2\tilde{\alpha}^2 + \tilde{\sigma}^2}Z, \frac{\tilde{\nu}^{(1)}}{r}\right)\right]$$

$$+ \lambda\mathbb{E}\left[M_{\tilde{\rho}(\cdot)}\left(\sqrt{\kappa^2\tilde{\alpha}^2 + \tilde{\sigma}^2}Z, \frac{\tilde{\nu}^{(2)}}{r}\right)\right]$$

$$= (1-\lambda)\mathbb{E}\left[M_{\tilde{\rho}(\cdot)}\left(\kappa\alpha^{(1)}Z_1 + \sigma^{(1)}Z_2, \frac{\tilde{\nu}^{(1)}}{r}\right)\right]$$

$$+ \lambda\mathbb{E}\left[M_{\tilde{\rho}(\cdot)}\left(\kappa\alpha^{(2)}Z_1 + \sigma^{(2)}Z_2, \frac{\tilde{\nu}^{(2)}}{r}\right)\right]$$

where inequality follows from **Step (1) strict convexity**.

- **Scenario 2**: If $(\alpha^{(1)}, \sigma^{(1)}) \neq (\alpha^{(2)}, \sigma^{(2)}), \tilde{\nu}^{(1)} = \tilde{\nu}^{(2)}$, this scenario is identical to the fixed $\tilde{\nu}$ case, which we have already established in Case 1 and Case 2 above.

- **Scenario 3**: If $(\alpha^{(1)}, \sigma^{(1)}) \neq (\alpha^{(2)}, \sigma^{(2)})$ and $\tilde{\nu}^{(1)} \neq \tilde{\nu}^{(2)}$, denote $\xi_1 = (\alpha^{(1)}, \sigma^{(1)})$, $\xi_2 = (\alpha^{(2)}, \sigma^{(2)})$. By the subadditivity of the $L_2$ norm, we have

$$\sqrt{\kappa^2\tilde{\alpha}^2 + \tilde{\sigma}^2} = \left\|D\left(\begin{array}{c}(1-\lambda)\alpha^{(1)} + \lambda\alpha^{(2)} \\ (1-\lambda)\sigma^{(1)} + \lambda\sigma^{(2)}\end{array}\right)\right\|_2 \leq (1-\lambda)\left\|D\left(\begin{array}{c}\alpha^{(1)} \\ \sigma^{(1)}\end{array}\right)\right\|_2 + \lambda\left\|D\left(\begin{array}{c}\alpha^{(2)} \\ \sigma^{(2)}\end{array}\right)\right\|_2$$

We prove strict convexity by definition:

$$\mathbb{E}\left[M_{\tilde{\rho}(\cdot)}\left(\kappa_1\tilde{\alpha}Z_1 + \tilde{\sigma}Z_2, \frac{(1-\lambda)\tilde{\nu}^{(1)} + \lambda\tilde{\nu}^{(2)}}{r}\right)\right]$$

$$= \mathbb{E}\left[M_{\tilde{\rho}(\cdot)}\left(\sqrt{\kappa_1^2\tilde{\alpha}^2 + \tilde{\sigma}^2}Z, \frac{(1-\lambda)\tilde{\nu}^{(1)} + \lambda\tilde{\nu}^{(2)}}{r}\right)\right]$$

$$(Eq.(44)) \quad \leq \mathbb{E}\left[M_{\tilde{\rho}(\cdot)}\left((1-\lambda)\|D\xi_1\|_2 Z + \lambda\|D\xi_2\|_2 Z, \frac{(1-\lambda)\tilde{\nu}^{(1)} + \lambda\tilde{\nu}^{(2)}}{r}\right)\right]$$

$$(\textbf{Step 1: strict convexity}) \quad < (1-\lambda)\mathbb{E}\left[M_{\tilde{\rho}(\cdot)}\left(\|D\xi_1\|_2 Z, \frac{\tilde{\nu}^{(1)}}{r}\right)\right] + \lambda\mathbb{E}\left[M_{\tilde{\rho}(\cdot)}\left(\|D\xi_2\|_2 Z, \frac{\tilde{\nu}^{(2)}}{r}\right)\right]$$

$$= (1-\lambda)\mathbb{E}\left[M_{\tilde{\rho}(\cdot)}\left(\kappa\alpha^{(1)}Z_1 + \sigma^{(1)}Z_2, \frac{\tilde{\nu}^{(1)}}{r}\right)\right]$$

$$+ \lambda\mathbb{E}\left[M_{\tilde{\rho}(\cdot)}\left(\kappa\alpha^{(2)}Z_1 + \sigma^{(2)}Z_2, \frac{\tilde{\nu}^{(2)}}{r}\right)\right]$$

Summarize Scenario 1-3 and Case 1-2, we have shown the strict convexity of $\mathcal{R}(\sigma, r, \tilde{\nu}, \alpha)$ for fixed $r > 0$.

**Step (2): strict concavity**

We aim to show that for fixed $\sigma, \tilde{\nu}, \alpha$, the function $\mathcal{R}(\sigma, r, \tilde{\nu}, \alpha)$ is strictly concave in $r$. Since all four terms in $\mathcal{R}(\sigma, r, \tilde{\nu}, \alpha)$ is concave in $r$, it suffices to prove the strict concavity in

$$L_1(r) := \mathbb{E}\left(\rho'(-\kappa Z_1)\min_t\left[\tilde{\rho}(t) + \frac{r}{2\tilde{\nu}}(t - (\kappa\alpha Z_1 + \sigma Z_2))^2\right]\right).$$

By dominated convergenc theorem and Lemma C.8, we can show that

$$\frac{dL_1}{dr} = \mathbb{E}\left[ \frac{1}{2r^2}\rho'(-\kappa Z_1)\tilde{\rho}'\left( Prox_{\tilde{\rho}}(\kappa\alpha Z_1 + \sigma Z_2; \frac{\tilde{\nu}}{r}) \right)^2 \right]$$

$$\frac{d^2 L_1}{dr^2} = \mathbb{E}\left[ -\frac{1}{r^4}\rho'(-\kappa Z_1)\tilde{\rho}'\left( Prox_{\tilde{\rho}}(\kappa\alpha Z_1 + \sigma Z_2; \frac{\tilde{\nu}}{r}) \right) \frac{\tilde{\rho}''\left( Prox_{\tilde{\rho}}(\kappa\alpha Z_1 + \sigma Z_2; \frac{\tilde{\nu}}{r}) \right)}{1 + \frac{\tilde{\nu}}{r}\left( Prox_{\tilde{\rho}}(\kappa\alpha Z_1 + \sigma Z_2; \frac{\tilde{\nu}}{r}) \right)} \right] < 0$$

Because the second derivative is strictly negative for all $r > 0$, the function $L_1(r)$ is strictly concave in $r$, and hence so is $\mathcal{R}$.

### C.5. Applying CGMT to connect PO and AO

Recall in the process of simplifying AO, we decompose $\beta$ in (26) and obtain the equality that direction$(\mathbf{P}^\perp \widehat{\beta}^{AO}) = $ direction$(\mathbf{P}^\perp \boldsymbol{g})$. Therefore, the solution of AO can be expressed as

$$\widehat{\beta}^{AO} = \widehat{\sigma}\boldsymbol{\theta}_g + \widehat{\alpha}\kappa\frac{\boldsymbol{\beta}_*}{\|\boldsymbol{\beta}_*\|} \tag{45}$$

where $\|\boldsymbol{\theta}_g\| = 1$ and direction$(\boldsymbol{\theta}_g) = $ direction$(\mathbf{P}^\perp \boldsymbol{g})$, and $\boldsymbol{g} \sim N(0, I_p)$ is independent of $\boldsymbol{\beta}_*$. Based on the convergence of optima $(\widehat{\sigma}, \widehat{r}, \widehat{\tilde{\nu}}, \widehat{\alpha}) \xrightarrow{a.s} (\sigma_*, r_*, \tilde{\nu}_*, \alpha_*)$, we have

$$\langle \widehat{\beta}^{AO}, \frac{\boldsymbol{\beta}_*}{\|\boldsymbol{\beta}_*\|} \rangle \xrightarrow{a.s} \alpha_*\kappa \tag{46}$$

$$\|\mathbf{P}^\perp \widehat{\beta}^{AO}\|_2 \xrightarrow{a.s} \sigma_* \tag{47}$$

To apply the asymptotic convergence of CGMT (Lemma C.2), for any $\epsilon > 0$, we introduce sets $\mathcal{S}_1, \mathcal{S}_2$ as follows:

$$\mathcal{S}_1 = \left\{ \boldsymbol{\beta} \in \mathbb{R}^p : \left| \langle \boldsymbol{\beta}, \frac{\boldsymbol{\beta}_*}{\|\boldsymbol{\beta}_*\|} \rangle - \alpha_*\kappa \right| < \epsilon \right\},$$

$$\mathcal{S}_2 = \left\{ \boldsymbol{\beta} \in \mathbb{R}^p : \left| \|\mathbf{P}^\perp \boldsymbol{\beta}\|_2 - \sigma_* \right| < \epsilon \right\}.$$

The convergence in (46) and (47) guarantees that as $n \to \infty, \widehat{\beta}^{AO} \in \mathcal{S}_j$ with probability 1 for $j \in \{1, 2\}$. To extend such a statement to the PO solution, we will show $\widehat{\beta}^{PO} \in \mathcal{S}_j$ with probability approaching 1 using Lemma C.2. First, we recall the PO in (22), AO in (23), and the scalar optimization in (33).

We start with showing that $\widehat{\beta}^{PO} \in \mathcal{S}_1$ with probability approaching 1. Let $\mathcal{S}_1^c := \mathcal{S}_\beta \backslash \mathcal{S}_1$. Denote $\Phi_{\mathcal{S}_1^c}(\tilde{\mathbf{H}})$ and $\phi_{\mathcal{S}_1^c}(\mathbf{g}, \mathbf{h})$ the optimal loss of the PO and AO, respectively, when the minimization over $\boldsymbol{\beta}$ is constrained over $\boldsymbol{\beta} \in \mathcal{S}_1^c$. In terms of AO, $\boldsymbol{\beta} \in \mathcal{S}_1^c$ is equivalent to put constraints on $\alpha$, we can express $\phi_{\mathcal{S}_1^c}(\mathbf{g}, \mathbf{h})$ as follows under same argument,

$$\phi_{\mathcal{S}_1^c}(\mathbf{g}, \mathbf{h}) = \min_{\substack{0 \leq \sigma \leq c_1, 0 < \tilde{\nu} \leq 6c_1 \\ |\alpha_1| \leq c_1/\kappa \\ |\alpha - \alpha_*|\kappa_1 \geq \epsilon}} \max_{r \geq 0} \mathcal{R}_n(\sigma, r, \tilde{\nu}, \alpha).$$

Recall in Section C.2, we show that $\phi(\mathbf{g}, \mathbf{h}) \xrightarrow{\mathbb{P}} \bar{\phi}$. Follow a similar argument, we can show there exist a constant $\bar{\phi}_{\mathcal{S}_1^c}$, defined as

$$\bar{\phi}_{\mathcal{S}_1^c} := \min_{\substack{0 \leq \sigma \leq c_1, 0 < \tilde{\nu} \leq 6c_1 \\ |\alpha_1| \leq c_1/\kappa \\ |\alpha - \alpha_*|\kappa_1 \geq \epsilon}} \max_{r \geq 0} \mathcal{R}(\sigma, r, \tilde{\nu}, \alpha).$$

such that $\phi_{\mathcal{S}_1^c}(\mathbf{g}, \mathbf{h}) \xrightarrow{\mathbb{P}} \bar{\phi}_{\mathcal{S}_1^c}$. Based on the uniqueness of the optima $(\sigma_*, r_*, \tilde{\nu}_*, \alpha_*)$, we have $\bar{\phi} < \bar{\phi}_{\mathcal{S}_1^c}$. Then based on Lemma C.2, we have

$$\lim_{n \to \infty} \mathbb{P}(\widehat{\beta}^{PO} \in \mathcal{S}_1) = 1. \tag{48}$$

With same argument, we can show that (48) holds for $S_2$. Define $\alpha(p) := \langle \boldsymbol{\beta}_*/\|\boldsymbol{\beta}_*\|_2, \widehat{\boldsymbol{\beta}}^{PO}\rangle/\|\boldsymbol{\beta}_*\|$ and $\sigma(p) := \|\mathbf{P}^\perp \widehat{\boldsymbol{\beta}}^{PO}\|_2$. Since we have proved that the events $\widehat{\boldsymbol{\beta}}^{PO} \in \mathcal{S}_j$ for $j = 1, 2$ happens with probability approaching 1, we arrive at following results:

$$\alpha_{(}p) \xrightarrow{\mathbb{P}} \alpha_*, \tag{49}$$

$$\sigma(p) \xrightarrow{\mathbb{P}} \sigma_*. \tag{50}$$

### C.5.1. PROVING ASYMPTOTICS WITH LOCALLY LIPSCHITZ FUNCTION

Suppose $\frac{1}{p} \sum_{j=1}^p \chi_{\sqrt{p}\beta_{0,j}} \rightsquigarrow \Pi$ for a distribution $\Pi$ with finite second monment. Then based on (49) and (50). We have for any locally Lipschitz function $\Psi$,

$$\frac{1}{p} \sum_{j=1}^p \Psi \left( \sqrt{p}[\widehat{\boldsymbol{\beta}}_{\lambda,j} - \alpha_* \boldsymbol{\beta}_{*,j}], \sqrt{p}\boldsymbol{\beta}_{*,j} \right) \xrightarrow{\mathbb{P}} \mathbb{E}[\Psi(\sigma_* Z, \beta)], \tag{51}$$

where $\beta \sim \Pi$ is independent of $Z \sim N(0,1)$. The proof follows closely along the lines of the argument in the proof of Theorem 3.3 in (Zhao et al., 2022), and is therefore omitted.

### C.6. Additional useful lemma

The next lemma establishes a bound on the norm of a normal random vector.

**Lemma C.3.** *Let $\boldsymbol{Z} \in \mathbb{R}^n$ be a vector of i.i.d. standard normal variables, then we have*

$$\mathbb{P}\left(\|\boldsymbol{Z}\| > 2\sqrt{n}\right) \leq \exp(-n/2)$$

Through the reformulation and transformation of the original optimization problem (3), we will frequently use the following lemma to flip the optimization order:

**Lemma C.4.** *(Sion, 1958, Sion's minimax theorem )*

*Let $X \subset \mathbb{R}^n$ and $Y \subset \mathbb{R}^m$ be convex space and one of which is compact. If $f : X \times Y \to \mathbb{R}$ is a continuous function that is concave-convex, i.e. $f(\cdot, y) : X \to \mathbb{R}$ is concave for fixed $y$, and $f(x, \cdot) : Y \to \mathbb{R}$ is convex for fixed $x$.*

*Then we have that*

$$\sup_{x \in X} \inf_{y \in Y} f(x,y) = \inf_{y \in Y} \sup_{x \in X} f(x,y).$$

The following result is also useful in our proof.

**Lemma C.5.** *Let $K$, $\sigma$, and $V$ be any positive numbers. Let $\boldsymbol{g}$ is a vector with the same dimension as $\boldsymbol{\theta}$ to be minimized. It holds that*

$$\min_{\|\boldsymbol{\theta}\|=1} \max_{r \in (0,V]} \{r\sigma \boldsymbol{g}^\top \boldsymbol{\theta} + rK\} = \max_{r \in (0,V]} \min_{\|\boldsymbol{\theta}\|=1} \{r\sigma \boldsymbol{g}^\top \boldsymbol{\theta} + rK\}$$

The next lemma is useful when we find the optimality condition for the scalar optimization problem.

**Lemma C.6** (Identities for logistic link). *Let $\rho'(t) := \frac{e^t}{1+e^t}$ and $Z_1 \sim N(0,1)$ independently. For any $\kappa > 0$, we have*

$$\mathbb{E}(\rho'(\kappa Z_1)) = \frac{1}{2} \qquad \mathbb{E}(Z_1^2 \rho'(\kappa Z_1)) = \frac{1}{2}$$

*Proof.* Direct consequences of the symmetry of the standard normal distribution. $\square$

**Lemma C.7** (Weighted Integral Inequality). *Let $g_1, g_2, g_3 : \mathcal{Z} \to \mathbb{R}$ be measurable functions, $f : \mathcal{Z} \to \mathbb{R}_+$ be a probability density function, and $q : \mathcal{Z} \to \mathbb{R}_+$ be a positive weight function. Suppose that:*

*1. $g_1(z) \leq g_2(z) + g_3(z)$ for all $z \in \mathcal{Z}$ (pointwise inequality),*

2. $\int g_1(z)f(z)\,dz < \int (g_2(z) + g_3(z))f(z)\,dz$ *(strict integral inequality)*,

3. $q(z) > 0$ *for almost every* $z \in \mathcal{Z}$.

*Then*

$$\int q(z)g_1(z)f(z)\,dz < \int q(z)(g_2(z) + g_3(z))f(z)\,dz. \tag{52}$$

*Proof.* Define $h(z) := g_2(z) + g_3(z) - g_1(z) \geq 0$ pointwise. By assumption (2),

$$\int h(z)f(z)\,dz = \int (g_2(z) + g_3(z))f(z)\,dz - \int g_1(z)f(z)\,dz > 0.$$

Since $h(z) \geq 0$ everywhere and $\int h(z)f(z)\,dz > 0$, the set

$$A := \{z \in \mathcal{Z} : h(z) > 0\}$$

must have positive $f$-measure, i.e., $\int_A h(z)f(z)\,dz > 0$.

For almost every $z \in A$, we have $h(z) > 0$, $q(z) > 0$ (by assumption (3)), and $f(z) > 0$. Therefore, $q(z)h(z)f(z) > 0$ for almost every $z \in A$, which implies

$$\int q(z)h(z)f(z)\,dz = \int_A q(z)h(z)f(z)\,dz + \int_{A^c} q(z)h(z)f(z)\,dz \geq \int_A q(z)h(z)f(z)\,dz > 0.$$

Expanding $h(z)$, we obtain

$$\int q(z)(g_2(z) + g_3(z) - g_1(z))f(z)\,dz > 0,$$

which is equivalent to the desired inequality. $\qquad\square$

The next lemma summarizes the partial derivatives of the Moreau envelope function, which will be used frequently when we derive the system of equations from the first order optimality condition in (37).

**Lemma C.8** ((Rockafellar & Wets, 2009)). *Let* $\Phi : \mathbb{R}^d \to \mathbb{R}$ *be a convex function. For* $\mathbf{v} \in \mathbb{R}^d$ *and* $t \in \mathbb{R}_+$, *the* Moreau envelope *of* $\Phi$ *is defined as*

$$M_\Phi(\mathbf{v}, t) = \min_{\mathbf{x} \in \mathbb{R}^d} \left\{ \Phi(\mathbf{x}) + \frac{1}{2t}\|\mathbf{x} - \mathbf{v}\|^2 \right\},$$

*and the* proximal operator *is the minimizer of this problem, i.e.,*

$$\mathrm{Prox}_{t\Phi}(\mathbf{v}) = \arg\min_{\mathbf{x} \in \mathbb{R}^d} \left\{ t\,\Phi(\mathbf{x}) + \frac{1}{2}\|\mathbf{x} - \mathbf{v}\|^2 \right\}.$$

*The derivatives of the Moreau envelope are given by*

$$\frac{\partial M_\Phi}{\partial \mathbf{v}} = \frac{1}{t}\big(\mathbf{v} - \mathrm{Prox}_{t\Phi}(\mathbf{v})\big), \qquad \frac{\partial M_\Phi}{\partial t} = -\frac{1}{2t^2}\big\|\mathbf{v} - \mathrm{Prox}_{t\Phi}(\mathbf{v})\big\|^2.$$

*In the scalar case, for a convex differentiable function* $\rho : \mathbb{R} \to \mathbb{R}$, *define*

$$\mathrm{Prox}(z; b) \equiv \arg\min_{x \in \mathbb{R}} \left\{ \rho(x) + \frac{1}{2b}(x - z)^2 \right\}, \qquad \Psi(z; b) \equiv b\,\rho'\big(\mathrm{Prox}(z; b)\big).$$

*Then for* $(z, b) \in \mathbb{R} \times \mathbb{R}_+$,

$$\Psi(z; b) = z - \mathrm{Prox}(z; b),$$

*and* $\Psi$ *is differentiable with partial derivatives*

$$\frac{\partial \Psi}{\partial z}(z; b) = \left.\frac{b\,\rho''(x)}{1 + b\,\rho''(x)}\right|_{x=\mathrm{Prox}(z;b)}, \qquad \frac{\partial \Psi}{\partial b}(z; b) = \left.\frac{\rho'(x)}{1 + b\,\rho''(x)}\right|_{x=\mathrm{Prox}(z;b)}.$$

## D. Validity of data splitting method

We begin with case $\boldsymbol{\Sigma} = \mathbf{I}_p$. Since we split the data into two halves, the ratio between sample size and dimension reduce, we use $\sigma_*(\delta)$ and $\alpha_*(\delta)$ to denote the solution of system of equation (5) with half $\delta$. Based on (49) and (50), we have

$$\|(\boldsymbol{I} - \boldsymbol{\beta}_*\boldsymbol{\beta}_*^\top/\|\boldsymbol{\beta}_*\|^2)\widehat{\boldsymbol{\beta}}^{(1)}\| \to \sigma_*(\delta/2), \quad \boldsymbol{\beta}_*^\top\widehat{\boldsymbol{\beta}}^{(1)}/\|\boldsymbol{\beta}_*\|^2 \to \alpha_*(\delta/2) \tag{53}$$

$$\|(\boldsymbol{I} - \boldsymbol{\beta}_*\boldsymbol{\beta}_*^\top/\|\boldsymbol{\beta}_*\|^2)\widehat{\boldsymbol{\beta}}^{(2)}\| \to \sigma_*(\delta/2), \quad \boldsymbol{\beta}_*^\top\widehat{\boldsymbol{\beta}}^{(2)}/\|\boldsymbol{\beta}_*\|^2 \to \alpha_*(\delta/2) \tag{54}$$

Let $\mathbf{P}^\perp = \mathbf{I} - \boldsymbol{\beta}_*\boldsymbol{\beta}_*^\top/\|\boldsymbol{\beta}_*\|^2$. It remains to show that cross product terms varnishs, i.e., $\left\langle \mathbf{P}^\perp\widehat{\boldsymbol{\beta}}^{(1)}, \mathbf{P}^\perp\widehat{\boldsymbol{\beta}}^{(2)} \right\rangle = \left\langle \frac{\mathbf{P}^\perp\widehat{\boldsymbol{\beta}}^{(1)}}{\|\mathbf{P}^\perp\widehat{\boldsymbol{\beta}}^{(1)}\|}, \frac{\mathbf{P}^\perp\widehat{\boldsymbol{\beta}}^{(2)}}{\|\mathbf{P}^\perp\widehat{\boldsymbol{\beta}}^{(2)}\|} \right\rangle \xrightarrow{p} 0$.

Let $\boldsymbol{Z} \sim N(0, \mathbb{I}_p)$, we first assume that

$$\frac{\mathbf{P}^\perp\widehat{\boldsymbol{\beta}}_\lambda}{\|\mathbf{P}^\perp\widehat{\boldsymbol{\beta}}_\lambda\|} \overset{d}{=} \frac{\mathbf{P}^\perp\boldsymbol{Z}}{\|\mathbf{P}^\perp\boldsymbol{Z}\|}. \tag{55}$$

Based on above equation, we can show that

$$\left\langle \frac{\mathbf{P}^\perp\widehat{\boldsymbol{\beta}}_\lambda^{(1)}}{\|\mathbf{P}^\perp\widehat{\boldsymbol{\beta}}_\lambda^{(1)}\|}, \frac{\mathbf{P}^\perp\widehat{\boldsymbol{\beta}}_\lambda^{(2)}}{\|\mathbf{P}^\perp\widehat{\boldsymbol{\beta}}_\lambda^{(2)}\|} \right\rangle \overset{d}{=} \left\langle \frac{\mathbf{P}^\perp\boldsymbol{Z}^{(1)}}{\|\mathbf{P}^\perp\boldsymbol{Z}^{(1)}\|}, \frac{\mathbf{P}^\perp\boldsymbol{Z}^{(2)}}{\|\mathbf{P}^\perp\boldsymbol{Z}^{(2)}\|} \right\rangle$$

$$= \frac{\sqrt{p}}{\|\mathbf{P}^\perp\boldsymbol{Z}^{(1)}\|} \frac{\sqrt{p}}{\|\mathbf{P}^\perp\boldsymbol{Z}^{(2)}\|} \times \frac{1}{p}\left( \boldsymbol{Z}^{(1)\top}\boldsymbol{Z}^{(2)} - \frac{(\boldsymbol{\beta}_*^\top\boldsymbol{Z}^{(1)})(\boldsymbol{\beta}_*^\top\boldsymbol{Z}^{(2)})}{\|\boldsymbol{\beta}_*\|^2} \right) \xrightarrow{a.s.} 0$$

It suffices to show (55). We provide a slightly general version of proof in the below.

We write $\mathbf{P} = \mathbf{A}\mathbf{A}^\top$ where $\mathbf{A}$ is a $p \times 1$ matrix. This projects onto a 1-dimensional subspace of $\mathbb{R}^p$. We write $\mathbf{P}^\perp = \mathbf{B}\mathbf{B}^\top$ where $B$ is a $p \times (p-1)$ matrix. This projects onto the orthogonal complement of the subspace spanned by $\mathbf{A}$, which is $(p-1)$-dimensional. We have $\mathbf{B}^\top\mathbf{A} = \mathbf{0} \in \mathbb{R}^{(p-1)\times 1}$, $\mathbf{A}^\top\mathbf{A} = 1$ and $\mathbf{B}^\top\mathbf{B} = \mathbf{I}_{p-1}$. For any $(p-1)\times(p-1)$ orthonormal matrix $\mathbf{G}$, $\mathbf{B}\mathbf{G}\mathbf{B}^\top$ rotates the subspace spanned by the columns of $\mathbf{B}$.

Consider $\mathcal{U} := \{\mathbf{A}\mathbf{A}^\top + \mathbf{B}\mathbf{G}\mathbf{B}^\top : \mathbf{G}$ is $(p-1)\times(p-1)($ orthinormal matrix $)\}$, the set of all orthonormal matrix $\mathbf{U} \in \mathbb{R}^{p\times p}$ such that $\mathbf{U}\boldsymbol{\beta}_* = \boldsymbol{\beta}_*$, and perform rotation on the unit sphere lying in $span\{\boldsymbol{\beta}_*\}^\perp$. By the isotropy of $N(0, \mathbf{I}_p)$, the distribution of $\frac{\mathbf{P}^\perp\boldsymbol{Z}}{\|\mathbf{P}^\perp\boldsymbol{Z}\|}$ is $\mathcal{U}$-invariant, that is, it is the uniform distribution on the unit sphere lying in $span\{\boldsymbol{\beta}_*\}^\perp$. Therefore, it suffices to show that the distribution of $\frac{\mathbf{P}^\perp\widehat{\boldsymbol{\beta}}_\lambda}{\|\mathbf{P}^\perp\widehat{\boldsymbol{\beta}}_\lambda\|}$ is also $\mathcal{U}$-invariant.

For any $\mathbf{U} \in \mathcal{U}$, there exists an orthonormal matrix $\mathbf{G}$ such that $\mathbf{U} = \mathbf{A}\mathbf{A}^\top + \mathbf{B}\mathbf{G}\mathbf{B}^\top$. We want to show

$$\mathbf{U}\frac{\mathbf{P}^\perp\widehat{\boldsymbol{\beta}}_\lambda}{\|\mathbf{P}^\perp\widehat{\boldsymbol{\beta}}_\lambda\|} \overset{d}{=} \frac{\mathbf{P}^\perp\widehat{\boldsymbol{\beta}}_\lambda}{\|\mathbf{P}^\perp\widehat{\boldsymbol{\beta}}_\lambda\|}.$$

Since $\|\mathbf{P}^\perp\widehat{\boldsymbol{\beta}}_\lambda\| = \|\mathbf{U}\mathbf{P}^\perp\widehat{\boldsymbol{\beta}}_\lambda\|$, it suffices to show that $\mathbf{U}\mathbf{P}^\perp\widehat{\boldsymbol{\beta}}_\lambda \overset{d}{=} \mathbf{P}^\perp\widehat{\boldsymbol{\beta}}_\lambda$.

We first show that $\mathbf{U}\widehat{\boldsymbol{\beta}}_\lambda \overset{d}{=} \widehat{\boldsymbol{\beta}}_\lambda$. Note that $\mathbf{U}\widehat{\boldsymbol{\beta}}_\lambda$ is the estimator in (3) with observed covariates replaced by $\{\mathbf{U}\boldsymbol{X}_i\}_{i=1}^n$. Since $\mathbf{U}$ is orthonormal, both covariates are random vector with i.i.d $N(0,1)$ as entries. Since $\mathbf{U} \in \mathcal{U}$, we have $\boldsymbol{\beta}_*^\top\mathbf{U}\boldsymbol{X}_i = \boldsymbol{\beta}_*^\top\boldsymbol{X}_i$ for $i \leq n$. Therefore, the joint distribution of the new observed data remains the same as the original joint distribution. As a result, the distribution of the $\boldsymbol{\beta}_\lambda$ remains the same, i.e., $\mathbf{U}\widehat{\boldsymbol{\beta}}_\lambda \overset{d}{=} \widehat{\boldsymbol{\beta}}_\lambda$.

Consequently, we derive that

$$\mathbf{U}\widehat{\boldsymbol{\beta}}_\lambda \overset{d}{=} \widehat{\boldsymbol{\beta}}_\lambda \implies \mathbf{B}\mathbf{B}^\top\mathbf{U}\widehat{\boldsymbol{\beta}}_\lambda \overset{d}{=} \mathbf{B}\mathbf{B}^\top\widehat{\boldsymbol{\beta}}_\lambda \implies \mathbf{B}\mathbf{G}\mathbf{B}^\top\widehat{\boldsymbol{\beta}}_\lambda \overset{d}{=} \mathbf{P}^\perp\widehat{\boldsymbol{\beta}}_\lambda.$$

We complete the proof by observing $\mathbf{U}\mathbf{P}^\perp = \mathbf{B}\mathbf{G}\mathbf{B}^\top$.

### D.1. Proof of Theorem 5.2

**For part 1,** according to (8), $\|\Sigma^{1/2}\widehat{\boldsymbol{\beta}}_\lambda^{(1)}\|^2 = \mathrm{Var}(\boldsymbol{X}^\top\widehat{\boldsymbol{\beta}}_\lambda^{(1)} \mid \widehat{\boldsymbol{\beta}}_\lambda^{(1)}) \xrightarrow{\mathbb{P}} \kappa^2\alpha_*^2(\delta/2,\kappa) + \sigma_*^2(\delta/2,\kappa)$. It suffices to show that condition on $\widehat{\boldsymbol{\beta}}_\lambda^{(1)}$,

$$S_1(\widehat{\boldsymbol{\beta}}_\lambda^{(1)}) := \frac{1}{n/2}\sum_{i\in D_2}\left[\boldsymbol{X}_i^\top\widehat{\boldsymbol{\beta}}_\lambda^{(1)} - \frac{1}{n/2}\sum_{i\in D_2}\boldsymbol{X}_i^\top\widehat{\boldsymbol{\beta}}_\lambda^{(1)}\right]^2 \xrightarrow{\mathbb{P}} \mathrm{Var}(\boldsymbol{X}^\top\widehat{\boldsymbol{\beta}}_\lambda^{(1)} \mid \widehat{\boldsymbol{\beta}}_\lambda^{(1)}).$$

Let $m = n/2$. Define $Z_i := X_i^\top\widehat{\boldsymbol{\beta}}_\lambda^{(1)}$ for $i \in D_2$. Because $\widehat{\boldsymbol{\beta}}_\lambda^{(1)}$ is computed using only $D_1$, it is independent of $\{X_i : i \in D_2\}$. Hence, conditional on $\widehat{\beta}^{(1)}$, the random variables $\{Z_i : i \in D_2\}$ are i.i.d. with

$$Z_i \mid \widehat{\boldsymbol{\beta}}_\lambda^{(1)} \sim N\left(0, v_1\right), \quad v_1 := \widehat{\boldsymbol{\beta}}_\lambda^{(1)\top}\Sigma\widehat{\boldsymbol{\beta}}_\lambda^{(1)}.$$

In particular, $\mathbb{E}\left[Z_i \mid \widehat{\beta}^{(1)}\right] = 0, \mathbb{E}\left[Z_i^2 \mid \widehat{\beta}^{(1)}\right] = v_1, \mathbb{E}\left[Z_i^4 \mid \widehat{\beta}^{(1)}\right] = 3v_1^2 < \infty$. So it remains to show:

$$\frac{1}{m}\sum_{i\in D_2}Z_i^2 \xrightarrow{P} v_1 \quad \text{and} \quad \bar{Z}^2 \xrightarrow{P} 0,$$

which is guaranteed by law of large number.

**For part 2,** under similar argument, we can show: condition on $\widehat{\boldsymbol{\beta}}_{-i,\lambda}^{(1)}$ and $\widehat{\boldsymbol{\beta}}_\lambda^{(2)}$

$$\frac{1}{n/2}\sum_{i\in D_1}\left(\boldsymbol{X}_i^\top\widehat{\boldsymbol{\beta}}_{-i,\lambda}^{(1)}\right)\left(\boldsymbol{X}_i^\top\widehat{\boldsymbol{\beta}}_\lambda^{(2)}\right) \xrightarrow{\mathbb{P}} \widehat{\boldsymbol{\beta}}_{-i,\lambda}^{(1)\top}\Sigma\widehat{\boldsymbol{\beta}}_\lambda^{(2)}. \tag{56}$$

The quantity on the RHS is closed to $\widehat{\boldsymbol{\beta}}_\lambda^{(1)\top}\Sigma\widehat{\boldsymbol{\beta}}_\lambda^{(2)}$, we can show that

$$\max_{i\in D_1}\left|\widehat{\boldsymbol{\beta}}_{-i,\lambda}^{(1)\top}\Sigma\widehat{\boldsymbol{\beta}}_\lambda^{(2)} - \widehat{\boldsymbol{\beta}}_\lambda^{(1)\top}\Sigma\widehat{\boldsymbol{\beta}}_\lambda^{(2)}\right| \le \lambda_{\max}(\Sigma)\left\|\widehat{\boldsymbol{\beta}}_\lambda^{(2)}\right\|_2\max_{i\in D_1}\left\|\widehat{\boldsymbol{\beta}}_{-i,\lambda}^{(2)} - \widehat{\boldsymbol{\beta}}_\lambda^{(1)}\right\|_2 = o_p(1), \tag{57}$$

where last equality follows from part 1 in Theorem 3.1 and leave-one-out stability assumption $\max_{i\in D_1}\|\widehat{\boldsymbol{\beta}}_{-i,\lambda}^{(1)} - \widehat{\boldsymbol{\beta}}_\lambda^{(1)}\|_2 \xrightarrow{\mathbb{P}} 0$.

Together with (56), (57) and $\left\langle\Sigma^{1/2}\widehat{\boldsymbol{\beta}}_\lambda^{(1)}, \Sigma^{1/2}\widehat{\boldsymbol{\beta}}_\lambda^{(2)}\right\rangle \xrightarrow{p} \kappa^2\alpha_*^2(\delta/2,\kappa)$, we have

$$\frac{1}{n/2}\sum_{i\in D_1}\left(\boldsymbol{X}_i^\top\widehat{\boldsymbol{\beta}}_{-i,\lambda}^{(1)}\right)\left(\boldsymbol{X}_i^\top\widehat{\boldsymbol{\beta}}_\lambda^{(2)}\right) \xrightarrow{\mathbb{P}} \kappa^2\alpha_*^2(\delta/2,\kappa).$$

### D.2. Proof of Theorem 5.5

*Proof of Theorem 5.5.* Fix $\delta > 2$ and $\lambda > 0$. Let the true signal strength be $\kappa \in \mathrm{int}(\mathcal{K})$.

By the conditions of Theorem 5.2, the empirical quantities in (13) and (14) provide consistent estimators of the corresponding population proxies $F_1(\kappa), F_2(\kappa)$. Equivalently, $\widehat{F}_{1,n} := \frac{1}{2}[S_1(\widehat{\boldsymbol{\beta}}_\lambda^{(1)}) + S_2(\widehat{\boldsymbol{\beta}}_\lambda^{(2)})] - S_3(\widehat{\boldsymbol{\beta}}_\lambda^{(1)}, \widehat{\boldsymbol{\beta}}_\lambda^{(2)})$ and $\widehat{F}_{2,n} := S_3(\widehat{\boldsymbol{\beta}}_\lambda^{(1)}, \widehat{\boldsymbol{\beta}}_\lambda^{(2)})$ satisfy

$$\widehat{F}_{1,n} \xrightarrow{p} F_1(\kappa), \qquad \widehat{F}_{2,n} \xrightarrow{p} F_2(\kappa). \tag{58}$$

By Assumption 5.4, for $j \in \{1, 2\}$, $F_j$ is continuously differentiable and strictly monotone on $\mathcal{K}$. Hence $F_j$ is one-to-one on $\mathcal{K}$ and admits an inverse $F_j^{-1}$ defined on $F_j(\mathcal{K})$. Moreover, strict monotonicity implies that $F_j^{-1}$ is continuous on $F_j(\mathcal{K})$.

Define the estimators (as in (13) and (14)) by

$$\widehat{\kappa}_{1,n} := F_1^{-1}(\widehat{F}_{1,n}), \qquad \widehat{\kappa}_{2,n} := F_2^{-1}(\widehat{F}_{2,n}).$$

Since $\widehat{F}_{j,n} \xrightarrow{p} F_j(\kappa)$ and $F_j^{-1}$ is continuous at $F_j(\kappa)$, the continuous mapping theorem yields

$$\widehat{\kappa}_{j,n} = F_j^{-1}(\widehat{F}_{j,n}) \xrightarrow{p} F_j^{-1}\big(F_j(\kappa)\big) = \kappa, \qquad j \in \{1,2\}.$$

Therefore, both estimators in (13) and (14) are consistent for $\kappa$. $\qquad\qquad\qquad\qquad\qquad\qquad$ □

## E. Additional details on Leave-one-out estimator

### E.1. Approximate Leave-one-out estimator

Without loss of generality, we consider the approximation of $\boldsymbol{X}_i^\top \widehat{\boldsymbol{\beta}}_{-i,\lambda}^{(1)}$ with any $i \in D_1$. To ease the notation, we drop the subscript $\lambda$ and superscript (1). Let $\mathcal{I} = \{1,\ldots,n/2\}$ be the indices of all observations in $D_1$ and $\mathcal{I}_{-i} = \{1,\ldots,i-1,i+1,\ldots,n/2\}$ be the indices of all but the $i$-th observation in $D_1$. Now we can write out the first-order optimality condition for $\widehat{\boldsymbol{\beta}}$ and $\widehat{\boldsymbol{\beta}}_{-i}$:

$$0 = \sum_{j\in\mathcal{I}} \boldsymbol{X}_j \left(Y_j - \rho'\left(\widehat{\boldsymbol{\beta}}^\top \boldsymbol{X}_j\right)\right) - \lambda \sum_{j\in\mathcal{I}} (\widehat{\boldsymbol{\beta}}^\top \boldsymbol{X}_j)\boldsymbol{X}_j,$$

$$0 = \sum_{j\in\mathcal{I}_{-i}} \boldsymbol{X}_j \left(Y_j - \rho'\left(\widehat{\boldsymbol{\beta}}_{-i}^\top \boldsymbol{X}_j\right)\right) - \lambda \sum_{j\in\mathcal{I}_{-i}} (\widehat{\boldsymbol{\beta}}_{-i}^\top \boldsymbol{X}_j)\boldsymbol{X}_j.$$

Taking the difference between these two equations yields

$$0 = \boldsymbol{X}_i \left(Y_i - \rho'\left(\widehat{\boldsymbol{\beta}}^\top \boldsymbol{X}_i\right)\right) + \sum_{j\in\mathcal{I}_{-i}} \boldsymbol{X}_j \left[\rho'\left(\widehat{\boldsymbol{\beta}}_{-i}^\top \boldsymbol{X}_j\right) - \rho'\left(\widehat{\boldsymbol{\beta}}^\top \boldsymbol{X}_j\right)\right]$$

$$+ \lambda \left(\sum_{j\in\mathcal{I}_{-i}} \boldsymbol{X}_j \boldsymbol{X}_j^\top \widehat{\boldsymbol{\beta}}_{-i} - \sum_{j\in\mathcal{I}_{-i}} \boldsymbol{X}_j \boldsymbol{X}_j^\top \widehat{\boldsymbol{\beta}} - \boldsymbol{X}_i \boldsymbol{X}_i^\top \widehat{\boldsymbol{\beta}}\right).$$

Because the difference between $\widehat{\boldsymbol{\beta}}_{-i}$ and $\widehat{\boldsymbol{\beta}}$ is small, so we can well approximate the difference $\rho'\left(\widehat{\boldsymbol{\beta}}_{-i}^\top \boldsymbol{X}_j\right) - \rho'\left(\widehat{\boldsymbol{\beta}}^\top \boldsymbol{X}_j\right)$ using a Taylor expansion of $\rho'$ around $\widehat{\boldsymbol{\beta}}^\top \boldsymbol{X}_j$. In other words, we have

$$0 \approx \boldsymbol{X}_i \left(Y_i - \rho'\left(\widehat{\boldsymbol{\beta}}^\top \boldsymbol{X}_i\right)\right) + \sum_{j\in\mathcal{I}_{-i}} \rho''\left(\widehat{\boldsymbol{\beta}}^\top \boldsymbol{X}_j\right) \boldsymbol{X}_j \boldsymbol{X}_j^\top \left(\widehat{\boldsymbol{\beta}}_{-i} - \widehat{\boldsymbol{\beta}}\right)$$

$$+ \lambda \sum_{j\in\mathcal{I}_{-i}} \boldsymbol{X}_j \boldsymbol{X}_j^\top (\widehat{\boldsymbol{\beta}}_{-i} - \widehat{\boldsymbol{\beta}}) - \lambda \boldsymbol{X}_i \boldsymbol{X}_i^\top \widehat{\boldsymbol{\beta}}$$

To simplify the notation, we introduce the followings for denoting the Hessian matrices appeared in the above display:

$$H = -\sum_{j\in\mathcal{I}} \rho''\left(\widehat{\boldsymbol{\beta}}^\top \boldsymbol{X}_j\right) \boldsymbol{X}_j \boldsymbol{X}_j^\top,$$

$$H_{-i} = -\sum_{j\in\mathcal{I}_{-i}} \rho''\left(\widehat{\boldsymbol{\beta}}^\top \boldsymbol{X}_j\right) \boldsymbol{X}_j \boldsymbol{X}_j^\top,$$

$$S_{-i} = \sum_{j\in\mathcal{I}_{-i}} \boldsymbol{X}_j \boldsymbol{X}_j^\top \quad, \quad S = \sum_{j\in\mathcal{I}} \boldsymbol{X}_j \boldsymbol{X}_j^\top.$$

Admitting this second order approximation, we have

$$\boldsymbol{X}_i \left(Y_i - \rho'\left(\widehat{\boldsymbol{\beta}}^\top \boldsymbol{X}_i\right)\right) \approx (H_{-i} - \lambda S_{-i}) \left(\widehat{\boldsymbol{\beta}}_{-i} - \widehat{\boldsymbol{\beta}}\right) + \lambda \boldsymbol{X}_i \boldsymbol{X}_i^\top \widehat{\boldsymbol{\beta}},$$

or

$$\left(\widehat{\boldsymbol{\beta}}_{-i} - \widehat{\boldsymbol{\beta}}\right) \approx (H_{-i} - \lambda S_{-i})^{-1} \left[\boldsymbol{X}_i \left(Y_i - \rho'\left(\widehat{\boldsymbol{\beta}}^\top \boldsymbol{X}_i\right)\right) - \lambda \boldsymbol{X}_i \boldsymbol{X}_i^\top \widehat{\boldsymbol{\beta}}\right].$$

Therefore, we can approximate the term $\widehat{\boldsymbol{\beta}}_{-i}^\top \boldsymbol{X}_i$ by

$$\tilde{l}_i := \widehat{\boldsymbol{\beta}}^\top \boldsymbol{X}_i + \boldsymbol{X}_i^\top (H_{-i} - \lambda S_{-i})^{-1} \left[\boldsymbol{X}_i \left(Y_i - \rho'\left(\widehat{\boldsymbol{\beta}}^\top \boldsymbol{X}_i\right)\right) - \lambda \boldsymbol{X}_i \boldsymbol{X}_i^\top \widehat{\boldsymbol{\beta}}\right]. \tag{59}$$

The derivation above involves an matrix inversion for each $i \in \mathcal{I}$. To obtain the inverse of $(H_{-i} - \lambda S_{-i})$ for all $i \in \mathcal{I}$ efficiently, we can take advantage of the fact that they are each a rank one update from the

$$H_{-i} - \lambda S_{-i} = H - \lambda S + \left[\lambda + \rho''\left(\widehat{\boldsymbol{\beta}}^\top \boldsymbol{X}_i\right)\right] \boldsymbol{X}_i \boldsymbol{X}_i^\top$$

Applying the Sherman-Morrison inverse formula, we have for each $i$:

$$(H_{-i} - \lambda S_{-i})^{-1} = (H - \lambda S)^{-1} - \frac{(H - \lambda S)^{-1}\left[\lambda + \rho''\left(\widehat{\boldsymbol{\beta}}^\top \boldsymbol{X}_i\right)\right] \boldsymbol{X}_i \boldsymbol{X}_i^\top (H - \lambda S)^{-1}}{1 + \left[\lambda + \rho''\left(\widehat{\boldsymbol{\beta}}^\top \boldsymbol{X}_i\right)\right] \boldsymbol{X}_i^\top (H - \lambda S)^{-1} \boldsymbol{X}_i}. \tag{60}$$

### E.2. A simple experiment to justify approximation

We conduct a single experiment with the following setup. The dimension is $p = 400$, sample size $n = 1600$, $\boldsymbol{\Sigma}$ and $\boldsymbol{\beta}_*$ are generated in same way described in Appendix A.4 with $\|\boldsymbol{\beta}_*\|_2 = 1$. The regularization parameter is set to $\lambda = 0.1$.

Table 11 compares the real leave-one-out (LOO) quantities $\{\boldsymbol{X}_i^\top \widehat{\boldsymbol{\beta}}_{-i,\lambda}^{(1)}, i \in D_1\}$ and $\{\boldsymbol{X}_i^\top \widehat{\boldsymbol{\beta}}_{-i,\lambda}^{(2)}, i \in D_2\}$ with their approximation (59). We present first 10 LOO quantities in $\{\boldsymbol{X}_i^\top \widehat{\boldsymbol{\beta}}_{-i,\lambda}^{(1)}, i \in D_1\}$ and $\{\boldsymbol{X}_i^\top \widehat{\boldsymbol{\beta}}_{-i,\lambda}^{(2)}, i \in D_2\}$ with their approximations. The approximations closely track the exact LOO values, with discrepancies that are uniformly small and negligible relative to the scale of the signals. This numerical evidence supports the accuracy of the proposed approximation.

*Table 11.* Comparison between real LOO quantities and their approximations (first 10 observations).

| Index $i$ | First split | | Second split | |
|---|---|---|---|---|
| | Real LOO | Approximation | Real LOO | Approximation |
| 1 | 0.2471214 | 0.2477251 | 2.4368342 | 2.4367580 |
| 2 | -1.0661952 | -1.0661990 | -0.0031643 | -0.0033081 |
| 3 | -0.1917419 | -0.1920940 | -0.5036449 | -0.5035390 |
| 4 | -0.6970390 | -0.6971530 | -4.1231264 | -4.1215712 |
| 5 | 1.2740254 | 1.2739269 | -0.7974870 | -0.7974673 |
| 6 | 0.8147184 | 0.8129966 | -0.6329518 | -0.6329617 |
| 7 | 1.8356401 | 1.8355068 | -1.6076504 | -1.6076396 |
| 8 | 1.2957842 | 1.2956204 | -0.7227844 | -0.7226167 |
| 9 | -4.3480549 | -4.3455375 | -2.3697001 | -2.3653737 |
| 10 | 3.5038383 | 3.5031352 | -0.5134615 | -0.5131449 |

### E.3. Sufficient Condition on Leave-One-Out Stability

We establish leave-one-out (LOO) stability for the generalized ridge estimator and prove that it holds with high probability. To keep the presentation concise, we first present the proof for the case $\Sigma = \mathrm{Cov}(X) = I_p$ in Section E.3.1, and then extend the result to general covariance matrices in Section E.3.2.

### E.3.1. CASE $\Sigma = I_p$

Without loss of generality, we establish the stability result by removing the first data point. Recall that the sample size is $n$, the dimension is $p$, with aspect ratio $n/p = \delta > 1$.

**Key lemma.** Using a Taylor expansion and concentration inequalities for (i) the singular values of the sample covariance matrix and (ii) $\|X_1\|_2$, we obtain the following result:

**Lemma E.1** (Single-point LOO stability). *Let $\widehat{\boldsymbol{\beta}}_\lambda$ be the generalized ridge estimator and $\widehat{\boldsymbol{\beta}}_{\lambda,-1}$ be the estimator without the first sample point. Then*

$$\mathbb{P}\left( \|\widehat{\boldsymbol{\beta}}_{\lambda,-1} - \widehat{\boldsymbol{\beta}}_\lambda\|_2 > \frac{3p^{5/6}}{(1 - \sqrt{1/\delta})^2 n} \right) \leq 6 \exp\left( -\frac{1}{2}\sqrt{p} \right).$$

Applying Lemma E.1 together with a union bound, we obtain that

$$\max_{i \in [n]} \|\widehat{\boldsymbol{\beta}}_{\lambda,-i} - \widehat{\boldsymbol{\beta}}_\lambda\|_2 \leq \frac{3p^{5/6}}{(1 - \sqrt{1/\delta})^2 n}$$

with probability at least $1 - 6\exp(-0.5\sqrt{p} + \log n)$.

**Proof of Lemma E.1.** We begin with the first-order optimality conditions for the full-sample and leave-one-out estimators:

$$\mathbf{0} = \sum_{j=1}^n \boldsymbol{X}_j \left( Y_j - \rho'\left(\widehat{\boldsymbol{\beta}}_\lambda^\top \boldsymbol{X}_j\right) \right) - \lambda \sum_{j=1}^n \left(\widehat{\boldsymbol{\beta}}_\lambda^\top \boldsymbol{X}_j\right) \boldsymbol{X}_j, \tag{61}$$

$$\mathbf{0} = \sum_{j=2}^n \boldsymbol{X}_j \left( Y_j - \rho'\left(\widehat{\boldsymbol{\beta}}_{\lambda,-1}^\top \boldsymbol{X}_j\right) \right) - \lambda \sum_{j=2}^n \left(\widehat{\boldsymbol{\beta}}_{\lambda,-1}^\top \boldsymbol{X}_j\right) \boldsymbol{X}_j. \tag{62}$$

Taking the difference between (61) and (62) and applying a Taylor expansion to $\rho'$, we obtain

$$\mathbf{0} = \boldsymbol{X}_1 \left( Y_1 - \rho'\left(\widehat{\boldsymbol{\beta}}_\lambda^\top \boldsymbol{X}_1\right) \right) + \sum_{j=2}^n \boldsymbol{X}_j \boldsymbol{X}_j^\top \rho''(t_j) \left(\widehat{\boldsymbol{\beta}}_{\lambda,-1} - \widehat{\boldsymbol{\beta}}_\lambda\right)$$

$$+ \lambda \sum_{j=1}^n \boldsymbol{X}_j \boldsymbol{X}_j^\top \left(\widehat{\boldsymbol{\beta}}_{\lambda,-1} - \widehat{\boldsymbol{\beta}}_\lambda\right) - \lambda \boldsymbol{X}_1 \boldsymbol{X}_1^\top \widehat{\boldsymbol{\beta}}_{\lambda,-1},$$

where $t_j$ lies on the line segment between $\widehat{\boldsymbol{\beta}}_\lambda^\top \boldsymbol{X}_j$ and $\widehat{\boldsymbol{\beta}}_{\lambda,-1}^\top \boldsymbol{X}_j$.

Define the positive definite matrices

$$H_{-1,\rho} = \sum_{j=2}^n \boldsymbol{X}_j \boldsymbol{X}_j^\top \rho''(t_j), \qquad H = \sum_{j=1}^n \boldsymbol{X}_j \boldsymbol{X}_j^\top.$$

Since $\rho(t) = \log(1 + e^t)$, we have $0 < \rho''(x) \leq 1/4$, which implies

$$H \prec H_{-1,\rho} + \lambda H \prec 2H.$$

Consequently,

$$\frac{1}{2} H^{-1} \prec (H_{-1,\rho} + \lambda H)^{-1} \prec H^{-1},$$

and therefore

$$\|\widehat{\boldsymbol{\beta}}_{\lambda,-1} - \widehat{\boldsymbol{\beta}}_\lambda\|_2 \leq \frac{\|\boldsymbol{X}_1\|_2 \cdot |\boldsymbol{X}_1^\top \widehat{\boldsymbol{\beta}}_{\lambda,-1}|}{\lambda_{\min}(H)} + \frac{\|\boldsymbol{X}_1\|_2}{2\lambda_{\min}(H)}. \tag{63}$$

The proof is completed by taking $t = p^{1/3}$ in the following concentration lemmas.

**Lemma E.2** (Gaussian concentration for linear forms). *Let $b \in \mathcal{B}_L$ be a random vector independent of $\boldsymbol{X}_1 \sim N(0, I_p)$. Then for any $t > 0$,*

$$\mathbb{P}\left( |\boldsymbol{X}_1^\top b| > t \right) \leq 2\exp\left( -\frac{t^2}{2L} \right).$$

**Lemma E.3** (Extreme eigenvalues of Wishart matrix). *Let* $X_j \overset{i.i.d.}{\sim} N(0, I_p)$ *for* $j \in [n]$, *and* $n/p = \delta > 1$. *Then for any* $t > 0$,

$$(1 - \sqrt{1/\delta} - t/\sqrt{n})^2 \leq \lambda_{\min}\left(\frac{1}{n}\sum_{j=1}^n X_j X_j^\top\right) \leq \lambda_{\max}\left(\frac{1}{n}\sum_{j=1}^n X_j X_j^\top\right) \leq (1 + \sqrt{1/\delta} + t/\sqrt{n})^2$$

*holds with probability at least* $1 - 2\exp(-t^2)$.

**Lemma E.4** (Norm of Gaussian vector). *Let* $X_1 \sim N(0, I_p)$. *Then for any* $t > 0$,

$$\|X_1\|_2 \leq \sqrt{p} + t$$

*with probability at least* $1 - \exp(-t^2/2)$.

E.3.2. EXTENSION TO GENERAL $\Sigma$

We now extend the result to general covariance matrices $\Sigma$ satisfying $0 < c_1 \leq \sigma_{\min}(\Sigma) \leq \sigma_{\max}(\Sigma) \leq c_2 < \infty$.

The derivation follows the same structure as in Section E.3.1. Define

$$H_{-1,\rho} = \sum_{j=2}^n X_j X_j^\top \rho''(t_j), \qquad H = \sum_{j=1}^n X_j X_j^\top.$$

Since $\rho(t) = \log(1 + e^t)$ implies $0 < \rho''(x) \leq 1/4$, we again have

$$H \prec H_{-1,\rho} + \lambda H \prec 2H,$$

which yields

$$\frac{1}{2}H^{-1} \prec (H_{-1,\rho} + \lambda H)^{-1} \prec H^{-1}.$$

Consequently,

$$\|\widehat{\boldsymbol{\beta}}_{\lambda,-1} - \widehat{\boldsymbol{\beta}}_\lambda\|_2 \leq \frac{\|X_1\|_2 \cdot |X_1^\top \widehat{\boldsymbol{\beta}}_{\lambda,-1}|}{\sigma_{\min}(H)} + \frac{\|X_1\|_2}{2\sigma_{\min}(H)}. \tag{64}$$

The proof is completed by taking $t = p^{1/3}$ in the following modified lemmas, which account for the general covariance structure via the transformation $X_j = \Sigma^{1/2} Z_j$ with $Z_j \sim N(0, I_p)$.

**Lemma E.5** (Gaussian concentration for linear forms, general $\Sigma$). *Let* $b \in \mathcal{B}_L$ *be a random vector independent of* $X_1 \sim N(0, \Sigma)$. *Then for any* $t > 0$,

$$\mathbb{P}\left(|X_1^\top b| > t\right) = \mathbb{P}\left(|Z_1^\top \Sigma^{1/2} b| > t\right) \leq 2\exp\left(-\frac{t^2}{2c_2 L}\right),$$

*where we use* $\|\Sigma^{1/2} b\|_2 \leq \sqrt{c_2} L$ *and* $Z_1 \sim N(0, I_p)$.

**Lemma E.6** (Extreme eigenvalues, general $\Sigma$). *Let* $Z_j \overset{i.i.d.}{\sim} N(0, I_p)$ *for* $j \in [n]$, *and* $n/p = \delta > 1$. *Then for any* $t > 0$,

$$\sigma_{\min}\left(\frac{1}{n}\sum_{j=1}^n Z_j Z_j^\top\right) \geq (1 - \sqrt{1/\delta} - t/\sqrt{n})^2$$

*holds with probability at least* $1 - 2\exp(-t^2)$. *The result for* $H = \sum_{j=1}^n X_j X_j^\top$ *follows from the inequality* $\lambda_{\min}(ABA) \geq \lambda_{\min}(B) \cdot \lambda_{\min}^2(A)$ *applied with* $A = \Sigma^{1/2}$ *and* $B = \frac{1}{n}\sum_{j=1}^n Z_j Z_j^\top$.

**Lemma E.7** (Norm of Gaussian vector, general $\Sigma$). *Let* $Z_1 \sim N(0, I_p)$. *Then for any* $t > 0$,

$$\|Z_1\|_2 \leq \sqrt{p} + t$$

*with probability at least* $1 - \exp(-t^2/2)$. *The result for* $X_1 = \Sigma^{1/2} Z_1$ *follows from* $\|\Sigma^{1/2} Z_1\|_2 \leq \sqrt{c_2}\|Z_1\|_2$.

## F. Predictive classification error analysis

We add analysis of predictive classification error(CE) for prediction $\widehat{Y} = \mathbf{1}\{\boldsymbol{X}_T^\top \widehat{\boldsymbol{\beta}}_\lambda \geq 0\}$ where $\boldsymbol{X}_T$ is new covariate. Specifically, we first derive the limit of CE evaluated on test data, and then conduct experiments to illustrate the effect of lambda. To make presentation clean, we consider $\Sigma = I_p$ in the following analysis of CE.

Let $(\boldsymbol{X}_T, Y_T)$ be a pair of future data sampled from the same population as the observed data, i.e., $\boldsymbol{X}_T \sim N(0, \mathbb{I}_p), Y_T \sim Bern(\rho'(\boldsymbol{X}_T^\top \boldsymbol{\beta}_*))$. Given the covariate vector $\boldsymbol{X}_T$ and the estimator $\widehat{\boldsymbol{\beta}}_\lambda$, the binary prediction is given by $\widehat{Y} = \mathbf{1}\{\boldsymbol{X}_T^\top \widehat{\boldsymbol{\beta}}_\lambda \geq 0\}$. We will use $\mathbb{E}_T$ to denote the expectation w.r.t. $(\boldsymbol{X}_T, Y_T)$. Therefore, $\mathbb{E}_T[\mathbf{1}\{\widehat{Y} \neq Y_T\}]$ is a random variable where randomness comes from $\widehat{\boldsymbol{\beta}}_\lambda$.

We first simplify $\mathbb{E}_T[\mathbf{1}\{\widehat{Y} \neq Y_T\}]$ as follows:

$$
\begin{aligned}
\mathbb{E}_T[\mathbf{1}\{\widehat{Y} \neq Y_T\}] &= \mathbb{E}_{\boldsymbol{X}_T}\left[\mathbb{E}_T\left(\mathbf{1}\left\{Y_T \neq \mathbf{1}\{\boldsymbol{X}_T^\top \widehat{\boldsymbol{\beta}}_\lambda \geq 0\}\right\} \mid \boldsymbol{X}_T\right)\right] \\
&= \mathbb{E}_{\boldsymbol{X}_T}\left[\rho'(\boldsymbol{X}_T^\top \boldsymbol{\beta}_*)\mathbf{1}\{\boldsymbol{X}_T^\top \widehat{\boldsymbol{\beta}}_\lambda < 0\} + (1 - \rho'(\boldsymbol{X}_T^\top \boldsymbol{\beta}_*))\mathbf{1}\{\boldsymbol{X}_T^\top \widehat{\boldsymbol{\beta}}_\lambda \geq 0\}\right].
\end{aligned}
\tag{65}
$$

The evaluation of the second equation in above equation relies on the following characterizations of $\boldsymbol{X}_T^\top \boldsymbol{\beta}_*$ and $\boldsymbol{X}_T^\top \widehat{\boldsymbol{\beta}}_\lambda$. Let $Z_1, Z_2$ be two independent standard normal random variables. We introduce two random variables:

$$
W_1 := \|\boldsymbol{\beta}_*\|_2 Z_1,
$$

$$
W_2 := \frac{1}{\|\boldsymbol{\beta}_*\|_2}\boldsymbol{\beta}_*^\top \widehat{\boldsymbol{\beta}}_\lambda Z_1 + \sqrt{\|\widehat{\boldsymbol{\beta}}_\lambda\|_2^2 - \left(\frac{1}{\|\boldsymbol{\beta}_*\|_2}\boldsymbol{\beta}_*^\top \widehat{\boldsymbol{\beta}}_\lambda\right)^2} Z_2.
$$

This construction of $(W_1, W_2)$ preserves the conditional distribution of $(\boldsymbol{X}_T^\top \boldsymbol{\beta}_*, \boldsymbol{X}_T^\top \widehat{\boldsymbol{\beta}}_\lambda)$ given the actual observed data, i.e.,

$$
\begin{aligned}
W_1 &\sim N(0, \|\boldsymbol{\beta}_*\|^2), \quad W_2 \sim N(0, \|\widehat{\boldsymbol{\beta}}_\lambda\|^2), \quad \mathrm{Cov}(\boldsymbol{X}_T^\top \boldsymbol{\beta}_*, W_2) = \boldsymbol{\beta}_*^\top \widehat{\boldsymbol{\beta}}_\lambda, \\
\boldsymbol{X}_T^\top \boldsymbol{\beta}_* &\sim N(0, \|\boldsymbol{\beta}_*\|^2), \quad \boldsymbol{X}_T^\top \widehat{\boldsymbol{\beta}}_\lambda \sim N(0, \|\widehat{\boldsymbol{\beta}}_\lambda\|^2), \quad \mathrm{Cov}(\boldsymbol{X}_T^\top \boldsymbol{\beta}_*, \boldsymbol{X}_T^\top \widehat{\boldsymbol{\beta}}_\lambda) = \boldsymbol{\beta}_*^\top \widehat{\boldsymbol{\beta}}_\lambda.
\end{aligned}
$$

Since $(W_1, W_2) \overset{D}{=} (\boldsymbol{X}_T^\top \boldsymbol{\beta}_*, \boldsymbol{X}_T^\top \widehat{\boldsymbol{\beta}}_\lambda)$ conditioning on the observed data, we can evaluate the above equation as follows:

$$
\begin{aligned}
&\mathbb{E}_{\boldsymbol{X}_T}\left[\rho'(\boldsymbol{X}_T^\top \boldsymbol{\beta}_*)\mathbf{1}\{\boldsymbol{X}_T^\top \widehat{\boldsymbol{\beta}}_\lambda < 0\} + (1 - \rho'(\boldsymbol{X}_T^\top \boldsymbol{\beta}_*))\mathbf{1}\{\boldsymbol{X}_T^\top \widehat{\boldsymbol{\beta}}_\lambda \geq 0\}\right] \\
=&\mathbb{E}_{(W_1, W_2)}\left[\rho'(W_1)\mathbf{1}\{W_2 < 0\} + (1 - \rho'(W_1)\mathbf{1}\{W_2 \geq 0\}\right] \\
=&\mathbb{E}_{(Z_1, Z_2)}\left[\rho'(\|\boldsymbol{\beta}_*\|_2 Z_1)\mathbf{1}\left\{\frac{1}{\|\boldsymbol{\beta}_*\|_2}\boldsymbol{\beta}_*^\top \widehat{\boldsymbol{\beta}}_\lambda Z_1 + \sqrt{\|\widehat{\boldsymbol{\beta}}_\lambda\|_2^2 - \left(\frac{1}{\|\boldsymbol{\beta}_*\|_2}\boldsymbol{\beta}_*^\top \widehat{\boldsymbol{\beta}}_\lambda\right)^2} Z_2 < 0\right\} \right. \\
&\left. + (1 - \rho'(\|\boldsymbol{\beta}_*\|_2 Z_1)\mathbf{1}\left\{\frac{1}{\|\boldsymbol{\beta}_*\|_2}\boldsymbol{\beta}_*^\top \widehat{\boldsymbol{\beta}}_\lambda Z_1 + \sqrt{\|\widehat{\boldsymbol{\beta}}_\lambda\|_2^2 - \left(\frac{1}{\|\boldsymbol{\beta}_*\|_2}\boldsymbol{\beta}_*^\top \widehat{\boldsymbol{\beta}}_\lambda\right)^2} Z_2 \geq 0\right\}\right] \\
=&\mathbb{E}_{Z_1}\left[\rho'(\|\boldsymbol{\beta}_*\|_2 Z_1)\Phi\left(-\frac{\boldsymbol{\beta}_*^\top \widehat{\boldsymbol{\beta}}_\lambda}{\sqrt{\|\boldsymbol{\beta}_*\|^2\|\widehat{\boldsymbol{\beta}}_\lambda\|^2 - (\boldsymbol{\beta}_*^\top \widehat{\boldsymbol{\beta}}_\lambda)^2}}Z_1\right) \right. \\
&\left. + (1 - \rho'(\|\boldsymbol{\beta}_*\|_2 Z_1)\Phi\left(\frac{\boldsymbol{\beta}_*^\top \widehat{\boldsymbol{\beta}}_\lambda}{\sqrt{\|\boldsymbol{\beta}_*\|^2\|\widehat{\boldsymbol{\beta}}_\lambda\|^2 - (\boldsymbol{\beta}_*^\top \widehat{\boldsymbol{\beta}}_\lambda)^2}}Z_1\right)\right] \\
=& \mathbb{E}_{Z_1}[\rho'(a_1 Z_1)\Phi(-a_2 Z_1)] + \mathbb{E}_{Z_1}[(1 - \rho'(a_1 Z_1))\Phi(a_2 Z_1)],
\end{aligned}
\tag{66}
$$

where we use the shorthands $a_1 := \|\boldsymbol{\beta}_*\|_2$ and $a_2 := \boldsymbol{\beta}_*^\top \widehat{\boldsymbol{\beta}}_\lambda / \sqrt{\|\boldsymbol{\beta}_*\|^2\|\widehat{\boldsymbol{\beta}}_\lambda\|^2 - (\boldsymbol{\beta}_*^\top \widehat{\boldsymbol{\beta}}_\lambda)^2}$ to simplify the notation.

Next we will study the convergence of $\mathbb{E}_{Z_1}[\rho'(a_1 Z_1)\Phi(-a_2 Z_1)]$; the convergence of $\mathbb{E}_{Z_1}[(1 - \rho'(a_1 Z_1))\Phi(a_2 Z_1)]$ can be showed using the same argument. Note that

$$\mathbb{E}_{Z_1}[\rho'(a_1 Z_1)\Phi(-a_2 Z_1)] = \int_{-\infty}^{\infty} [\rho'(a_1 z)\Phi(-a_2 z)\phi(z)dz].$$

We will show $\mathbb{E}_{Z_1}[\rho'(a_1 Z_1)\Phi(-a_2 Z_1)]$ converges in probability to $\mathbb{E}_{Z_1}[\rho'(\kappa Z_1)\Phi(-\alpha_* \kappa/\sigma_* Z_1)]$. Let $\boldsymbol{v} = (v_1, v_2)$ be a two-dimensional vector. We define the function $h(\boldsymbol{v}, z) := \rho'(v_1 z)\Phi(-v_2 z)\phi(z)$, which is continuous with respect to $\boldsymbol{v}$ for any $z \in \mathbb{R}$. Furthermore, $|h(\boldsymbol{v}, z)| \leq \phi(z)$ for any $z \in \mathbb{R}$. By Dominated Convergence Theorem, the function $g(\boldsymbol{v}) := \int_{-\infty}^{\infty} h(\boldsymbol{v}, z)dz$ is continuous with respect to $\boldsymbol{v}$. According to $\|\boldsymbol{\beta}_*\|_2^2 \xrightarrow{\mathbb{P}} \kappa^2, \left\|\widehat{\boldsymbol{\beta}}_\lambda\right\|_2^2 \xrightarrow{\mathbb{P}} \alpha_*^2 \kappa^2 + \sigma_*^2, \frac{\widehat{\boldsymbol{\beta}}_\lambda^\top \boldsymbol{\beta}_*}{\|\widehat{\boldsymbol{\beta}}_\lambda\|_2 \|\boldsymbol{\beta}_*\|_2} \xrightarrow{\mathbb{P}}$ $\frac{\alpha_* \kappa}{\sqrt{\alpha_*^2 \kappa^2 + \sigma_*^2}}$ and applying Slutsky's theorem, we conclude that

$$\boldsymbol{a} := \begin{pmatrix} a_1 \\ a_2 \end{pmatrix} = \begin{pmatrix} \|\boldsymbol{\beta}_*\|_2 \\ \boldsymbol{\beta}_*^\top \widehat{\boldsymbol{\beta}}_\lambda / \sqrt{\|\boldsymbol{\beta}_*\|^2 \|\widehat{\boldsymbol{\beta}}_\lambda\|^2 - (\boldsymbol{\beta}_*^\top \widehat{\boldsymbol{\beta}}_\lambda)^2} \end{pmatrix} \xrightarrow{\mathbb{P}} \begin{pmatrix} \kappa \\ \alpha_* \kappa / \sigma_* \end{pmatrix} := \boldsymbol{a}_*.$$

By the continuous mapping theorem, we have $g(\boldsymbol{a}) \xrightarrow{\mathbb{P}} g(\boldsymbol{a}_*)$, i.e.,

$$\mathbb{E}_{Z_1}[\rho'(a_1 Z_1)\Phi(-a_2 Z_1)] \xrightarrow{\mathbb{P}} \mathbb{E}_{Z_1}[\rho'(\kappa Z_1)\Phi(-\alpha_* \kappa/\sigma_* Z_1)]. \tag{67}$$

Similarly, we can show

$$\mathbb{E}_{Z_1}[(1 - \rho'(a_1 Z_1))\Phi(a_2 Z_1)] \xrightarrow{\mathbb{P}} \mathbb{E}_{Z_1}[(1 - \rho'(\kappa Z_1))\Phi(\alpha_* \kappa/\sigma_* Z_1)]. \tag{68}$$

Based on (65), (66), (67) and (68), the following convergence of the generalization error holds:

$$\mathbb{E}_T[\mathbf{1}\{\widehat{Y} \neq Y_T\}] \xrightarrow{\mathbb{P}} \mathbb{E}_{Z_1}[\rho'(\kappa Z_1)\Phi(-\alpha_* \kappa/\sigma_* Z_1)] + \mathbb{E}_{Z_1}[(1 - \rho'(\kappa Z_1))\Phi(\alpha_* \kappa/\sigma_* Z_1)]. \tag{69}$$

We evaluate the right-hand side using numerical integration.

To investigate the effect of $\lambda$ on CE, we adopt the same experimental setting as in Figure 4. Specifically, we vary $\lambda$ from 0.1 to 1.5 and consider $\delta = 4$ with $\kappa = 1, 2$. The results are presented in table 12:

*Table 12.* Effect of $\lambda$ on limit of predictive classification error

| $\lambda$ | 0.1 | 0.2 | 0.3 | 0.4 | 0.5 | 0.6 | 0.7 | 0.8 | 0.9 | 1.0 | 1.1 | 1.2 | 1.3 | 1.4 | 1.5 |
|---|---|---|---|---|---|---|---|---|---|---|---|---|---|---|---|
| $\kappa = 1$ | 0.396071 | 0.396049 | 0.396050 | 0.396056 | 0.396062 | 0.396067 | 0.396073 | 0.396075 | 0.396077 | 0.396081 | 0.396084 | 0.396089 | 0.396095 | 0.396101 | 0.396097 |
| $\kappa = 2$ | 0.291771 | 0.291989 | 0.292111 | 0.292183 | 0.292229 | 0.292266 | 0.292281 | 0.292303 | 0.292314 | 0.292323 | 0.292333 | 0.292343 | 0.292354 | 0.292366 | 0.292361 |

We note that the effect of $\lambda$ on the absolute value of CE is very small, but the pattern is consistent with that of the squared error in Figure 4. For $\kappa = 1$, the CE exhibits a U-shaped curve, while for $\kappa = 2$, the CE increases monotonically.

