# OpenReview forum: "Signal Strength Estimation in Logistic Regression Using Data Splitting"
_ICML.cc/2026/Conference — ICML 2026 regular_

### Official Review · Reviewer_E6AK · 2026-03-06

**Soundness:** 4
**Presentation:** 4
**Significance:** 3
**Originality:** 3
**Overall Recommendation:** 5
**Confidence:** 3

**Summary:**

This article builds upon work developed in Candes and Sur (2019), related to estimating the “signal strength” $\kappa^2 = \operatorname{Var} X^T \beta_*$ in logistic regression, where $\beta_*$ is the vector of true coefficients and $X \sim N( 0, \Sigma )$. In particular, the authors are concerned with the regime where the numbers of samples $n$ and number of covariates $p$ grow so that $n/p$ converges to a constant strictly larger than $2$. The primary contribution is an estimator that includes regularization to handle the case where the data are separable (and hence the MLE does not exist), along with showing indefinability issues from estimators proposed in prior work (Yadlowsky et al. 2021). This work also allows for the covariance matrix of the data to be unknown.

**Compliance With Llm Reviewing Policy:**

Affirmed.

**Final Justification:**

The authors have addressed all of my concerns. In my view, this paper warrants acceptance, despite other reviewers' concerns. I am happy to discuss this further with the reviewers and AC, as needed.

**Key Questions For Authors:**

1) How should one go about choosing the tolerance level and $\lambda$ in practical settings?
2) Relatedly, how sensitive is performance of this method to choices of these parameters (either empirically or theoretically)?
3) What, precisely, goes wrong in the setting where $0 < n/p \le 2$?
4) The authors introduce Algorithms 1 and 2 (the latter is in the appendix). When should practitioners use one or the other? Is it possible to determine which is better-suited based solely on observed data?

**Limitations:**

yes

**Strengths And Weaknesses:**

Strengths:
1) The submission is quite original in that it (a) highlights limitations of existing methods (b) introduces new methodology to overcome those limitations.
2) The paper addresses estimating signal strength for when the data are separable and the covariance matrix is unknown, which can occur when the signal is strong. This closes an important gap in the literature.
3) The writing is quite clear overall.

Weaknesses:
1) The motivation for the work is poorly introduced. It would be helpful to provide context as to why Candes and Sur (2019) is important enough to warrant further investigation and why it is important to develop theory for estimating the signal strength parameter in the regime considered in this paper.
2) The algorithms require a tolerance level, but the authors offer no guidance as to how it should be selected, not do they mention what tolerance level is used in Section 6, although their algorithm 2 says the tolerance level is set to $0.05$. The choice of $\lambda$ should also be mentioned in the experimental setup.

Typos and minor suggestions
1) As mentioned above, the motivation of the work is a bit weak in its presentation. More explicitly highlighting the issues of the MLE in the moderately high-dimensional regime specifically for practitioners in the introduction section could be useful to a broader audience.
2) “equation" should be pluralized in the statement of Proposition 3.3.
3) $X_{i,j}$ notation used in Proposition 3.3 appears not to have been defined. It is clear from context, more or less, but it would be better to make it explicit.
4) Proposition 3.3 makes some distributional assumptions on entries of $\beta_*$ that don't seem to be addressed explicitly.

---

> ### Author Rebuttal · Authors · 2026-03-29
>
> We thank the reviewer for the positive assessment of our work. Following the reviewer’s suggestions and the identified typos, we have:
>
> - Corrected grammatical errors, clarified notation, and expanded the discussion on the assumptions regarding the entries of $\beta_*$;
> - Added more motivation in the introduction to better highlight the value and context of our work;
> - Clearly specified the tuning parameters used in the numerical experiments.
>
> We now address the reviewer’s concerns.
>
> **Reply to Weakness 2.**
>
> We thank the reviewer for raising this important point. In our initial implementation, we evaluated $F_2(\kappa)$ over a grid on $[0,5]$ with spacing $0.05$ and used linear interpolation to approximate the whole function. However, we agree that this approach is both computationally expensive and unnecessarily ad hoc. Moreover, a fixed tolerance like $\epsilon = 0.05$ may suffice for the example shown but is not guaranteed to be appropriate for other settings. To address this, we suggest replacing the grid search with the bisection method to directly find the root of $F_2(\kappa) - \widetilde{\kappa^2 \alpha_*^2}(\delta / 2, \kappa)=0$. This approach is computationally efficient, eliminating both the need for a tolerance parameter and the concerns associated with grid-based evaluation. We have discussed this in the revised paper accordingly.
>
> **Reply to Key Question 1.**
>
> In practice, the tuning parameter $\lambda$ can be selected via leave-one-out cross-validation. Using the approximation developed in Appendix I of the original submission, we obtain a fast and accurate estimate of $X_i^\top \widehat{\beta}_{-i,\lambda}$. This allows efficient computation of the deviance for each $\lambda$, enabling selection of an optimal value.
>
> **Reply to Key Question 2.**
>
> We conducted a sensitivity analysis for $\lambda$ with values $0.1, 0.2, 0.4, 0.8$, fixing $p = 500$, $\kappa = 2$, and varying $\delta = n/p$. Over 100 independent runs, we report the mean and quantiles of $\hat{\kappa}$ in the table below, which shows that the estimator is relatively insensitive to the choice of $\lambda$.
>
> | $\delta$ | $\lambda$ | $\kappa_{\text{mean}}$ | $\kappa_{\text{median}}$ | $\kappa_{\text{q25}}$ | $\kappa_{\text{q75}}$ |
> |----------|----------|------------------------|--------------------------|------------------------|------------------------|
> | 4 | 0.1 | 2.16 | 2.16 | 1.99 | 2.31 |
> | 4 | 0.2 | 2.10 | 2.10 | 1.93 | 2.26 |
> | 4 | 0.4 | 2.08 | 2.07 | 1.91 | 2.24 |
> | 4 | 0.8 | 2.07 | 2.07 | 1.90 | 2.24 |
> | 8 | 0.1 | 2.08 | 2.07 | 2.02 | 2.15 |
> | 8 | 0.2 | 2.04 | 2.03 | 1.98 | 2.10 |
> | 8 | 0.4 | 2.01 | 2.00 | 1.95 | 2.07 |
> | 8 | 0.8 | 2.00 | 1.99 | 1.94 | 2.06 |
>
> **Reply to Key Question 3.**
>
> The condition on $n/p$ ensures that the split-sample covariance matrices $\hat{\Sigma}^{(1)}$ and $\hat{\Sigma}^{(2)}$ are positive definite, with minimum eigenvalues bounded away from 0.
>
> The motivation is as follows. When the data are separable, there exists a vector $\beta_1$ such that $X_i^\top \beta_1 \geq 0$ for all $i \in \\{Y_i = 1\\}$ and $X_i^\top \beta_1 < 0$ for all $i \in \\{Y_i = 0\\}$. In this case, the negative log-likelihood $\sum_{i=1}^n \log(1 + \exp(-(2Y_i - 1) X_i^\top \beta))$ can be made arbitrarily close to 0 by taking $\beta = c \beta\_1$ with $c \to \infty$, leading to unbounded growth in $\\|\beta\\|$.
>
> To prevent this degeneracy, if the minimum eigenvalue of $\hat{\Sigma}$ satisfies $\lambda_{\text {min }}(\hat{\Sigma})>c_1>0$ and $\lambda$ is bounded away from 0, then
> $\lambda \beta_1^{\top} \hat{\Sigma} \beta_1 \geq \lambda c_1\\|\beta_1\\|^2,$
> which diverges as $\\|\beta_1\\| \rightarrow \infty$, thereby controlling the norm of $\beta$ and preventing the degeneracy.
>
> We note that this condition may be stronger than necessary. Even if $\hat{\Sigma}$ is not positive definite, the regularization term $\lambda c^2 \beta_1^\top \hat{\Sigma} \beta_1$ can still diverge as $c \to \infty$ for a separating direction $\beta_1$. However, analyzing this scenario is technically involved, so we adopt this sufficient condition for clarity and leave its refinement to future work.
>
> **Reply to Key Question 4.**
>
> Thank you for this important question. From Figure 2, we observe that the slope of $F_2$ (left panel) is relatively insensitive to $\delta$, suggesting that Algo 1 performs robustly across different regimes of $n/p$. In contrast, the right panel shows that as $n/p$ increases, the slope of $F_1$ decreases, further favoring Algo 1 in high $\delta$ settings.
> When $n/p$ is small, one practical strategy is to compute estimates from both algorithms and average them to reduce variance. Overall, we recommend using Algo 1 in practice due to its robustness with respect to $n/p$.
>
> We have incorporated the reviewer's suggestions to improve clarity and completeness, including strengthening the motivation, clarifying assumptions and notation, and adding experimental details. We thank the reviewer for the constructive feedback.

---

> > ### Author Rebuttal · Reviewer_E6AK · 2026-03-31
> >
> > Thank you for the clarifications.

---

> > > ### Author Response · Authors · 2026-04-06
> > >
> > > We thank the reviewer for the positive update and for taking the time to reconsider our responses. We are glad that our clarifications have addressed the concerns.

---

### Official Review · Reviewer_Eg8s · 2026-03-07

**Soundness:** 3
**Presentation:** 2
**Significance:** 3
**Originality:** 2
**Overall Recommendation:** 3
**Confidence:** 4

**Summary:**

This paper studies signal strength estimation in moderate-dimensional logistic regression where p/n → constant. The authors analyze a generalized ridge estimator with a non-decomposable, sample-covariance-based regularizer, derive its asymptotics, and propose a data-splitting strategy to estimate proxy quantities (σ²* and κ²α²*). They also claim their method works even when data are separable.

**Compliance With Llm Reviewing Policy:**

Affirmed.

**Final Justification:**

Thank you for the detailed follow-up, including the expanded LOO stability proof for the general covariance case and the additional empirical comparisons. I genuinely appreciate the substantial effort you have put into addressing my remaining concerns. However, while these new results clarify several technical points, my reservations about the computational trade-offs of the data-splitting strategy and the overall significance of the methodological contribution relative to existing baselines remain. Ultimately, I still feel that the work does not quite meet the high bar for acceptance at ICML. Therefore, I will maintain my current score. I wish you the best of luck in refining and publishing this work in the future.

**Key Questions For Authors:**

1Can you prove the LOO stability condition for your generalized ridge estimator? If provable, this would meaningfully strengthen the paper.
2How sensitive is performance to λ and ε? All experiments use λ = 0.1 and ε = 0.05. Please provide systematic experiments across a range of λ values  and discuss whether a data-driven selection rule exists. If misspecification substantially degrades κ̂, the practical value is severely limited.
3Can you quantify the statistical efficiency loss from data splitting relative to full-sample approaches?
4Can you demonstrate the method on any real dataset? The gap between synthetic Gaussian experiments and real applications is substantial.

**Limitations:**

Yes

**Strengths And Weaknesses:**

Soundness:
The analysis of the non-decomposable regularizer (Theorem 3.1) is technically detailed, and the identification of the non-monotonicity issue with η² = κ²α²* + σ²* is a valid theoretical observation. However, several gaps weaken the theoretical completeness. The leave-one-out stability condition in Theorem 5.2 (part 2) is assumed without proof, the consistency of S3 and the entire κ²α²*-based estimator depends on it. The Gaussian design assumption is essential for the analysis. Furthermore, the paper lacks systematic sensitivity analysis for key parameters: all experiments fix λ = 0.1, and the tolerance ε is set without justification.The absence of any parameter sensitivity study is a significant shortcoming.
Presentation:
However, the notation is heavy and notably inconsistent throughout the paper. For instance, the normal distribution is denoted as N in some places and as \mathcal{N} in others.
Significance:
The problem is relevant in principle, but practical scope is narrow. The method requires Gaussian or near-Gaussian designs in the proportional regime. Data splitting halves the effective sample size, yet this efficiency cost is not quantified. All experiments are purely synthetic, there is no demonstration on real data.
Originality:
The analysis of the sample-covariance-weighted ridge regularizer is technically novel. However, each component (data splitting, CGMT, generalized ridge) is well-established.

---

> ### Author Rebuttal · Authors · 2026-03-29
>
> Thank you for your feedback. We address your four questions below and discuss originality at the end.
>
> **Reply to Q1.**
>
> We prove LOO stability holds with high probability. For brevity, we assume $\Sigma=Cov(X)=I_p$; the general case follows by imposing lower(upper) bound on min(max) eigenvalues of $\Sigma$.
>
> WLOG, we establish stability by removing the first data point. With $n / p:=\delta>1$, Taylor expansion and concentration inequalities for the covariance singular values and $\\|X_1\\|_2$ yield:
> $$
> \mathbb{P}\left(\\|\widehat{\beta}\_{\lambda,-1}-\widehat{\beta}\_{\lambda}\\|_2 > \frac{3p^{5/6}}{(1-\sqrt{1/\delta})^2 n} \right)
> \leq 6 \exp\left(-\tfrac{1}{2}\sqrt{p}\right).
> $$
>
> By applying a union bound, we can control $\max_i \\|\widehat{\beta}\_{\lambda,-i}-\widehat{\beta}\_{\lambda}\\|_2$, which establishes LOO stability.
>
> **Reply to Q2.**
> We conduct experiments for $\lambda \in \{0.1, 0.2, 0.4, 0.8\}$ and $\delta = n/p$. We fix $p=500$ and $\kappa=2$. Over 100 independent runs, we report the mean and quantiles of the error $|\hat{\kappa}-\kappa|$. The results are shown below. We observe that the estimation is stable across different values of $\lambda$. Similar behavior is observed for the estimation of $\alpha_*$ and $\sigma_*$.
>
> | $\delta$ | $\lambda$ | mean | median | 25% | 75% |
> |----------|-----------|------|--------|------|------|
> | 4 | 0.1 | 0.2130 | 0.1790 | 0.0725 | 0.3140 |
> | 4 | 0.2 | 0.1940 | 0.1560 | 0.0899 | 0.2760 |
> | 4 | 0.4 | 0.1920 | 0.1450 | 0.0838 | 0.2740 |
> | 4 | 0.8 | 0.1920 | 0.1550 | 0.0779 | 0.2700 |
> | | | | | | |
> | 8 | 0.1 | 0.1010 | 0.0807 | 0.0440 | 0.1550 |
> | 8 | 0.2 | 0.0795 | 0.0655 | 0.0272 | 0.1290 |
> | 8 | 0.4 | 0.0752 | 0.0613 | 0.0243 | 0.1150 |
> | 8 | 0.8 | 0.0761 | 0.0633 | 0.0261 | 0.1160 |
>
> For a data-driven selection rule, we have added this procedure in the revised paper. The idea is to use the LOO approximation (Appendix I in the original submission) to approximate $X_i^{\top} \widehat{\beta}_{-i,\lambda}$, and then compute the deviance for each $\lambda$ to select the optimal value.
>
> We chose $\epsilon = 0.05$ because we initially evaluated $F_2(\kappa)$ over grid points on $[0,5]$ with spacing $0.05$ and used linear interpolation. Evaluation on grid points is unnecessary, we thank the reviewer for pointing out this issue. In practice, one can efficiently find root of $F_2(\kappa) - \widetilde{\kappa^2 \alpha_*^2}(\delta / 2, \kappa)=0$ using the bisection method, which is both accurate and fast, eliminating the need for grid evaluation. We have added a remark in the revised paper to clarify this point.
>
> **Reply to Q3.**
> Our method uses data-splitting method to estimate $\kappa^2 \alpha_*^2(\delta/2,\kappa)$. To gain efficiency, we can leverage full data information by averaging across multiple data splits.
>
> We compare the efficiency our proposed MDS-based method across different numbers of splits. Over 100 independent experiments, we compute MSE $E(\hat{\kappa}-\kappa)^2$. The efficiency is measured relative to the case with 100 splits:
>
> | Number of splits | 1 | 2 | 4 | 8 | 16 | 32 | 64 | 100 |
> |------------------|---|---|---|---|----|----|----|-----|
> | MSE | 0.132 | 0.082 | 0.074 | 0.067 | 0.062 | 0.056 | 0.053 | 0.053 |
> | Eff. | 0.401 | 0.651 | 0.722 | 0.803 | 0.851 | 0.944 | 1.000 | 1.000 |
>
> We find that a single split has about half the efficiency of using many splits. Performance improves with more splits, and around 50 splits are sufficient.
>
> **Reply to Q4.**
>
> We analyze a scRNA-seq dataset to identify genes associated with the glucocorticoid response (GR), following the preprocessing in arXiv:2007.01237. We focus on the top 600 most variable genes ( $p=600, n=2400$ ), standardized to zero mean and unit variance.
>
> We apply our generalized ridge estimator with $\lambda=0.1$. Using Algorithm 1, we estimate the signal strength as $\hat{\kappa}=2.22$. Based on the corresponding parameters $(\kappa=2.22, \delta=n/p=4)$, we construct confidence intervals for the coefficients and select genes whose intervals exclude zero.
>
> In total, we identify 23 significant genes. The first 13 genes overlap with those identified in arXiv:2007.01237 and are supported by the literature:
> SERPINA6, FKBP5, NFKBIA, RPL10, HSPB1, EEF1A1, HSPA1A, BCL6, S100A11, NUPR1, LY6E, DDIT4, TACSTD2.
> The remaining genes (HSPB8, ZNF703, GPSM2, DHCR24, ANXA2, ADIRF, YWHAE, CD9, MARS, CTSD) also show evidence of association with glucocorticoid response or related biological pathways based on existing literature.
>
> In terms of originality, our work provides the first theoretically justified signal strength estimation. We would like to refer the reviewer to the three points listed in the rebuttal for Reviewer Pbew.

---

> > ### Author Rebuttal · Reviewer_Eg8s · 2026-04-03
> >
> > I thank the authors for the additional work. However, Q1: the probability bound is reasonable in direction, but it remains a proof sketch and only covers the isotropic case $\Sigma = I$. The Hessian structure differs nontrivially under general $\Sigma$, and this cannot be dismissed with “follows similarly.” A complete proof should appear in the revision. Q3: the multi-split efficiency table is informative, but it does not address my original question: how does the method compare in both statistical accuracy and computational cost to full-sample competitors (SLOE, ProbeFrontier) that require no splitting? With 50 splits, approximately 100 optimization problems must be solved, so a wall-clock time comparison would be needed. Q4: the real-data analysis is a welcome addition. However, no competing method is applied to the same dataset. The core concerns therefore remain insufficiently addressed, and I maintain my score.

---

> > > ### Author Response · Authors · 2026-04-04
> > >
> > > Thank you for your follow-up questions.
> > >
> > > **Q1**: Now we focus on the general $\Sigma$ with $0<c_1 \leq \sigma_{\min }(\Sigma) \leq \sigma_{\max }(\Sigma) \leq c_2<\infty$. Set positive definite matrix $H\_{-1, \rho}=\sum_{j=2}^n X_j X_j^{\top} \rho^{''}\left(t_j\right)$ for any $t_j \in \mathbb{R}$ and $H=\sum\_{j=1}^n X_j X_j^{\top}$, since $\rho(t)=\log \left(1+e^t\right)$, we have $0<\rho^{''}(x) \leq 1 / 4$, we have $H \prec H_{-1, \rho}+H \prec 2 H$, which implies that $H^{-1} / 2 \prec\left(H_{-1, \rho}+H\right)^{-1} \prec H^{-1}$. Then we have
> > > $$\left\\|\widehat{\beta}\_{\lambda,-1}-\widehat{\beta}\_{\lambda}\right\\|_2 \leq \frac{\\|X_1\\|_2 \|X_1^{\top} \widehat{\beta}\_{\lambda,-1}\| }{\sigma\_{min }(H)} +\frac{ \\| X_1 \\|_2}{2 \lambda} \frac{1}{\sigma\_{min }(H)}$$
> > > The proof is completed by taking $t=p^{1 / 3}$ in the following results.
> > > - Let $b \in \mathcal{B}\_L$ be a (random) vector independent of $X_1 \sim N(0, \Sigma)$, then we have
> > > $$
> > > \mathbb{P}\left(\\|X_1^{\top} b\\|_2>t\right)=\mathbb{P}\left(\\|Z_1^{\top} \Sigma^{1 / 2} b\\|_2>t\right) \leq 2 \exp \left(-\frac{t^2}{2 c_2 L}\right)
> > > $$
> > > here we use $\\|\Sigma^{1 / 2} b\\|_2 \leq \sqrt{c_2} L$ and $Z_1 \sim N\left(0, I_p\right)$.
> > > - Let $Z_j \stackrel{i i d}{\sim} N\left(0, I_p\right)$ for $j \in[n]$, and $n / p=\delta>1$, then we have
> > > $
> > > (1-\sqrt{1 / \delta}-t / \sqrt{n})^2 \leq \sigma_{\min }\left(\frac{1}{n} \sum_{j=1}^n Z_j Z_j^{\top}\right)
> > > $
> > > holds with probability at least $1-2 \exp \left(-t^2\right)$. And next use $\lambda_{\min }(A B A) \geq \lambda_{\min }(B) \cdot \lambda_{\min }^2(A)$ with $A=\Sigma^{1 / 2}$ and $B=\frac{1}{n} \sum_{j=1}^n Z_j Z_j^{\top}$.
> > > - Let $Z_1 \sim N\left(0, I_p\right)$, then $\\|Z_1\\|_2 \leq \sqrt{p}+t$ with probability at least $1-\exp \left(-t^2 / 2\right)$. And next use $\\|\Sigma^{1 / 2} Z_1\\|_2 \leq \sqrt{c_2}\\|Z_1\\|_2$.
> > >
> > > **Q3**: When $n/p = \delta$ is small, the MLE may not exist, and consequently SLOE and ProbeFrontier cannot be used. In the main text of the paper, we demonstrate the performance of our method for estimating the signal strength when the MLE does not exist. Here we compare the three methods for larger $\delta$, where we numerically verify that the MLE exists for all experiments.
> > >
> > > The results presented below show that our data splitting method (with 50 splits) performs equally well as SLOE and ProbeFrontier in most cases. Moreover, our method provides accurate estimates of $\kappa$ regardless of whether the MLE exists.
> > > | δ   | $\kappa$ | p   | DS             | SLOE           | ProbeFrontier  |
> > > |:---:|:--------:|:---:|:--------------:|:--------------:|:--------------:|
> > > | **6** | 1.0 | 400 | 0.063 (0.007) | 0.058 (0.006) | 0.055 (0.006) |
> > > |     |     | 800 | 0.041 (0.005) | 0.043 (0.004) | 0.047 (0.005) |
> > > |     | 1.5 | 400 | 0.072 (0.008) | 0.067 (0.009) | 0.069 (0.009) |
> > > |     |     | 800 | 0.052 (0.005) | 0.049 (0.005) | 0.056 (0.005) |
> > > |     | 2.0 | 400 | 0.093 (0.010) | 0.088 (0.008) | 0.089 (0.009) |
> > > |     |     | 800 | 0.075 (0.007) | 0.066 (0.007) | 0.076 (0.007) |
> > > |     | 2.5 | 400 | 0.120 (0.013) | 0.088 (0.011) | 0.104 (0.009) |
> > > |     |     | 800 | 0.092 (0.010) | 0.079 (0.009) | 0.090 (0.009) |
> > > | **8** | 1.0 | 400 | 0.056 (0.006) | 0.051 (0.005) | 0.068 (0.007) |
> > > |     |     | 800 | 0.035 (0.003) | 0.034 (0.003) | 0.043 (0.004) |
> > > |     | 1.5 | 400 | 0.059 (0.007) | 0.054 (0.006) | 0.066 (0.007) |
> > > |     |     | 800 | 0.045 (0.005) | 0.041 (0.005) | 0.050 (0.005) |
> > > |     | 2.0 | 400 | 0.073 (0.008) | 0.057 (0.007) | 0.078 (0.008) |
> > > |     |     | 800 | 0.059 (0.006) | 0.053 (0.005) | 0.078 (0.008) |
> > > |     | 2.5 | 400 | 0.097 (0.009) | 0.072 (0.009) | 0.090 (0.013) |
> > > |     |     | 800 | 0.079 (0.009) | 0.066 (0.007) | 0.092 (0.010) |
> > >
> > > And you are right, there is a trade-off between stability and computation: more splits require more time to compute.
> > > | Number of Splits | 1 | 2 | 4 | 8 | 16 | 32 | 64 | 100 |
> > > |----------------:|---|---|---|---|----|----|----|-----|
> > > | Time (seconds)   | 0.036 | 0.017 | 0.032 | 0.098 | 0.123 | 0.236 | 0.474 | 0.790 |
> > >
> > > **Q4** Compared with MDS and the knockoff method in the reference below, our method selects 23 genes; Knockoff selects 13 genes; MDS selects 31 genes. All methods identify the following genes:
> > > SERPINA6, FKBP5, NFKBIA, RPL10, HSPB1, EEF1A1, HSPA1A, BCL6, S100A11, NUPR1, LY6E, DDIT4, and TACSTD2.
> > >
> > > Dai, C., Lin, B., Xing, X., and Liu, J. S. A scale-free approach for false discovery rate control in generalized linear models.
> > > Journal of the American Statistical Association, pp. 1–15, 2023.
> > >
> > > We thank the reviewer for asking more detailed question. We would also like to emphasize the novelty and originality of our paper. Credit to Reviewer XnoZ: 'It borrows the idea of data splitting to estimate the signal strength, which is totally new in this area.' To the best of our knowledge, this is the first time a theoretically justified signal strength estimation has been provided.

---

### Official Review · Reviewer_XnoZ · 2026-03-08

**Soundness:** 3
**Presentation:** 4
**Significance:** 3
**Originality:** 3
**Overall Recommendation:** 4
**Confidence:** 2

**Summary:**

This article proposes a logistic regression with regularization method (Eq.1) to address the issue of separable data in high-dimensional logistic regression. The author theoretically proved the upper bound of the difference between the learned parameters and the optimal parameters (Theorem 3.1), and estimated the signal strength using data-split methods. Overall, the job content is very fulfilling, with solid theories and sufficient experiments.

**Compliance With Llm Reviewing Policy:**

Affirmed.

**Final Justification:**

The rebuttal addressed my main concerns and problems.

**Key Questions For Authors:**

1. The author has conducted many theoretical verification experiments and estimated signal strength. Should the accuracy of classification also be considered?

2. Does the regularization term affect the performance of model classification in your proposed method?

3. Can authors further demonstrate the difference between your regularization and L_1 norm/L_2 norm regularization through experiments?

**Limitations:**

Yes

**Strengths And Weaknesses:**

Soundness:
The submission is technically sound. It addresses the basic problem in machine learning community: Logistic regression. And it uses the basic techniques, including regularization and data-split methods. The assumptions of the theorem are common.
Presentation:
The paper is easy to follow: From the background to the related work, the the methods and theoretical guarantees are shown sequentially.

Significance:
It addresses the basic problem in high dimensional logistic regression: when the data is separable, the learned parameter may be vary large. Thus the regularization is proposed to handle this problem.

Originality: It borrows the idea of data-split to estimate the signal strength, which is totally new in this region.

---

> ### Author Rebuttal · Authors · 2026-03-29
>
> Thank you for acknowledging the strengths of our work and for your helpful suggestions. Before addressing your key questions, we would like to refer you to our rebuttal to Reviewer Pbew, specifically Points 1 and 2, where we provide a detailed discussion of the estimation of signal strength along with further explanation of the significance and novelty of our work.
>
> **Reply to Questions 1 and 2.**
> We have added an analysis of the predictive classification error (CE) in the revised paper. Specifically, we derive the asymptotic limit of the CE and conduct experiments to illustrate the effect of $\lambda$. For clarity, we assume $\Sigma = I_p$.
>
> Let $(X_T, Y_T)$ be test data, with $X_T\sim N(0,I_p)$ and $Y_T\sim \mathrm{Bern}(\rho'(X_T^\top \beta_*))$.  Given $\widehat{\beta}\_{\lambda}$, the prediction is  $\widehat{Y} = \mathbf{1}\\{X_T^\top \widehat{\beta}\_{\lambda} \geq 0\\}$.  The CE is
> $$ E_T[1\\{\widehat{Y} \neq Y_T\\}]= E_{X_T} [\rho'(X_T^\top \beta_*) 1\\{X_T^{\top} \widehat{\beta}\_{\lambda} < 0\\}+ (1-\rho'(X_T^\top \beta_*)) 1\\{X_T^{\top} \widehat{\beta}\_{\lambda} \geq 0\\}]. $$
> The analysis is involved so we only present idea. Intuitively, $\widehat{\beta}\_{\lambda} \approx \alpha_*\beta_*+\sigma_* Z$. Conditioning on the data:
> - $X_T^\top\beta_*\sim N(0,\\|\beta_*\\|^2)$
> - $X_T^\top\widehat{\beta}\_{\lambda}\sim N(0,\\|\widehat{\beta}\_{\lambda}\\|^2)$
> - $Cov(X_T^\top \beta_*, X_T^\top \widehat{\beta}\_{\lambda}) = \beta_*^\top \widehat{\beta}\_{\lambda}$
>
> By the limits of these quantities and the continuous mapping theorem, the CE converges to
> $$
> E_{Z_1}[\rho^{\prime}(\kappa Z_1) \Phi(-\frac{\alpha_* \kappa}{\sigma_*} Z_1)]+E_{Z_1}[(1-\rho^{\prime}(\kappa Z_1)) \Phi(\frac{\alpha_* \kappa}{\sigma_*} Z_1)] .
> $$
> with $Z_1\sim N(0,1)$, we evaluate this expression using numerical integration.
>
> To study the effect of $\lambda$, we use the same setting as Figure 4 (vary $\lambda \in [0.1, 1.5]$, $\delta=4$, $\kappa=1,2$):
>
> | $\lambda$ | 0.1 | 0.2 | 0.3 | 0.4 | 0.5 | 0.6 | 0.7 | 0.8 | 0.9 | 1.0 | 1.1 | 1.2 | 1.3 | 1.4 | 1.5 |
> |----------|-----|-----|-----|-----|-----|-----|-----|-----|-----|-----|-----|-----|-----|-----|-----|
> | $\kappa=1$ | 0.396071 | 0.396049 | 0.396050 | 0.396056 | 0.396062 | 0.396067 | 0.396073 | 0.396075 | 0.396077 | 0.396081 | 0.396084 | 0.396089 | 0.396095 | 0.396101 | 0.396097 |
> | $\kappa=2$ | 0.291771 | 0.291989 | 0.292111 | 0.292183 | 0.292229 | 0.292266 | 0.292281 | 0.292303 | 0.292314 | 0.292323 | 0.292333 | 0.292343 | 0.292354 | 0.292366 | 0.292361 |
>
> The effect of $\lambda$ on CE is small in magnitude, but the trend matches the squared error in Figure 4:
> - $\kappa=1$: U-shaped curve
> - $\kappa=2$: monotone increase
> Therefore, we conclude that regularization will affect the CE, and the effect will follow the same trend as we see in the squared error.
>
> **Reply to Question 3.**
> We add experiments comparing our method with $L_1$ and $L_2$ regularization. When $\Sigma = I_p$, all methods perform similarly, so we omit these results.
>
> To highlight differences, we consider anisotropic covariance: $\Sigma = \mathrm{diag}(\lambda_1, \dots, \lambda_p)$ with $\lambda_i \in [1/4, 10]$  with  $\beta_* \propto \cdot (\lambda_1, \dots, \lambda_p)^{-1}$ and $\kappa=\\|\beta_* \\|_2$.
> We compare four regularizations: $\lambda \\|\hat{\Sigma}^{1/2}\beta\\|^2$ , $\lambda \\|\Sigma^{1/2}\beta\\|^2$ , $\lambda \\|\beta\\|_1$ , $\lambda \\|\beta\\|^2$, where $\hat{\Sigma}$ is sample covariance matrix.  The first two are scale-adaptive: they shrink more in high-variance directions (small true coefficients) and less in low-variance directions (large coefficients), aligning with the data-generating process. We compare estimation error in columns 3-6 and CE in columns 7-10.
>
> | $\delta$ | $\kappa$ | $\hat{\Sigma}$ | $\Sigma$ | L1 | L2 | $\hat{\Sigma}$ | $\Sigma$   | L1   | L2  |
> |----------|--------------|---------------|----------|----|----|---------------------|---------------|----------|----------|
> | 2 | 1.0 | 1.01 | 0.765 | 1.68 | 1.14 | 0.437 | 0.423 | 0.444 | 0.447 |
> | 2 | 1.5 | 1.08 | 0.944 | 1.78 | 1.33 | 0.396 | 0.369 | 0.400 | 0.411 |
> | 2 | 2.0 | 1.27 | 1.23 | 1.85 | 1.62 | 0.354 | 0.326 | 0.368 | 0.379 |
> | 3 | 1.0 | 0.725 | 0.645 | 1.00 | 0.881 | 0.408 | 0.404 | 0.417 | 0.425 |
> | 3 | 1.5 | 0.855 | 0.836 | 1.11 | 1.11 | 0.357 | 0.348 | 0.368 | 0.379 |
> | 3 | 2.0 | 1.10 | 1.13 | 1.20 | 1.42 | 0.314 | 0.302 | 0.326 | 0.341 |
> | 4 | 1.0 | 0.603 | 0.573 | 0.773 | 0.763 | 0.400 | 0.392 | 0.408 | 0.416 |
> | 4 | 1.5 | 0.775 | 0.782 | 0.876 | 1.01 | 0.337 | 0.331 | 0.349 | 0.361 |
> | 4 | 2.0 | 1.07 | 1.10 | 0.995 | 1.35 | 0.294 | 0.292 | 0.307 | 0.325 |
>
> These results show that covariance-aware regularization outperforms $L_1$ and $L_2$. This suggests that if we believe the covariate has certain heterogeneity, we can consider data-driven regularization. $\lambda \\|\hat{\Sigma}^{1/2}\beta\\|^2$.
>
> We hope our response addresses the reviewer’s concerns.

---

> > ### Author Rebuttal · Reviewer_XnoZ · 2026-04-03
> >
> > My concerns have been adequately addressed. Thank you for your rebuttal

---

> > > ### Author Response · Authors · 2026-04-06
> > >
> > > We thank the reviewer for the positive update and for taking the time to reconsider our responses. We are glad that our clarifications have addressed the concerns.

---

### Official Review · Reviewer_Pbew · 2026-03-12

**Soundness:** 3
**Presentation:** 2
**Significance:** 2
**Originality:** 2
**Overall Recommendation:** 4
**Confidence:** 2

**Summary:**

The approach in the article is centered around the modeling of the MLE solution for two class separation problem as $\alpha_*$ scaled version of the true solution $\beta_*$ with additive Gaussian uncertainty with scaling $\sigma_*$.

Here both $\beta_*$ and $\sigma_*$ are functions of the true signal level $\kappa^2=Var(\beta_*^T X)$. The article takes the estimation of the signal level estimation to the center to utilize this MLE characterization for analysis and variable selection purposes.


For this purpose, it considers sample covariance weighted ridge estimator setting. An important configuration parameter of interest is $\delta=\frac{n}{p}$ where $n$ is the sample size and $p$ is the data dimension.

The article pursues theoretical analysis to characterize the scaling constants behavior conditioned on $\delta$ and the regularization constant $\lambda$. They show that the previous results on signal estimation may not be valid for low $\delta$ regimes. Based on the analysis framework, they also offer a signal splitting strategy for estimating signal level.

**Compliance With Llm Reviewing Policy:**

Affirmed.

**Key Questions For Authors:**

In my opinion more careful wording and better/careful explanations would make it more easier to follow the theoretical content of the article:

For example, it is not clear what the following statements imply:
"We show that the scaling constants of the resulting estimator are characterized by the solution to a system of equations. The analysis of this data-driven regularization is of independent interest."


Or the statements such as "....$\kappa^2\sigma_*^2$ has a monotone relationship with $\kappa$. ". In addition, the follow up sentence "These observations suggest a new perspective for designing signal strength estimators." How?

**Limitations:**

Yes.

**Strengths And Weaknesses:**

STRENGTHS
- The article seem to offer novel theoretical results related to logistic regression centered around the signal level estimation problem. As I am not familiar with the problem and the existing approaches, I can not fairly judge the contribution level. However, theoretical treatment based on the statistical analysis seems to be solid.


WEAKNESSES

I found the presentation to be not very clear in terms of describing goals and achievements. I believe the authors could make their work more accessible and clarify their contributions better by improving their discussion.

---

> ### Author Rebuttal · Authors · 2026-03-29
>
> We thank the reviewer for acknowledging the significance of our work and for pointing out the unclear presentation. The text has been revised accordingly.
>
> For Contribution 1, the text now reads:
> "We rigorously analyze a generalized ridge estimator $\widehat{\beta}\_{\lambda}$ that uses the sample covariance matrix as the weighting matrix when the dimension scales with the sample size. The analysis of this generalized ridge falls outside the scope of existing analyses, such as those in \citet{salehi2019impact}, because the regularization $R(\beta)$ cannot be written as $\sum_{j=1}^p R\_j(\beta_j)$. We show that (i) the proposed regularization guarantees the existence of the estimator when $\delta>1$, and (ii) the asymptotic normality of the generalized ridge estimator. Specifically, defining the projection matrix  $P=\beta_*\beta_*^\top/\\|\beta_*\\|^2$ we have  $P\widehat{\beta}\_{\lambda}\approx \alpha_* \beta_*$ and  $ (I-P)\widehat{\beta}\_{\lambda}\approx \sigma_* N(0,I/p)$, where $(\alpha_*,\sigma_*)$ are function of $\kappa,\delta$ and regularization level $\lambda$."
>
> For Contribution 2, the text now reads:
> "We investigate the relationship between $\kappa^2 \alpha_*^2 + \sigma_*^2$ and $\kappa$. Our results reveal that this mapping is not one-to-one: distinct values of $\kappa$ can produce identical values of $\kappa^2 \alpha_*^2 + \sigma_*^2$. Consequently, this quantity alone does not uniquely determine $\kappa$, highlights a  limitation of previously proposed strategies, such as those in \citet{yadlowsky2021sloe}. Moreover, we show that (i) as the signal strength $\kappa$ increases, the $\sigma_*$  monotonically decreases, and the decreasing speed becomes larger when $n / p$ is smaller; and (ii) as the signal strength $\kappa$ increases, $\kappa^2 \alpha_*^2$ is monotonically increasing. These observations suggest that (i) there is a one-to-one relationship between $\sigma_*^2$ and $\kappa$, and (ii) there is a one-to-one relationship between $\kappa^2 \alpha_*^2$ and $\kappa$. Therefore, if either $\kappa^2 \alpha_*^2$ or $\sigma_*^2$ can be consistently estimated, then $\kappa$  can be consistently estimated."
>
> We hope our revisions have clarified our contributions. Below, we provide further explanation of the significance and novelty of our work.
>
> 1. For logistic regression with $p \asymp n$ and without low-dimensional structures such as sparsity, the behavior of the MLE is roughly $P\widehat{\beta}\_{MLE}\approx \alpha_* \beta_*$ and  $ (I-P)\widehat{\beta}\_{MLE}\approx \sigma_* N(0,I/p)$, the  constants $(\alpha_*,\sigma_*)$ are solutions to the system of equations (2). Similar conclusions hold for other regularized estimators. To use this theory for methods such as confidence intervals, these constants must be estimated. These constants are solutions to some known system of equations parametrized by the signal strength $\kappa$. Once we know $\kappa$, we know $(\alpha_*,\sigma_*)$. This motivates us to estimate $\kappa$.
>
> 2. Our work addresses a fundamental problem in logistic regression: when $Cov(X_i)=I_p$, we aim to consistently estimate
> $\\|\beta_*\\|\_2$ when $p\asymp n$. The difficulty lies in the minimax lower bound, which implies that no estimator $\hat{\beta}$ is consistent, i.e., $\\|\hat{\beta}-\beta_*\\|^2 \asymp 1$. This is why we seek quantities that have a one-to-one relationship with $\kappa$. In our paper, we identify $\kappa^2 \alpha_*^2$ or $\sigma_*^2$ as the desired quantities for various $\delta$ and we use the data-splitting technique to provide consistent estimation, e.g., $\sigma_*^2=f_{\delta/2}(\kappa)$, then $\hat{\kappa}=f_{\delta/2}^{-1} (\hat{\sigma}_*)$.
>
> Here we cite the comment from reviewer XnoZ: 'It borrows the idea of datasplit to estimate the signal strength, which is totally new in this region.' Building on the consistency of $\kappa^2 \alpha_*^2$, we can consistently estimate $\kappa$. This approach is new and can be directly applied to many M-estimators when $Cov(X_i)=I_p$. To the best of our knowledge, this is the first time a theoretically justified signal strength estimation has been provided.
>
> 3. Because it is common in high dimensions for the MLE to not exist, our other main contribution is to extend the previous strategy to scenarios where the MLE does not exist, while also allowing for a general $\Sigma=Cov\left(X_i\right)$, where $\Sigma$ is positive definite with lower(upper) bound on min(max) eigenvalue. To achieve these goals, we consider the generalized ridge estimator as a proper choice due to (1) its existence under mild conditions, and (2) the fact that using the leave-one-out technique, we do not need to estimate $\Sigma$. Therefore, it is necessary to analyze the exact asymptotic properties of the generalized ridge estimator, which we do in our paper. Regarding the theoretical analysis itself, this is a new result that has not been studied before.
>
> We hope these explanations help clarify the significance and novelty of our work.

---

> > ### Author Rebuttal · Reviewer_Pbew · 2026-04-03
> >
> > I would like to thank the authors for their clarifications.

---

> > > ### Author Response · Authors · 2026-04-06
> > >
> > > We thank the reviewer for the positive update and for taking the time to reconsider our responses. We are glad that our clarifications have addressed the concerns.

---

### Decision · Program_Chairs · 2026-04-30

**Decision:**

Accept (regular)

**Comment:**

The problem of estimating signal strength in moderate/high-dimensional logistic regression is an important one, and a technique that applies when the data are separable and the covariance unknown is welcome. The reviewers agreed that the technical analysis is detailed, and the concerns raised were adequately addressed in the rebuttals. Including the relevant parts of the discussion, such as the LOO stability analysis and clarifications/improvements in presentation (particularly, motivating the problem and situating the paper more clearly in the context of existing work on high-dimensional  logistic regression) would make the final paper stronger and more complete.